# Changes in PM$_{2.5}$ Peat Combustion Source Profiles with Atmospheric Aging in an Oxidation Flow Reactor

Judith C. Chow[1,2*], Junji Cao[2,3], L.-W. Antony Chen[4], Xiaoliang Wang[1], Qiyuan Wang[2,3], Jie Tian[2,3], Steven Sai Hang Ho[1,5], Adam C. Watts[1], Tessa B. Carlson[1], Steven D. Kohl[1], John G. Watson[1,2]

[1]Division of Atmospheric Sciences, Desert Research Institute, Reno, Nevada, USA

[2]Key Laboratory of Aerosol Chemistry and Physics, Institute of Earth Environment, Chinese Academy of Sciences, Xi'an, 710061, China.

[3]CAS Center for Excellence in Quaternary Science and Global Change, Xi'an, 710061, China

[4]Department of Environmental and Occupational Health, University of Nevada, Las Vegas, Nevada, USA

[5]Hong Kong Premium Services and Research Laboratory, Hong Kong, China

Revised and resubmitted to

Atmospheric Measurement Techniques Discussion

Date

19 August 2019

[*]Corresponding Author: judith.chow@dri.edu

**Abstract**

Smoke from laboratory chamber burning of peat fuels from Russia, Siberia, U.S.A. (Alaska and Florida), and Malaysia representing boreal, temperate, subtropical, and tropical regions was sampled before and after passing through a potential aerosol mass-oxidation flow reactor (PAM-OFR) to simulate intermediate-aged (~2 days) and well-aged (~7 days) source profiles. Species abundances in $PM_{2.5}$ between aged and fresh profiles varied by several orders of magnitude with two distinguishable clusters, centered around 0.1% for reactive and ionic species and centered around 10 % for carbon.

Organic carbon (OC) accounted for 58–85 % of $PM_{2.5}$ mass in fresh profiles with low EC abundances (0.67–4.4 %). OC abundances decreased by 20–33 % for well-aged profiles, with reductions of 3–14 % for the volatile OC fractions (e.g., OC1 and OC2, thermally evolved at 140 and 280 °C). Ratios of organic matter (OM) to OC abundances increased by 12–19 % from intermediate- to well-aged smoke. Ammonia ($NH_3$) to $PM_{2.5}$ ratios decreased after intermediate aging.

Well-aged $NH_4^+$ and $NO_3^-$ abundances increased to 7–8 % of $PM_{2.5}$ mass, associated with decreases in $NH_3$, low temperature OC, and levoglucosan abundances for Siberia, Alaska, and Everglades (Florida) peats. Elevated levoglucosan was found for Russian peats, accounting for 35–39 % and 20–25 % of $PM_{2.5}$ mass for fresh and aged profiles, respectively. The water-soluble organic carbon (WSOC) fractions of $PM_{2.5}$ were over two-fold higher in fresh Russian ($37.0 \pm 2.7$ %) than in Malaysian ($14.6 \pm 0.9$ %) peats. While Russian peat OC emissions were largely water-soluble, Malaysian peat emissions were mostly water-insoluble, with WSOC/OC ratios of 0.59–0.71 and 0.18–0.40, respectively.

This study shows significant differences between fresh and aged peat combustion profiles among the four biomes that can be used to establish speciated emission inventories for atmospheric modeling and receptor model source apportionment. A sufficient aging time (~one week) is needed to allow gas-to-particle partitioning of semi-volatilized species, gas-phase oxidation, and particle volatilization to achieve representative source profiles for regional-scale source apportionment.

Keywords: fresh and aged source profiles, atmospheric aging, organic mass, organic carbon,

levoglucosan, oxidation flow reactor (OFR)

## 1 Introduction

Receptor-oriented source-apportionment models have played a major role in establishing the weight of evidence (U.S.EPA, 2007) for pollution control decisions. These models, particularly the different solutions (Watson et al., 2016) to the Chemical Mass Balance (CMB) equations (Hidy and Friedlander, 1971), rely on patterns of chemical abundances in different source types that can be separated from each other when superimposed in ambient samples of volatile organic compounds (VOC) and suspended particulate matter (PM). These patterns, termed "source profiles," have been measured in diluted exhaust emissions and resuspended mineral dusts for a variety of representative emitters. Many of these source profiles are compiled in country-specific source profile data bases (Cao, 2018; CARB, 2019; Liu et al., 2017; Mo et al., 2016; Pernigotti et al., 2016; U.S.EPA, 2019) and have been widely used for source apportionment and speciated emission inventories.

Chemical profiles measured at the source have been sufficient to identify and quantify nearby, and reasonably fresh, source contributions. These source types include gasoline- and diesel-engine exhaust, biomass burning, cooking, industrial processes, and fugitive dust. Ambient VOC and PM concentrations have been reduced as a result of control measures applied to these sources, and additional reductions have been implemented for toxic materials such as lead, nickel, vanadium, arsenic, diesel particulate matter, and several organic compounds. As these fresh emission contributions in neighborhood- and urban-scale environments (Chow et al., 2002) decrease, regional-scale contributions that may have aged for intermediate (~2 days) or long (~7 days) periods prior to arrival at a receptor gain in importance. These profiles experience augmentation and depletion of chemical abundances owing to photochemical reactions among their gases and particles, as well as interactions upon mixing with other source emissions.

Peatland fires produce long-lasting thick smoke that leads to adverse atmospheric, climate, ecological, and health impacts. Smoke from Indonesian and Malaysian peatlands is a major concern in the countries of southeast Asia (Wiggins et al., 2018) and elsewhere; it is transported over long distances. Aged peat smoke profiles are likely to differ from fresh emissions, as well as among the different types of peat in other parts of the world.

Ground-based, aircraft, shipboard, and laboratory peat combustion experiments have been carried out to better represent global peat fire emissions and estimate their environmental impacts (e.g., Akagi et al., 2011; Iinuma et al., 2007; Nara et al., 2017; Stockwell et al., 2014; 2016). Most

peat fire studies report emission factors (EFs) for pyrogenic gases (e.g., methane, carbon
monoxide, and carbon dioxide) and fine particle (PM$_{2.5}$, particles with aerodynamic diameter <2.5
microns) mass, with a few studies reporting EFs for organic and elemental carbon (OC and EC)
(Hu et al., 2018).
Despite this lack of peat-specific fresh and aged source profiles, results have been
published for source apportionment in Indonesia (See et al., 2007), Malaysia (Fujii et al., 2017),
Singapore (Budisulistiorini et al., 2018), and Ireland (Dall'Osto et al., 2013; Kourtchev et al., 2011;
Lin et al., 2019). These have involved sampling under near-source and far from-source dominated
environments, such as the 2015 Indonesia burning episode to determine changes in thermally-
derived carbon fractions with aging (Tham et al., 2019), and inference of aged peat-burning
profiles from positive matrix factorization (PMF) application to chemically-speciated ambient PM
samples (Fujii et al., 2017). Budisulistiorini et al. (2018) observe that "…atmospheric processing
of aerosol particles in haze from Indonesian wildfires has scarcely been investigated. This lack of
study inhibits a detailed treatment of atmospheric processes in the models, including aerosol aging
and secondary aerosol formation."
Changes in source profiles have been demonstrated in large smog chambers (Pratap et al.,
2019), wherein gas/particle mixtures are illuminated with ultraviolet (UV) light for several hours
and their end products are measured. Such chambers are specially constructed and limited to
laboratory testing. A more recent method for simulating such aging is the oxidation flow reactor
(OFR), based on the early studies of Kang et al. (2007), revised and improved by several
researchers (e.g., Jimenez, 2018; Lambe et al., 2011), and commercially available from Aerodyne
(2019a, b). Although the Aerodyne potential aerosol mass (PAM)-OFR has many limitations, as
explained in the supplemental material (Section S.1), it is a practical method for understanding
how profiles might change with different degrees of atmospheric aging. A growing users group
(PAMWiki, 2019) provides increasing knowledge of its characteristics and operations.
Laboratory peat combustion EFs for gaseous carbon and nitrogen species corresponding
with the profiles described here, as well as PM$_{2.5}$ mass and major chemical species (e.g., carbon
and ions), are reported by Watson et al. (2019). The PM$_{2.5}$ speciated source profiles derive from
six peat fuels collected from Odintsovo, Russia; Pskov, Siberia; Northern Alaska and Florida,
U.S.A.; and Borneo, Malaysia; representing boreal, temperate, subtropical, and tropical climate
regions. Comparisons between fresh (diluted and unaged) and aged (represent intermediate-aged
[~2 days] and well-aged [~7 days] laboratory simulated oxidation with an OFR) PM$_{2.5}$ speciated
profiles are made to highlight chemical abundance changes with photochemical aging. The
objectives of this study are to: 1) evaluate similarities and differences among the peat source
profiles from four biomes; 2) examine the extent of gas-to-particle oxidation and volatilization
between 2- and 7-days of simulated atmospheric aging; and 3) characterize carbon and nitrogen
properties in peat combustion emissions.

## 2 Experiment

The supplemental material describes sampling configuration shown in Fig. S1 and OFR
operation. Briefly, peat smoke generated in a laboratory combustion chamber (Tian et al., 2015)
was diluted with clean air (by factors of three to five) to allow for nucleation and condensation at
ambient temperatures (Watson et al., 2012). These diluted emissions were then passed through
an unmodified Aerodyne PAM-OFR in the OFR185 mode without ozone (O$_3$) injection. Hydroxyl
radical (OH) production as a function of UV lamp voltage was estimated by inference from sulfur
dioxide (SO$_2$) decay using well-established rate constants. UV lamps were operated at 2 and 3.5
volts with a flow rate of 10 L min$^{-1}$ and a plug-flow residence time of ~80 s in the 13.3 L anodine-
coated reactor, which translates to OH exposures (OH$_{exp}$) of ~2.6 x 10$^{11}$ and ~8.8 x 10$^{11}$ molecules-
sec cm$^{-3}$ at 2 volts and 3.5 volts, respectively.
Transport times between source and receptor of 1 to 10 days are typical of peat burning
plumes, and the two OH$_{exp}$ estimates were selected to examine intermediate (~2 days) and long-
term (~7 days) atmospheric aging. Other emissions aging experiments (e.g., Bhattarai et al., 2018)
cite Mao et al. (2009) for a 24-hour average atmospheric OH concentration (OH$_{atm}$) of 1.5x10$^6$
molecules cm$^{-3}$. This number appears nowhere in the text of Mao et al. (2009), but it corresponds
to the ground-level median value in Mao's Figure 8 plot of OH vs. altitude for Asian outflows over
the Pacific Ocean. The individual measurements in the plot range from OH$_{atm}$ near-zero to 5.3x10$^6$
molecules cm$^{-3}$. Altshuller (1989) concluded that "The literature contains reports of atmospheric
OH radical concentrations measured during daylight hours ranging from 10$^5$ molecule cm$^{-3}$ to over
10$^8$ molecule cm$^{-3}$, but almost all of the values reported are below 5x10$^7$ molecules cm$^{-3}$." Stone
et al. (2012) report atmospheric values ranging from 1.1x10$^5$ molecules cm$^{-3}$ in polar environments
to 1.5x10$^7$ molecules cm$^{-3}$ in a vegetated forest. Uncertainties in OH$_{exp}$ within the OFR are,
therefore, not the controlling uncertainty in estimating profile aging times. Added to this
uncertainty are reactions among emission constituents that are not embodied in the OFR185 mode
that tend to suppress $OH_{exp}$ with respect to that estimated by the $SO_2$ calibration (Li et al., 2015;
Peng et al., 2015; Peng et al., 2016; Peng and Jimenez, 2017; Peng et al., 2018).  The "OFR
Exposure Estimator" available from the PAMWiki (2019) intends to estimate this $OH_{exp}$, but
detailed VOC from these experiments are insufficient to apply it.  The nominal 2- and 7-day aging
times determined by dividing $OH_{exp}$ by Mao's $1.5x10^6$ molecules $cm^{-3}$ are subject to these
uncertainties, which may increase or decrease the aging time estimates.  However, these
uncertainties, along with other uncertainties related to peat sample selection, moisture content, and
laboratory burning conditions do not negate the value of the measurements reported here.  There
are distinct differences in the fresh, intermediate-aged, and well-aged profiles that address the
concerns expressed by Budisulistiorini et al. (2018).
Forty smoldering-dominated peat combustion tests were conducted that included three to
six tests for each type of peat fuel (Table S1).  The following analysis uses time-integrated (~40–
60 minutes) gaseous and $PM_{2.5}$ filter pack samples collected upstream and downstream of the OFR,
representing fresh and aged peat combustion emissions, respectively.
**2.1    $PM_{2.5}$ mass and chemical analyses**
Measured chemical abundances included $PM_{2.5}$ precursor gases (i.e., nitric acid [$HNO_3$]
and ammonia [$NH_3$]) as well as $PM_{2.5}$ mass and major components (e.g., elements, ions, and
carbon).  Water-soluble organic carbon (WSOC), carbohydrates, and organic acids that are
commonly used as markers in source apportionment for biomass burning were also quantified
(Chow and Watson, 2013; Watson et al., 2016).
The filter pack sampling configurations for the four upstream and two downstream
channels along with filter types and analytical instrument specifications are shown in Fig. 1.
Multiple sampling channels accommodate different filter substrates that allow for comprehensive
chemical speciation.  Additional upstream Teflon-membrane and quartz-fiber filters were taken
for more specific nitrogen and organic compound analyses that are not reported here.  The limited
flow through the OFR precludes additional downstream sampling.
Teflon-membrane filters (i.e., channels one and five in Fig. 1) were submitted for: 1)
gravimetric analysis by microbalance with ±1 μg sensitivity before and after sampling to acquire
$PM_{2.5}$ mass concentrations (Watson et al., 2017); 2) filter light reflectance and transmittance by
ultraviolet/visible (UV/Vis) spectrometer (200–900 nm) equipped with an integrating sphere that
measures transmitted/reflected light at 1 nm interval (Johnson, 2015); 3) 51 elements (i.e., sodium
[Na] to uranium [U]) by energy-dispersive x-ray fluorescence (XRF) analysis (Watson et al.,
1999); and 4) organic functional groups by Fourier Transform Infrared (FTIR) spectrometry.
Results from UV/Vis and FTIR spectrometry will be reported elsewhere.
Half of the quartz-fiber filter (i.e., channels two and six) was analyzed for: 1) four anions
(i.e., chloride [$Cl^-$], nitrite [$NO_2^-$], nitrate [$NO_3^-$], and sulfate [$SO_4^=$]), three cations (i.e., water-
soluble sodium [$Na^+$], potassium [$K^+$], and ammonium [$NH_4^+$]), and nine organic acids (including
four mono- and five di-carboxylic acids) by ion chromatography (IC) with a conductivity detector
(CD) (Chow and Watson, 2017); 2) 17 carbohydrates including levoglucosan and its isomers by
IC with a pulsed amperometric detector (PAD); and 3) WSOC by combustion and non-dispersive
infrared (NDIR) detection.  A portion (0.5 $cm^2$) of the other half quartz-fiber filter was analyzed
for OC, EC, and brown carbon (BrC) by the IMPROVE_A multiwavelength thermal/optical
reflectance/transmittance method (Chen et al., 2015; Chow et al., 2007; 2015b); the IMPROVE_A
protocol (Chow et al., 2007) reports eight operationally defined thermal fractions  (i.e., OC1 to
OC4 evolved at 140, 280, 480, and 580 °C in helium atmosphere; EC1 to EC3 evolved at 580,
740, and 840 °C in helium/oxygen atmosphere; and pyrolyzed carbon [OP]) that further
characterize carbon properties under different combustion and aging conditions.  Citric acid and
sodium chloride impregnated cellulose-fiber filters placed behind the Teflon-membrane and
quartz-fiber filters, respectively, acquired $NH_3$ as $NH_4^+$ and $HNO_3$ as volatilized nitrate,
respectively, with analysis by IC-CD.
Detailed chemical analyses along with quality assurance/quality control (QA/QC)
measures are documented in Chow and Watson (2013).  For each analysis, a minimum of 10 % of
the samples were submitted for replicate analysis to estimate precisions.  Precisions associated
with each concentration were calculated based on error propagation (Bevington, 1969) of the
analytical and sampling volume precisions (Watson et al., 2001).
**2.2   $PM_{2.5}$ source profiles**
Concentrations of two gases (i.e., $NH_3$ and $HNO_3$) and 125 chemical species acquired from
each sample pair (fresh vs. aged) were normalized by the $PM_{2.5}$ gravimetric mass to obtain source
profiles with species-specific fractional abundances.  The following analyses are based on the
average of 24 paired profiles (shown in Table 1), grouped by upstream (fresh) and downstream
(aged) samples for 2- and 7-day aging (i.e., denoted as Fresh 2 vs. Aged 2 and Fresh 7 vs. Aged 7)
for each of the six peats with 25 % fuel moisture.  Composite profiles are calculated based on the
average of individual abundances and the standard deviation of the average within each group
(Chow et al., 2002). Although the standard deviation is termed the source profile abundance
uncertainty, it is really an estimate of the profile variability for the same fuels and burning
conditions, which exceeds the propagated measurement precision.
To assess changes with fuel moisture content, tests of three sets of Putnam (FL1) peats at
60 % fuel moisture were conducted with resulting profiles shown in Table S2. A few samples
were voided due to filter damage or sampling abnormality, which produced five unpaired (either
fresh or aged) individual profiles (Table S3). These profiles are reported as they might be useful
for future source apportionment studies.
**2.3   Equivalence measures**
The Student $t$-test is commonly used to estimate the statistical significance of differences
between chemical abundances. Two additional measures are used to determine the similarities
and differences between profiles: 1) the correlation coefficient ($r$) between the source profile
abundances ($F_{ij}$, the fraction of species i in peat j) divided by the source profile variabilities ($\sigma_{ij}$)
that quantifies the strength of association between profiles; and 2) the distribution of weighted
differences (residual [$R$]/uncertainty [$U$] = [$F_{i1} - F_{i2}$]/[$\sigma^2_{i1} + \sigma^2_{i2}$]$^{0.5}$) for $< 1\sigma$, $1\sigma$-$2\sigma$, $2\sigma$-$3\sigma$, and
$>3\sigma$. The percent distribution of $R/U$ ratios is used to understand how many of the chemical
species differ by multiples of the uncertainty of the difference. These measures are also used in
the effective variance-chemical mass balance (EV-CMB) receptor model solution that uses the
variance ($r^2$) and the $R/U$ ratio to quantify agreement between measured receptor concentrations
and those produced by the source profiles and source contribution estimates (Watson, 2004).
**3   Results and discussion**
**3.1   Similarities and differences among peat profiles**
The equivalence measures are used to provide guidance in compositing and comparing the
40 sets of fresh vs. aged profiles. The first comparison is made between two Florida samples from
locations separated by ~485 km (i.e., Putnam County Lakebed [FL1] and Everglades National Park
[FL2]), representing different geological areas and land uses. Panel A of Table S4 shows that the
two profiles yield high correlations ($r$ >0.994), but are statistically different ($P$ <0.002); with over
93 % of the chemical abundance differences within $\pm 3\sigma$. However, when combining both fresh
Florida profiles (i.e., all Fresh 2 vs. all Fresh 7 in Panel B), statistical differences are not found,
with over 98 % of abundance differences within $\pm 1\sigma$ and $P$ >0.5. Notice that statistical
differences are found between the two fresh Florida profiles (i.e., FL1 Fresh 2 vs. FL2 Fresh 2 and
FL1 Fresh 7 vs. FL2 Fresh 7 in Panel A) with few (< 0.81 % and 5.6 %) R/U ratios exceeding $3\sigma$;
combining the two Florida profiles may cancel out some of the differences. However, paired
comparisons of other combined profiles show statistical differences with low $P$-values ($P < 0.002$).
To further demonstrate the differences, these two Florida profiles are classified as Subtropical 1
and Subtropical 2 to compare with other biomes.
Similarities and differences in peat profiles by biome are summarized in Table 2.
Comparisons are made for: 1) paired fresh vs. aged profiles (i.e., All Fresh vs. All Aged; Fresh 2
vs. Aged 2; and Fresh 7 vs. Aged 7); 2) different experimental tests (i.e., Fresh 2 vs. Fresh 7); and
3) two aging times (i.e., Aged 2 vs. Aged 7). Equivalence measures show that most of these
profiles are highly correlated ($r > 0.97$, mostly $> 0.99$) but statistically different ($P < 0.05$), with a
few exceptions.
Group comparisons between fresh and aged samples (Panel A of Table 2) show statistical
differences for all but Putnam (FL1) peat ($P > 0.94$). This is consistent with Watson et al (2019)
where atmospheric aging (7 days) reduced organic carbon EFs (i.e., $EF_{OC}$) by ~20 − 33 % for all
but Putnam (FL1) peats ($EF_{OC}$ remained within ±0.5 %). As OC is a major component of $PM_{2.5}$,
no apparent changes in OC and carbon fractions abundances may dictate the lack of statistical
differences between the fresh and aged profiles.
Paired comparisons for 2-day aging (Panel B of Table 2) show no statistical differences
between the Fresh 2 vs. Aged 2 Putnam (FL1) and Malaysian profiles ($P > 0.30$ and 0.95), which
may be due to the low number of samples (n=2) in the comparison; this results in no statistical
differences for combined Putnam (FL1) and Malaysian peat comparison ($P > 0.62$). Similar to the
findings of combining both fresh Florida profiles (i.e., all Fresh2 vs. all Fresh 7 in Table S4), the
two fresh Alaskan profiles (Fresh 2 vs. Fresh 7 in Panel D of Table 2) do not show statistical
differences ($P > 0.12$).
Compositing profiles by averaging each of the measured abundances may disguise some
useful information. For receptor model source apportionment, region-specific profiles are most
accurate for estimating source contributions.
Student $t$-tests for the gravimetric $PM_{2.5}$ mass concentrations ($\mu g\ m^{-3}$) measured upstream
and downstream of the OFR (Table S5) show statistically significant differences ($P < 0.05$)
between fresh vs. aged PM$_{2.5}$ (i.e., Fresh 2 vs Aged 2 and Fresh 7 vs Aged 7).  Fresh 2 and Fresh
7 PM$_{2.5}$ mass concentrations are similar, as expected from replicate tests for the same conditions.
Increases in some species abundances offset decreases on other abundances, resulting in similar
PM$_{2.5}$ levels for "all Fresh vs. all Aged" comparison.
**3.2    Sum of species to PM$_{2.5}$ mass ratios**
The sum of the major PM chemical abundances should be less than unity since oxygen,
hydrogen, and liquid water content are not measured (Chow et al., 1994; 1996).  As shown in Table
S6, the sums of elements, ions, and carbon explain averages of ~70–90 % of PM$_{2.5}$ mass for fresh
profiles except for Russian peat (62–64 %).  The "sum of species" decreased by an average of 6
% and 11 % after 2- and 7-days, respectively.  These differences are consistent with loss of semi-
volatile organic compounds (SVOCs) in the low temperature carbon fractions, although they are
offset by formation of oxygenated compounds during aging.  This is true for all but Putnam (FL1)
peat, for which the "sum of species" explains nearly the same fraction of PM$_{2.5}$ for the fresh and
aged profiles.
**3.3    Comparison between fresh and aged profiles**
Fresh and aged chemical abundances are compared in Fig. 2.  Species abundances vary by
several orders of magnitude but exhibit two distinguishable clusters: centered around 0.1 % for
reactive and secondary ionic species (e.g., $NH_4^+$, $NO_3^-$, and $SO_4^=$) and centered around 10 % for
carbon compounds (e.g., OC fractions and WSOC).  While most gaseous $NH_3$/PM$_{2.5}$ ratios exceed
10 %, $HNO_3$/PM$_{2.5}$ ratios are well below 1 %.  Reactive/ionic species and carbon components are
mostly above and below the 1:1 line, respectively, implying particle formation and evaporation
after atmospheric aging.  Large variabilities are found for individual species as noted by the
standard deviations associated with each average.
Figure 3 shows the ratio of averages between aged and fresh profiles with increasing ratios
from 2- to 7-day aging.  Atmospheric aging increased oxalic acid, $NO_3^-$, $NH_4^+$, and $SO_4^=$
abundances (likely due to conversion of nitrogen and sulfur gases [e.g., $NH_3$, NO, $NO_2$, and $SO_2$]
to particles), but decreased $NH_3$, levoglucosan, and low temperature OC1 and OC2 abundances in
most cases.  Large variations are found among measured species (left panels in Fig. 3) as ratios
range several orders of magnitude for mineral and ionic species.  Consistent with Fig. 2 where
most carbon compounds are close to but below the 1:1 line, the right panels in Fig. 3 show the
reduction of carbonaceous abundances with aged/fresh ratios between 0.1 and 1. Higher aged/fresh
ratios in low temperature OC1 and OC2 after 7-day aging are consistent with additional
volatilization with longer aging time.
Atmospheric aging should not change the abundances of mineral species (e.g., Al, Si, Ca,
Ti, and Fe), except to the extent that the $PM_{2.5}$ mass (to which all species are normalized) increases
or decreases with aging.  Large standard deviations associated with the ratio of averages for
mineral species in the left panels of Fig. 3 illustrate variabilities among different combustion tests
for the less abundant species.

### 3.4    Carbon abundances
#### 3.4.1    Organic carbon and thermally-evolved carbon fractions
Total carbon (TC, sum of OC and EC) constitutes the largest fraction of $PM_{2.5}$ (Table 1),
accounting for 59–87 % and 43–77 % of the $PM_{2.5}$ mass for the fresh and aged profiles,
respectively.  OC dominates TC with low EC abundances (0.67–4.4 %), as commonly found in
smoldering-dominated biomass combustion (Chakrabarty et al., 2006; Chen et al., 2007).  The
largest OC fractions are high temperature OC3 (15–30 % of $PM_{2.5}$), consistent with past studies
for biomass burning emissions (Chen et al., 2007; Chow et al., 2004).
OC abundances decreased with aging time.  As shown in Fig. S2, upstream (Fresh 2 and
Fresh 7) OC abundances ranged from 58–85 % and decreased by 4–12 % and 20–33 % after 2-
and 7-day aging, respectively.  The exception is for Putnam (FL1) peat, where the OC
abundances were similar (changed by ~0.5 to 1.5%) between fresh and aged profiles.  Part, but
not all of this reduction is due to increasing abundances of non-carbon components, particularly
nitrogen-containing species that add to $PM_{2.5}$ mass. OC abundance decreases after aging for
other profiles may have contributed to the statistical differences found between fresh and aged
$PM_{2.5}$ mass (Table S5). With the exception of Putnam (FL1) peat, the additional 7–22% OC
degradation from 2- to 7-day aging implies that much of the OC changes require about a week of
aging time.
The Student $t$-test for fresh and aged profiles shows statistical differences ($P <0.05$) for
TC, OC, and low temperature OC1 and OC2, but similarities for OC3 and OC4.  High
temperature OC3 and OC4 contain more polar and/or high molecular-weight organic
components (Chen et al., 2007) that are less likely to photochemically degrade. Large fractions
of pyrolyzed carbon (OP of 7–13 %) are also found, indicative of higher molecular-weight
compounds that are likely to char (Chow et al., 2001; Chow et al., 2004; Chow et al., 2018).
Reduction in OC abundances after atmospheric aging is attributed mostly to decreases in
low temperature OC1 and OC2 abundances in the OFR as shown in the fresh vs. aged ratios of
average abundances (Fig. 3). Figure S3a shows reductions in OC1 abundances after 2- and 7-
days of atmospheric aging is apparent but at a similar level: ranging from 2–10 % and 3–14 %,
respectively. Additional OC1 reductions from 2- to 7-days are most apparent for Russia and
Everglades (FL2) peats at the 6–10 % level. Similar reductions are found for OC2 (Fig. S3b):
ranging from 3–11 % and 3–12 % after the 2- and 7-days of aging, respectively. Prolonged aging
times resulted in additional 4–8 % OC2 reduction for all but Russian and Putnam (FL1) peats. As
oxidation of organic compounds with OH radicals is an efficient chemical aging process (Chim
et al., 2018), some of the VOCs and SVOCs may have been liberated (Smith et al., 2009).
**3.4.2    Organic mass (OM) and OM/OC ratios**
Reduction of the "sum of species" and OC abundances from fresh to aged profiles can be
offset by the formation of oxygenated organic compounds as the profiles age.    Different
assumptions have been used to transform OC to organic mass (OM) to account for unmeasured H,
O, N, and S in organic compounds (Cao, 2018; Chow et al., 2015a; Riggio et al., 2018).    As single
multipliers for OC cannot capture changes by oxidation in the OFR, OM is calculated by
subtracting mineral components (using the IMPROVE soil formula by Malm et al. (1994)), major
ions (i.e., $NH_4^+$, $NO_3^-$, and $SO_4^=$), and EC from $PM_{2.5}$ mass to account for unmeasured mass in
organic compounds (Chow et al., 2015a; Frank, 2006).    This approach assumes that no major
chemical species are unmeasured and that the remaining mass consists of H, O, N, and S associated
with OC in forming OM.
Table 3 shows that OM/OC ratios ranged from 1.1–1.7 and 1.3–2.2 for fresh and aged
profiles, respectively.    The lower OM/OC ratios in fresh emissions are consistent with those
reported for other types of biomass burning (Chen et al., 2007; Reid et al., 2005).    Figure S4 shows
a general upward trend in OM/OC ratios after atmospheric aging with additional 14–21 %
increases from 2- to 7-days for all but Putnam (FL1) peat. The increase in OM/OC ratios with
aging are likely due to an increase in oxygenated organics. The OM/OC ratio of 1.20 ± 0.05 for
fresh Borneo, Malaysian peat is consistent with the 1.26 ± 0.04 ratio for fresh peat burning
emissions in Central Kalimantan, Indonesia (Jayarathne et al., 2018), both located on the Island of
Borneo.
The highest OM/OC ratios are found for Russian peat, ranging 1.6–1.7 for fresh profiles
and increasing to 2.1–2.2 for aged profiles, consistent with formation of low vapor pressure
oxygenated compounds in the OFR. Watson et al. (2019) report that the Russian peat fuel contains
the lowest carbon (44.20 ± 1.01 %) and highest oxygen (38.64 ± 0.78 %) contents among the six
peats. The low carbon contents in peat fuel and source profiles are consistent with the lowest "sum
of species" found in Russian peat, with 62–64 % and 50–52 % of $PM_{2.5}$ mass for the fresh and
aged profiles, respectively. After 7-day aging for Siberian peat, the increasing OM/OC ratios from
1.2 ± 0.14 to 1.5 ± 0.18 are similar to the increase from 1.22 to 1.42 reported by Bhattarai et al.

(2018).

**3.4.3 Water-soluble organic carbon (WSOC)**
WSOC abundances in $PM_{2.5}$ were over two-fold higher in fresh Russian (36–37 %) than
Malaysian (15–17 %) peat. The 15–17 % WSOC in $PM_{2.5}$ for fresh Borneo, Malaysian peat
(Table 1) is consistent with the 16 ± 11 % from Central Kalimantan, Indonesia peat (Jayarathne
et al., 2018). However, the WSOC/$PM_{2.5}$ ratio is not a good indicator of changes in WSOC
abundances during atmospheric aging as $PM_{2.5}$ also contains non-water-soluble and non-
carbonaceous aerosol. Table S7 shows large variabilities associated with the differences (i.e.,
aged minus fresh), suggesting that no differences exist within ±3 standard deviations. The only
exceptions are for the 7-day Putnam (FL1) peat and 2-day Malaysian peat, where aging resulted
in 7–8 % increases of WSOC abundances in $PM_{2.5}$.
As WSOC is part of the OC, the WSOC/OC ratio is a better indicator of atmospheric
aging. WSOC/OC ratios (Table 3) vary between fresh (0.18–0.64) and aged (0.31–0.71) profiles.
Figure S5 shows a general increase of WSOC/OC ratios from fresh to aged profiles. Longer
aging time from 2- to 7-days results in 5–10 % higher WSOC/OC ratios for all but the two
Florida peats. OC water-solubility also varies by peat type. Russian peat OC emissions are
largely water-soluble, whereas Malaysian peat emissions are mostly water-insoluble, with
WSOC/OC ratios of 0.59–0.71 and 0.18–0.40, respectively.
**3.4.4 Carbohydrates**
Bates et al. (1991) found that peat from Sumatra, Indonesia consisted of 18–46 %
carbohydrate (mainly levoglucosan) relative to total carbon based on nuclear magnetic resonance
spectroscopy. Levoglucosan and its isomers (mannosan and galactosan) are saccharide derivatives
formed from incomplete combustion of cellulose and hemi-cellulose (Kuo et al., 2008;
Louchouarn et al., 2009) and have been used as markers for biomass burning in receptor model
source apportionment (Bates et al., 1991; Watson et al., 2016). These carbohydrate-derived
pyrolysis products undergo heterogeneous oxidation when exposed to OH radicals in the OFR
(Hennigan et al., 2010; Kessler et al., 2010).

Only five of the 17 carbohydrates (Table 1) were detected, with noticeable variations (e.g.,

>2 orders of magnitude) in levoglucosan for boreal and temperate peats. Levoglucosan abundances
account for 35–39 % and 20–25 % of $PM_{2.5}$ mass for fresh and aged Russian profiles, respectively.
On a carbon basis, Table 3 shows that levoglucosan-carbon (with an OM/OC ratio of 2.25)
accounts for 42–48 % and 30–35 % of WSOC and 27–28 % and 21–24 % of OC for fresh and
aged Russian profiles, respectively. These levels are less than the 96 ± 3.8 % levoglucosan or
~42.7 % of levoglucosan-carbon in OC reported for German and Indonesian peats (Iinuma et al.,
2007). Elevated levoglucosan is also found for Siberian and Alaskan peats, ranging from 4–18 %
in $PM_{2.5}$. However, the levoglucosan abundances are low (1–4 %) for the subtropical and tropical
peats. Aging time of 7 days resulted in an additional 1–4 % levoglucosan degradation relative to
2 days with the exception of an additional 9 % reduction for Russian peat.

The extent of levoglucosan degradation depends on organic aerosol composition, OH

exposure in the OFR, and vapor-wall losses (Bertrand et al., 2018a; 2018b; Pratap et al., 2019).
Figure 4 shows the presence of levoglucosan-carbon for the Russian and Alaskan peats after 2-
and 7-day aging, at the levels of 8–11 % and 2–9 %, respectively, in line with a chemical lifetime
longer than 2 days. This is consistent with the estimated 1.2–3.9 days of levoglucosan lifetimes
under different environments reported by Lai et al. (2014). However, other studies (Hennigan et
al., 2010; May et al., 2012; Pratap et al., 2019) found that levoglucosan experiences rapid gas-
phase oxidation, resulting in ~1–2 day lifetimes at ambient temperatures.

Among the carbohydrates, Jayarathne et al. (2018) reported 4.6 ± 4.0 % of levoglucosan in

OC for fresh Indonesia peat. Converting to levoglucosan-carbon in Jayarathne et al. (2018) yields
a fraction of 2 %, consistent with findings for Malaysian peat (1.4–2.4 %) in this study.

While the presence of levoglucosan in peat smoke is apparent, its isomer, galactosan was

not detectable. Mannosan is detectable in cold climate peats with 1–5 % of $PM_{2.5}$ for the Russian
and Alaskan peats and up to 1.3 % for Siberian peat. Apparent degradations from 3.9 to 2.5 % and
from 5.0 to 2.1 % in mannosan abundances are found for Russian peat (Table 1) after 2- and 7-
days, respectively. A 2- to 3-fold reduction in mannosan is also shown after 7 days aging for the
Siberian and Alaskan peats. Similar observations apply to glycerol in Russian peat, ranging 1.9–
3.5 % and 1.3–1.7 % of $PM_{2.5}$ for fresh and aged profiles, respectively. Other detectable
carbohydrates are galactose and mannitol, typically present at one hundredth of one percent of the
levoglucosan abundance.

### 3.4.5   Organic acids

Organic acids have been associated with many anthropogenic sources, including engine
exhaust, biomass burning, meat cooking, bioaerosol, and biogenic emissions. Past studies show
the presence of low molecular-weight dicarboxylic acids in biomass burning emissions (e.g.,
Falkovich et al., 2005; Veres et al., 2010).
Only four of the ten measured organic acids (Table 1) (i.e., formic acid, acetic acid, oxalic
acid, and propionic acid) were detectable with variable abundances (<0.02–3.9 %). The largest
changes between fresh and aged profiles are found for oxalic acid, ranging from <0.02–0.43 % of
$PM_{2.5}$ for fresh profiles, with ~10- to 20-fold increase after 2 days (0.6–1.3 %), and with one to
two orders of magnitude increases after 7 days (1.1–3.9 %). With the exception of Putnam (FL1)
peat (1.1 ± 0.19 %), oxalic acid accounts for >2.9 % of $PM_{2.5}$ mass after 7 days.
Acetic acid abundances are stable between fresh and aged profiles, mostly in the range of
0.2–0.5 % except for a 6-fold increase from 0.23 ± 0.15 % (Fresh 7) to 1.5 ± 2.0 % (Aged 7) for
Siberian peat with large variability among the tests. Formic acid and propionic acid abundances
are low (<0.5 and <0.02 %, respectively), but increase with aging. Extending the aging time from
2- to 7-days resulted in a notable increase in organic acid abundances, consistent with the increases
in WSOC/OC ratios (Table 3). By biome, the highest abundances for organic acids in $PM_{2.5}$ are
found for aged (Aged 7) Siberian peat, with 3.9 ± 1.4 % oxalic acid, 1.5 ± 2.0 % acetic acid, and
0.44 ± 0.28 % formic acid (Table 1).

### 3.5   Nitrogen species, sulfate, and chloride abundances

Ammonia normalized to $PM_{2.5}$ mass is high for fresh profiles, ranging 17–64 %, except for
the low $NH_3$ content in Russian peat (6–8 %). These abundances are reduced to 3–14 % and 1–7
% after 2- and 7-day aging, respectively. As shown in Fig. 5, most of the $NH_3$ rapidly diminished
after 2 days, with increasing particle-phase $NH_4^+$ and $NO_3^-$ after 7 days. The highest $NH_3$ to $PM_{2.5}$
ratios are found for fresh Everglades (FL2) peat profiles (51–64 %), ~2–8 fold higher than other
peats. These high and low $NH_3/PM_{2.5}$ ratios are consistent with the nitrogen contents in peat fuel:
3.93 ± 0.08 % for Everglades and 1.50 ± 0.52 % for Russian peats (Watson et al., 2019).
Ionic abundances are typically <0.5 %, especially in fresh profiles.  Abundances of $NH_4^+$
in $PM_{2.5}$ are low (0.0005–0.13 %) for fresh emissions, but increase to 0.05–1.0 % after 2 days and
3.4–6.7 % after 7 days, with the exception of Putnam (FL1) peat (1.01 ± 0.05 % $NH_4^+$).  Extending
the aging time from 2- to 7-days results in an additional increase of ~1–7 % $NH_4^+$ abundances, in
contrast to $NH_3$ that is largely depleted after 2 days.
Figure 5b shows increasing in $NO_3^-$ abundances with aging, 0.04–0.23 % for fresh profiles,
increasing to 0.74–2.64 % after 2 days, and to 2.0–8.2 % after 7 days with the exception of Putnam
(FL1) peat (1.10 ± 0.18 % $NO_3^-$).  After 7 days, $NH_4^+$ and $NO_3^-$ account for ~4–7 % and ~8 % of
$PM_{2.5}$ mass, respectively, for Siberian, Alaskan, and Everglades (FL2) peats.  No specific trend is
evident for $NO_2^-$, mostly <0.002 %, with ~0.2 % for some fresh Siberian and Alaskan peats.  The
ratio of gaseous $HNO_3$ to $PM_{2.5}$ is low, in the range of 0.2–0.5 % without much changes between
fresh and aged profiles. $HNO_3$ created through photochemistry is largely neutralized by the
abundant $NH_3$ in the emissions, resulting in the increasing $NH_4^+$ and $NO_3^-$ to $PM_{2.5}$ in aged profiles.
The reaction of $NH_3$ with $HNO_3$ to form ammonium nitrate ($NH_4NO_3$) is the main pathway
for inorganic aerosol formation, owing to low sulfur content in the peat fuels (Watson et al., 2019).
$SO_4^=$ abundances are low in fresh profiles (0.13–1.4 %), but they increase 2–3 fold after 2 days
aging except for the Alaskan (0.35–0.46 %) and Everglades (FL2) (1.3–1.4 %) profiles.  More
apparent changes are found for 7 days with the largest increase in $SO_4^=$ from 0.13 to 1.96 % for
the Malaysian peats –indicating formation of ammonium sulfate ([$NH_4$]$_2SO_4$).  The ion balance
shows more $NH_4^+$ than needed to completely neutralize $NO_3^-$ and $SO_4^=$ (Chow et al., 1994).  Some
$NH_4^+$ may be present as ammonium chloride ($NH_4Cl$), however, the abundance of chloride ($Cl^-$) is
low (<0.3 %).  The large increase in $NO_3^-$ and $SO_4^=$ after 7 days implies that a 2-day aging time is
not sufficient to allow the full formation of secondary $NH_4NO_3$ and ($NH_4$)$_2SO_4$.
**3.6    Mass reconstruction**
Mass reconstruction is applied to understand the changes in major chemical composition
between the fresh and aged profiles.  As shown in Fig. 6, the largest component of $PM_{2.5}$ is OM,
accounting for 94–99 % and 80–95 % of $PM_{2.5}$ mass for fresh and aged profiles, respectively.
Although the 7-day aging time increased the OM/OC ratios (by 12–19 %), the abundances of
OM in $PM_{2.5}$ are reduced (3–18 %). This can be attributed to the combined effects of increased
oxygenated organics; SVOC volatilization (Smith et al., 2009); and an increase in ionic species
as shown in the average aged/fresh ratios in Fig. 3. Figure 6 shows increases in ionic species
(i.e., sum of $NH_4^+$, $NO_3^-$, and $SO_4^=$), with low abundances (0.3–1.7 %) in fresh profiles, and
increasing 3–16 % after aging. The sum of ionic species accounts for 11–16 % of $PM_{2.5}$ mass for
the Siberian, Alaskan, Everglades (FL2), and Malaysian peats after 7 days, mainly due to the
increase in $NH_4^+$ and $NO_3^-$ as shown in Fig. 5.

Elemental abundances are low (<0.0001 %), mostly below the lower quantifiable limits.

Table 1 only lists 34 of the 51 elements (Na to U) detected by XRF. Using the IMPROVE soil
formula (assuming metal oxides of major mineral species (Malm et al., 1994) yielded 0.07–2.9 %
of mineral components. The IMPROVE soil formula has been applied in many other studies (e.g.,
Chan et al., 1997; Pant et al., 2015; Rogula-Kozlowska et al., 2012) which provides an adequate
estimate of geological mineral in reconstructed mass. Since geological minerals are not a major
component of $PM_{2.5}$, variations in the assumption regarding metal oxides or multipliers do not
contribute to large variations in reconstructed mass (Chow et al., 2015a).

This study indicates that an aging time of ~2 days represents the intermediate-aged source

profile, whereas 7 days represents the profile with adequate residence time to complete the
atmospheric process.

**3.7   Changes in source profiles by fuel moisture content**

The effect of fuel moisture content on source profiles is mostly unknown. The 25 % fuel

moisture content selected for this study intends to better simulate the conditions of moderate to
severe droughts where most peat fires occur. Increasing fuel moisture content from ~25 to 60 %
for the three Putnam (FL1) peat fuels yielded 12 % higher EFs for $CO_2$ ($EF_{CO_2}$), but 12–20 %
lower EFs for CO, NO, $NO_2$, and $PM_{2.5}$ mass (Watson et al., 2019). Tests of fuel-moisture content
on profile changes are available for only 2-day aging. Equivalence measures (Table S8) show
statistical differences ($P$ <0.001) between 25 % and 60 % moisture profiles for either fresh or aged
profiles with high correlations ($r$ >0.997) and over 93 % of species abundance fall within $\pm 3\sigma$.
While OC abundances in $PM_{2.5}$ are comparable for the fresh and aged profiles (70–72 %) for 25
% fuel moisture, a reduction of 18 % OC in $PM_{2.5}$ is found for 60 % fuel moisture (from 82 to 64
%) after aging (Table S2). The higher fuel moisture content also reduced WSOC by 6 % and
levoglucosan by 1.3 % with <1 % increases for $NH_4^+$ and organic acids. After aging, the $NH_3$ to
$PM_{2.5}$ ratios decreased from 28 % to 5 % and from 20 % to 8 % for the 25 % and 60 % fuel
moisture, respectively. These results are not conclusive as most measurements are associated with
high variabilities.

## 4    Summary and conclusion

Fresh and aged peat fire emission profiles from laboratory combustion chamber and potential aerosol mass-oxidation flow reactor (PAM-OFR) for six types of peats representing boreal (Odintsovo, Russia and Pskov, Siberia), temperate (Northern Alaska, USA), subtropical (Putnam County Lakebed and Everglades National Park, Florida, USA), and tropical (Borneo, Malaysia) biomes are compared.  Analyses are focused on the average of 24 paired profiles grouped by six peats and by fresh vs. aged profiles for 2- and 7-days of simulated atmospheric aging that represent intermediate-aged and well-aged source profiles, respectively.

Equivalence measures show that these profiles are highly correlated ($r$ >0.97, mostly >0.99) but statistically different ($P$ <0.05) between different biomes, suggesting that these profiles should be used independently for receptor model source apportionment studies in different climate regions.

The sum of chemical species (i.e., elements, ions, and carbon) explains an average of ~70–90 % of $PM_{2.5}$ mass for fresh profiles except for Russian peat (62–64 %), confirming that major $PM_{2.5}$ chemical species are measured.  Aging times of 2- and 7-days resulted in an average mass depletion of 6 % and 11 %, respectively.  These differences are caused by: 1) loss of SVOCs with aging, as indicated by lower abundances of OC1 and OC2 (evolved at 140 and 280 °C) in the aged profiles; and 2) replacement of the lost OC mass with unmeasured oxygen associated with secondary organic aerosol formation in the OFR.

Species abundances in $PM_{2.5}$ between aged and fresh profiles varied by several orders of magnitude but exhibited two distinguishable clusters, with reactive/ionic species (e.g., $NH_4^+$, $SO_4^=$, oxalic acid, and $HNO_3$) constituting 0.1–1 % and carbon compounds (e.g., OC, organic carbon fractions [OC1–OC4], and WSOC) constituting >1 % (mostly >10 %) of $PM_{2.5}$ mass.  Most $NH_3/PM_{2.5}$ ratios are >10 % whereas $HNO_3/PM_{2.5}$ ratios are <1 %.

Total carbon (TC, sum of OC and EC) is the largest component, accounting for 59–87 % and 43–77 % of the $PM_{2.5}$ mass for the fresh and aged profiles, respectively.  With predominant smoldering combustion, the majority of the TC is OC, with low EC abundances (0.67–4.4 %). Further degradation in OC abundances (7–22 %) from 2- to 7-day aging implies an incomplete transformation with short aging time.  Different thermal carbon fractions are used to characterize combustion and aging conditions.  While most of the OC thermally evolved at high temperatures (OC3 at 480 °C), losses of low temperature OC1 and OC2 are found, indicating a shift of gas-

particle partitioning of SVOC to gas-phase, where particle volatilization outweighed gas-to-
particle conversion.
Formation of oxygenated compounds is pronounced after aging, with organic mass (OM)
to OC ratios increasing by 14–21 % from 2- to 7-day aging.  The WSOC abundance in $PM_{2.5}$ varies
from 15–17 % and 37–37 % for fresh Malaysian and Russian peats, respectively.   While
levoglucosan accounts for ~1–4 % of $PM_{2.5}$ mass for fresh subtropical and tropical peats, elevated
levels (6–39 %) are found for boreal and temperate peats. Increasing the atmospheric aging time
from 2- to 7-days results in additional formation of organic acid and ionic species (e.g., oxalic
acid, $NO_3^-$, $NH_4^+$, and $SO_4^=$), but enhanced losses of $NH_3$, levoglucosan, and low temperature OC1
and OC2.
Among the four climate regions, Russian peat with the lowest carbon (44 %) and highest
oxygen (39 %) content, resulted in ~59–71 % of WSOC in OC along with the highest levoglucosan
(20–39 % of $PM_{2.5}$) and lowest $NH_3$/$PM_{2.5}$ ratios (3–8 %).  It also yielded the highest oxygenated
compounds after aging with OM/OC ratios of 2.1–2.2.  This contrasts with Malaysian peats that
are mostly water-insoluble (WSOC/OC of 0.18–0.40) with low oxygenated compounds after aging
(OM/OC ratios of 1.2–1.5).  Large increases are found for oxalic acid abundances from fresh
(<0.02–0.43 %) to 7-day aging (1–4%).
With the exception of Russian peats, fresh profiles contain high $NH_3$/$PM_{2.5}$ ratios (17–64
%) with low abundances after aging (3–14 % for 2 days and 1–7 % for 7 days).  Extending the
aging time from 2- to 7-days results in an increase to ~7–8 % $NH_4^+$ and $NO_3^-$ abundances.
Although the week-long aging time increased the OM/OC ratios, abundances of OM in $PM_{2.5}$ were
reduced by 3–18 %.
Source profiles can change with aging during transport from source to receptor.  This study
shows significant differences between fresh and aged peat combustion profiles among the four
biomes that can be used to establish speciated emission inventories for air quality modeling.  A
sufficient aging time (~one week) is needed to allow gas-to-particle partitioning of semi-
volatilized species, gas-phase oxidation, and volatilization to achieve representative source
profiles for receptor-oriented source apportionment.

## 5 Author contribution

JCC, JGW, JC, L-WAC, and XW jointly designed the study, performed the data analyses, and prepared the manuscript. ACW collected the peat fuels and provided technical advice. QW, JT, and SSHH carried out the peat combustion experiments. TBC and SDK assembled the database and performed the similarity and difference tests between the fresh and aged profiles.

## 6 Competing interests

The authors declare that there are no conflicts of interest.

## 7 Acknowledgements

This research was primarily supported by the National Science Foundation (NSF, AGS1464501) as well as internal funding from both the Desert Research Institute, Reno, NV, USA, and Institute of Earth Environment, Chinese Academy of Sciences, Xian, China.

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

Table 1. Average fresh and aged peat combustion source profiles (in % of PM$_{2.5}$ mass) for six types of peats

| | Average ± Standard Deviation of Percent PM$_{2.5}$ Mass[a] | | | | | | | |
| | Boreal | | | | | | | |
| | Odintsovo, Russia | | | | Pskov, Siberia | | | |
| Aging Time | 2 days | | 7 days | | 2 days | | 7 days | |
| | Fresh 2 | Aged 2 | Fresh 7 | Aged 7 | Fresh 2 | Aged 2[b] | Fresh 7 | Aged 7 |
| Peat IDs in the average[c] | PEAT030, PEAT031, PEAT032 | | PEAT033, PEAT034, PEAT035 | | PEAT023, PEAT025, PEAT026 | | PEAT027, PEAT028, PEAT029 | |
|---|---|---|---|---|---|---|---|---|
| Nitric Acid (HNO$_3$) | 0.18 ± 0.080 | 0.32 ± 0.15 | 0.21 ± 0.059 | 0.24 ± 0.085 | 0.18 ± 0.052 | 0.27 ± 0.074 | 0.27 ± 0.075 | 0.39 ± 0.15 |
| Ammonia (NH$_3$) | 6.0095 ± 0.93 | 3.21 ± 0.78 | 7.84 ± 0.31 | 4.56 ± 1.36 | 18.21 ± 3.97 | 8.81 ± 4.047 | 22.81 ± 5.88 | 7.090 ± 5.59 |
| | | | | | | | | |
| Water-Soluble Sodium (Na$^+$) | 0.018 ± 0.0015 | 0.024 ± 0.0013 | 0.022 ± 0.011 | 0.032 ± 0.0074 | 0.017 ± 0.0011 | 0.047 ± 0.020 | 0.0688 ± 0.038 | 0.058 ± 0.053 |
| Water-Soluble Potassium (K$^+$) | 0.034 ± 0.036 | na[d] | 0.11 ± 0.087 | na[d] | 0.020 ± 0.016 | na[d] | 0.0230 ± 0.014 | na[d] |
| Chloride (Cl$^-$) | 0.16 ± 0.022 | 0.12 ± 0.019 | 0.25 ± 0.053 | 0.092 ± 0.011 | 0.11 ± 0.031 | 0.11 ± 0.048 | 0.17 ± 0.014 | 0.086 ± 0.033 |
| Nitrite (NO$_2^-$) | 0.037 ± 0.063 | 0.00095 ± 0.0016 | 0.00 ± 0.00028 | 0.00086 ± 0.00077 | 0.0013 ± 0.0023 | 0.0023 ± 0.0036 | 0.20 ± 0.34 | 0.0056 ± 0.0033 |
| Nitrate (NO$_3^-$) | 0.23 ± 0.20 | 0.74 ± 0.080 | 0.13 ± 0.043 | 2.0029 ± 0.71 | 0.11 ± 0.11 | 1.79 ± 0.52 | 0.15 ± 0.076 | 8.23 ± 4.34 |
| Sulfate (SO$_4^=$) | 0.30 ± 0.33 | 0.67 ± 0.46 | 0.15 ± 0.044 | 0.84 ± 0.24 | 0.28 ± 0.15 | 0.68 ± 0.19 | 0.27 ± 0.025 | 1.15 ± 0.63 |
| Ammonium (NH$_4^+$) | 0.13 ± 0.14 | 1.045 ± 0.93 | 0.097 ± 0.057 | 3.38 ± 1.38 | 0.0014 ± 0.0012 | 0.21 ± 0.18 | 0.0463 ± 0.044 | 6.66 ± 3.67 |
| | | | | | | | | |
| OC1 (140°C) | 11.82 ± 2.58 | 6.42 ± 3.94 | 15.67 ± 3.60 | 4.096 ± 0.72 | 11.81 ± 2.20 | 5.014 ± 0.70 | 11.40 ± 0.65 | 4.34 ± 1.69 |
| OC2 (280°C) | 13.16 ± 1.42 | 9.84 ± 1.094 | 12.029 ± 1.049 | 9.032 ± 1.27 | 20.59 ± 1.87 | 15.45 ± 2.65 | 21.21 ± 2.38 | 10.54 ± 0.18 |
| OC3 (480°C) | 17.69 ± 3.013 | 14.60 ± 1.93 | 17.33 ± 2.39 | 13.99 ± 2.46 | 25.93 ± 3.62 | 26.78 ± 8.46 | 29.63 ± 5.62 | 19.74 ± 0.79 |
| OC4 (580°C) | 6.69 ± 0.49 | 5.83 ± 0.51 | 6.090 ± 1.61 | 4.40 ± 0.84 | 5.79 ± 0.21 | 8.85 ± 1.27 | 8.72 ± 3.83 | 6.31 ± 2.35 |
| Pyrolized Carbon (OP) | 8.26 ± 2.086 | 8.61 ± 4.35 | 9.29 ± 1.0016 | 9.39 ± 1.30 | 9.52 ± 2.15 | 12.12 ± 4.27 | 10.34 ± 1.82 | 12.76 ± 1.58 |
| Organic Carbon (OC)[g] | 57.61 ± 5.21 | 45.29 ± 9.90 | 60.42 ± 5.37 | 40.90 ± 4.87 | 73.65 ± 6.82 | 68.21 ± 13.33 | 81.30 ± 9.29 | 53.69 ± 5.32 |
| | | | | | | | | |
| EC1 (580°C) | 6.47 ± 1.64 | 6.77 ± 2.33 | 6.51 ± 0.53 | 9.31 ± 1.50 | 7.84 ± 2.19 | 9.23 ± 0.82 | 5.31 ± 0.57 | 7.79 ± 1.28 |
| EC2 (740°C) | 3.60 ± 2.32 | 3.36 ± 2.52 | 4.61 ± 0.034 | 2.051 ± 0.50 | 4.92 ± 3.76 | 5.98 ± 4.73 | 5.87 ± 0.74 | 7.038 ± 2.48 |
| EC3 (840°C) | 0.00 ± 0.00020 | 0.00 ± 0.00022 | 0.00 ± 0.00020 | 0.00 ± 0.00021 | 0.00 ± 0.00021 | 0.00 ± 0.00028 | 0.00 ± 0.00029 | 0.00 ± 0.00032 |
| Elemental Carbon (EC)[g] | 1.82 ± 1.26 | 1.52 ± 0.36 | 1.83 ± 0.69 | 1.98 ± 0.75 | 3.23 ± 0.80 | 3.090 ± 0.83 | 0.83 ± 1.30 | 2.076 ± 0.36 |
| | | | | | | | | |
| Total Carbon (TC) | 59.43 ± 4.49 | 46.81 ± 10.23 | 62.25 ± 4.95 | 42.88 ± 4.76 | 76.88 ± 6.37 | 71.30 ± 13.96 | 82.14 ± 10.57 | 55.77 ± 5.58 |
| | | | | | | | | |
| Water-Soluble OC (WSOC) | 36.97 ± 2.71 | 31.80 ± 3.15 | 35.77 ± 2.30 | 29.21 ± 6.31 | 23.84 ± 1.84 | 29.88 ± 7.10 | 32.50 ± 0.71[e] | 29.88 ± 8.88 |
| Formic acid (CH$_2$O$_2$) | 0.17 ± 0.074 | 0.23 ± 0.054 | 0.23 ± 0.090 | 0.32 ± 0.18 | 0.045 ± 0.016 | 0.18 ± 0.054 | 0.067 ± 0.0097 | 0.44 ± 0.28 |
| Acetic acid (C$_2$H$_4$O$_2$) | 0.61 ± 0.38 | 0.63 ± 0.37 | 0.67 ± 0.15 | 0.88 ± 0.47 | 0.20 ± 0.16 | 0.34 ± 0.15 | 0.23 ± 0.15 | 1.46 ± 2.03 |
| Oxalic acid (C$_2$H$_2$O$_4$) | 0.10 ± 0.063 | 0.97 ± 0.20 | 0.28 ± 0.22 | 2.88 ± 0.77 | 0.062 ± 0.013 | 1.31 ± 0.47 | 0.076 ± 0.019 | 3.90 ± 1.43 |
| Propionic acid (C$_3$H$_5$O$_2$) | 0.036 ± 0.032 | 0.12 ± 0.15 | 0.066 ± 0.032 | 0.020 ± 0.031 | 0.00 ± 0.00015 | 0.026 ± 0.045 | 0.032 ± 0.032 | 0.00 ± 0.00023 |
| | | | | | | | | |
| Levoglucosan (C$_6$H$_{10}$O$_5$) | 35.35 ± 7.90 | 24.95 ± 8.97 | 38.66 ± 2.089 | 19.63 ± 4.044 | 6.66 ± 2.58 | 4.21 ± 0.59 | 9.39 ± 2.077 | 3.80 ± 0.35 |
| Mannosan (C$_6$H$_{10}$O$_5$) | 3.93 ± 1.18 | 2.52 ± 1.068 | 5.039 ± 0.58 | 2.14 ± 0.85 | 0.053 ± 0.092 | 0.00 ± 0.00044 | 1.28 ± 0.54 | 0.46 ± 0.16 |
| Galactose/Maltitol (C$_6$H$_{12}$O$_6$/C$_{12}$H$_{24}$O$_{11}$) | 0.00 ± 0.00016 | 0.00 ± 0.00017 | 0.063 ± 0.11 | 0.00 ± 0.00016 | 0.0058 ± 0.010 | 0.00 ± 0.00023 | 0.00 ± 0.00023 | 0.082 ± 0.14 |
| Glycerol (C$_3$H$_8$O$_3$) | 1.90 ± 0.19 | 1.73 ± 0.42 | 3.54 ± 2.14 | 1.25 ± 0.17 | 0.00 ± 0.0000029 | 0.00 ± 0.0000040 | 0.43 ± 0.43 | 0.00 ± 0.0000046 |
| Mannitol (C$_6$H$_{14}$O$_6$) | 0.00 ± 0.000056 | 0.00 ± 0.000061 | 0.062 ± 0.11 | 0.00 ± 0.000058 | 0.00 ± 0.000058 | 0.00 ± 0.000081 | 0.00 ± 0.0000836 | 0.17 ± 0.30 |
| | | | | | | | | |
| Aluminum (Al) | 0.073 ± 0.66 | 0.15 ± 0.87 | 0.22 ± 0.73 | 0.29 ± 2.74 | 0.086 ± 1.49 | 0.00 ± 0.0074 | 0.075 ± 0.83 | 0.20 ± 0.17 |
| Silicon (Si) | 0.0069 ± 0.12 | 0.12 ± 0.44 | 0.013 ± 0.12 | 0.68 ± 0.24 | 0.022 ± 0.19 | 0.22 ± 0.00089 | 0.0050 ± 0.044 | 0.47 ± 0.79 |
| Phosphorous (P) | 0.00 ± 0.000084 | 0.00018 ± 0.00025 | 0.00079 ± 0.0014 | 0.00 ± 0.000095 | 0.00 ± 0.000090 | 0.00 ± 0.00017 | 0.00 ± 0.00012 | 0.00 ± 0.000093 |


    Table 1 (cont'd)

| | Average ± Standard Deviation of Percent PM$_{2.5}$ Mass[a] | | | | | | | |
|---|---|---|---|---|---|---|---|---|
| | Boreal | | | | | | | |
| | Odintsovo, Russia | | | | Pskov, Siberia | | | |
| Aging Time | 2 days | | 7 days | | 2 days | | 7 days | |
| | Fresh 2 | Aged 2 | Fresh 7 | Aged 7 | Fresh 2 | Aged 2[b] | Fresh 7 | Aged 7 |
| Peat IDs in the average[c] | PEAT030, PEAT031, PEAT032 | | PEAT033, PEAT034, PEAT035 | | PEAT023, PEAT025, PEAT026 | | PEAT027, PEAT028, PEAT029 | |
| Sulfur (S) | 0.024 ± 0.0088 | 0.081 ± 0.046 | 0.040 ± 0.056 | 0.26 ± 0.095 | 0.081 ± 0.030 | 0.090 ± 0.000098 | 0.028 ± 0.034 | 0.31 ± 0.0057 |
| Chlorine (Cl) | 0.12 ± 0.027 | 0.035 ± 0.019 | 0.18 ± 0.030 | 0.032 ± 0.0025 | 0.11 ± 0.015 | 0.057 ± 0.000068 | 0.081 ± 0.018 | 0.027 ± 0.0064 |
| | | | | | | | | |
| Potassium (K) | 0.030 ± 0.011 | 0.48 ± 0.44 | 0.041 ± 0.018 | 0.13 ± 0.035 | 0.15 ± 0.19 | 0.096 ± 0.00025 | 0.11 ± 0.14 | 0.30 ± 0.017 |
| Calcium (Ca) | 0.018 ± 0.016 | 0.040 ± 0.056 | 0.031 ± 0.025 | 0.0034 ± 0.0048 | 0.00 ± 0.00 | 0.00 ± 0.00092 | 0.00 ± 0.00065 | 0.028 ± 0.039 |
| Scandium (Sc) | 0.064 ± 0.11 | 0.00 ± 0.0021 | 0.00 ± 0.0021 | 0.00 ± 0.0023 | 0.079 ± 0.14 | 0.00 ± 0.0041 | 0.031 ± 0.053 | 0.00 ± 0.0022 |
| Titanium (Ti) | 0.0046 ± 0.0056 | 0.00 ± 0.000076 | 0.0055 ± 0.0049 | 0.0013 ± 0.0018 | 0.0079 ± 0.014 | 0.00 ± 0.00015 | 0.00 ± 0.00010 | 0.00 ± 0.000078 |
| Vanadium (V) | 0.00 ± 0.000013 | 0.00 ± 0.000014 | 0.00 ± 0.000014 | 0.00 ± 0.000015 | 0.00070 ± 0.0012 | 0.00 ± 0.000027 | 0.00 ± 0.000019 | 0.00 ± 0.000015 |
| | | | | | | | | |
| Chromium (Cr) | 0.0012 ± 0.0020 | 0.00039 ± 0.00056 | 0.00 ± 0.000046 | 0.00084 ± 0.0012 | 0.00079 ± 0.0014 | 0.00 ± 0.000091 | 0.0010 ± 0.00095 | 0.00 ± 0.000049 |
| Manganese (Mn) | 0.0014 ± 0.0022 | 0.00053 ± 0.00074 | 0.0037 ± 0.0033 | 0.00 ± 0.00018 | 0.0018 ± 0.0022 | 0.020 ± 0.00032 | 0.0031 ± 0.0031 | 0.0051 ± 0.0072 |
| Iron (Fe) | 0.038 ± 0.021 | 0.091 ± 0.098 | 0.062 ± 0.043 | 0.26 ± 0.32 | 0.039 ± 0.035 | 0.029 ± 0.00056 | 0.013 ± 0.017 | 0.015 ± 0.021 |
| Cobalt (Co) | 0.000032 ± 0.000056 | 0.00 ± 0.0000094 | 0.000037 ± 0.000064 | 0.00049 ± 0.00069 | 0.00018 ± 0.00031 | 0.00 ± 0.000018 | 0.00 ± 0.000013 | 0.00 ± 0.000018 |
| Nickel (Ni) | 0.00 ± 0.000022 | 0.0026 ± 0.0037 | 0.000029 ± 0.000050 | 0.00 ± 0.000025 | 0.00 ± 0.000024 | 0.00086 ± 0.000045 | 0.0014 ± 0.0017 | 0.00041 ± 0.00039 |
| | | | | | | | | |
| Copper (Cu) | 0.0055 ± 0.0029 | 0.15 ± 0.11 | 0.0052 ± 0.0038 | 0.046 ± 0.054 | 0.0072 ± 0.0041 | 0.014 ± 0.00028 | 0.047 ± 0.052 | 0.11 ± 0.067 |
| Zinc (Zn) | 0.0017 ± 0.0015 | 0.054 ± 0.066 | 0.0047 ± 0.0041 | 0.053 ± 0.070 | 0.0053 ± 0.0030 | 0.0034 ± 0.00016 | 0.0058 ± 0.0056 | 0.0019 ± 0.00081 |
| Arsenic (As) | 0.00086 ± 0.0015 | 0.00 ± 0.000038 | 0.00 ± 0.000037 | 0.00 ± 0.000040 | 0.00076 ± 0.0013 | 0.0050 ± 0.000073 | 0.000069 ± 0.00012 | 0.00013 ± 0.00019 |
| Selenium (Se) | 0.00021 ± 0.00036 | 0.0026 ± 0.0037 | 0.00067 ± 0.00076 | 0.00029 ± 0.00041 | 0.0018 ± 0.0022 | 0.0026 ± 0.00013 | 0.00035 ± 0.00031 | 0.00029 ± 0.00041 |
| Bromine (Br) | 0.00041 ± 0.00036 | 0.0030 ± 0.0031 | 0.00096 ± 0.0014 | 0.0021 ± 0.0019 | 0.0072 ± 0.0043 | 0.0032 ± 0.000036 | 0.0092 ± 0.0066 | 0.0066 ± 0.0014 |
| | | | | | | | | |
| Rubidium (Rb) | 0.00052 ± 0.00090 | 0.0029 ± 0.000079 | 0.0020 ± 0.0019 | 0.00049 ± 0.00069 | 0.00031 ± 0.00054 | 0.00 ± 0.000045 | 0.00066 ± 0.00068 | 0.0024 ± 0.0034 |
| Strontium (Sr) | 0.0033 ± 0.0032 | 0.0017 ± 0.0018 | 0.0032 ± 0.0027 | 0.0033 ± 0.0013 | 0.0027 ± 0.0028 | 0.0039 ± 0.000045 | 0.0072 ± 0.0042 | 0.0047 ± 0.0066 |
| Yttrium (Y) | 0.00079 ± 0.0013 | 0.000066 ± 0.000093 | 0.0031 ± 0.0035 | 0.00077 ± 0.0011 | 0.0014 ± 0.0012 | 0.0015 ± 0.000045 | 0.0045 ± 0.0045 | 0.0053 ± 0.0049 |
| Zirconium (Zr) | 0.0040 ± 0.0024 | 0.0034 ± 0.0014 | 0.0013 ± 0.0018 | 0.0017 ± 0.0024 | 0.0051 ± 0.0019 | 0.00 ± 0.00017 | 0.0060 ± 0.0088 | 0.0033 ± 0.0021 |
| Niobium (Nb) | 0.00072 ± 0.0012 | 0.0023 ± 0.0013 | 0.00036 ± 0.00038 | 0.00063 ± 0.00089 | 0.00040 ± 0.00069 | 0.00064 ± 0.000082 | 0.00039 ± 0.00067 | 0.00044 ± 0.00062 |
| | | | | | | | | |
| Molybdenum (Mo) | 0.0020 ± 0.0035 | 0.00 ± 0.000090 | 0.0015 ± 0.0011 | 0.0030 ± 0.0010 | 0.0029 ± 0.0051 | 0.00 ± 0.00017 | 0.0013 ± 0.0022 | 0.0026 ± 0.0037 |
| Silver (Ag) | 0.0010 ± 0.0015 | 0.00 ± 0.00011 | 0.00 ± 0.00011 | 0.00 ± 0.00012 | 0.00 ± 0.00011 | 0.00 ± 0.00022 | 0.0083 ± 0.0074 | 0.00 ± 0.00012 |
| Cadmium (Cd) | 0.0034 ± 0.0059 | 0.0038 ± 0.0053 | 0.0023 ± 0.0039 | 0.0023 ± 0.0033 | 0.00 ± 0.00016 | 0.00 ± 0.00030 | 0.0024 ± 0.0029 | 0.00 ± 0.00016 |
| Indium (In) | 0.00 ± 0.00010 | 0.00 ± 0.00011 | 0.0059 ± 0.0011 | 0.0060 ± 0.0016 | 0.00065 ± 0.0011 | 0.018 ± 0.00021 | 0.0027 ± 0.0047 | 0.00 ± 0.00011 |
| Tin (Sn) | 0.0028 ± 0.0048 | 0.0095 ± 0.013 | 0.0013 ± 0.0022 | 0.0037 ± 0.0053 | 0.0098 ± 0.010 | 0.0075 ± 0.00038 | 0.0092 ± 0.014 | 0.0089 ± 0.013 |
| | | | | | | | | |
| Antimony (Sb) | 0.00 ± 0.00028 | 0.0086 ± 0.012 | 0.00 ± 0.00029 | 0.00 ± 0.00032 | 0.00 ± 0.00030 | 0.000053 ± 0.00058 | 0.00 ± 0.00041 | 0.00 ± 0.00031 |
| Cesium (Cs) | 0.025 ± 0.040 | 0.0085 ± 0.012 | 0.023 ± 0.033 | 0.014 ± 0.020 | 0.0057 ± 0.0099 | 0.00 ± 0.0016 | 0.0046 ± 0.0079 | 0.00 ± 0.00086 |
| Barium (Ba) | 0.014 ± 0.024 | 0.00 ± 0.00071 | 0.011 ± 0.020 | 0.00 ± 0.00068 | 0.023 ± 0.020 | 0.00 ± 0.0012 | 0.00 ± 0.00086 | 0.00 ± 0.0067 |
| Lanthanum (La) | 0.048 ± 0.043 | 0.00 ± 0.0012 | 0.049 ± 0.043 | 0.059 ± 0.083 | 0.017 ± 0.030 | 0.00 ± 0.0024 | 0.094 ± 0.085 | 0.020 ± 0.028 |
| Wolfram (W) | 0.0023 ± 0.0014 | 0.0073 ± 0.010 | 0.0077 ± 0.013 | 0.011 ± 0.0016 | 0.00079 ± 0.0014 | 0.00 ± 0.00047 | 0.0047 ± 0.0082 | 0.0048 ± 0.00054 |
| | | | | | | | | |
| Gold (Au) | 0.0029 ± 0.0027 | 0.00 ± 0.000071 | 0.00080 ± 0.0014 | 0.0024 ± 0.0033 | 0.00 ± 0.000071 | 0.012 ± 0.00014 | 0.0038 ± 0.0065 | 0.0018 ± 0.0025 |
| Mercury (Hg) | 0.0015 ± 0.0014 | 0.00 ± 0.000038 | 0.00081 ± 0.0014 | 0.00 ± 0.000040 | 0.0013 ± 0.0023 | 0.00 ± 0.000073 | 0.000065 ± 0.00011 | 0.00 ± 0.000039 |
| Lead (Pb) | 0.0026 ± 0.0024 | 0.0018 ± 0.0025 | 0.0024 ± 0.0028 | 0.0053 ± 0.0074 | 0.00 ± 0.000071 | 0.00 ± 0.00014 | 0.0050 ± 0.00088 | 0.0027 ± 0.0032 |
| Uranium (U) | 0.0018 ± 0.0031 | 0.0017 ± 0.0024 | 0.00096 ± 0.0017 | 0.0024 ± 0.0035 | 0.0028 ± 0.0027 | 0.00 ± 0.00025 | 0.0025 ± 0.0033 | 0.0046 ± 0.0066 |

Table 1 (cont'd)

| | Average ± Standard Deviation of Percent PM$_{2.5}$ Mass | | | | | | | |
|---|---|---|---|---|---|---|---|---|
| | Temperate | | | | Subtropical | | | |
| | Northern Alaska, USA | | | | Putnam County Lakebed, Florida (FL1) | | | |
| Aging Time | 2 days | | 7 days | | 2 (25%) days | | 7 (25%) days | |
| | Fresh 2 | Aged 2 | Fresh 7 | Aged 7[b] | Fresh 2 | Aged 2 | Fresh 7 | Aged 7 |
| Peat IDs in the average[c] | PEAT013, PEAT014, PEAT019 | | PEAT020, PEAT022 | | PEAT008, PEAT009 | | PEAT005, PEAT006 | |
| Nitric Acid (HNO$_3$) | 0.40 ± 0.19 | 0.31 ± 0.15 | 0.29 ± 0.22 | 0.28 ± 0.10 | 0.18 ± 0.033 | 0.39 ± 0.17 | 0.32 ± 0.25 | 0.23 ± 0.0055 |
| Ammonia (NH$_3$) | 16.64 ± 8.41 | 6.39 ± 3.76 | 27.73 ± 11.16 | 5.13 ± 0.80 | 28.03 ± 2.90 | 4.76 ± 0.52 | na[f] | 1.39 ± 0.62 |
| Water-Soluble Sodium (Na$^+$) | 0.047 ± 0.035 | 0.13 ± 0.15 | 0.047 ± 0.036 | 0.053 ± 0.022 | 0.015 ± 0.00033 | 0.033 ± 0.00033 | 0.030 ± 0.0058 | 0.032 ± 0.0048 |
| Water-Soluble Potassium (K$^+$) | 0.042 ± 0.068 | na[d] | 0.035 ± 0.010 | na[d] | 0.010 ± 0.015 | na[d] | 0.029 ± 0.0042 | na[d] |
| Chloride (Cl$^-$) | 0.21 ± 0.050 | 0.25 ± 0.19 | 0.29 ± 0.029 | 0.11 ± 0.0042 | 0.14 ± 0.035 | 0.18 ± 0.10 | 0.14 ± 0.041 | 0.087 ± 0.0049 |
| Nitrite (NO$_2^-$) | 0.15 ± 0.25 | 0.0015 ± 0.0019 | 0.00 ± 0.00040 | 0.0014 ± 0.00094 | 0.053 ± 0.071 | 0.011 ± 0.015 | 0.00044 ± 0.00062 | 0.0012 ± 0.00037 |
| Nitrate (NO$_3^-$) | 0.20 ± 0.16 | 1.45 ± 0.79 | 0.17 ± 0.053 | 8.19 ± 5.96 | 0.16 ± 0.12 | 0.87 ± 0.15 | 0.040 ± 0.000070 | 1.10 ± 0.18 |
| Sulfate (SO$_4^=$) | 0.46 ± 0.38 | 0.35 ± 0.16 | 0.26 ± 0.24 | 0.64 ± 0.23 | 0.89 ± 0.97 | 1.60 ± 1.33 | 0.22 ± 0.013 | 1.29 ± 0.13 |
| Ammonium (NH$_4^+$) | 0.11 ± 0.19 | 0.66 ± 0.78 | 0.0028 ± 0.00085 | 4.30 ± 0.098 | 0.00070 ± 0.00099 | 0.052 ± 0.074 | 0.00046 ± 0.000031 | 1.0080 ± 0.048 |
| OC1 (140°C) | 14.58 ± 4.92 | 10.33 ± 4.49 | 9.28 ± 4.049 | 3.76 ± 1.77 | 9.54 ± 2.50 | 7.48 ± 3.12 | 13.15 ± 3.56 | 10.087 ± 1.63 |
| OC2 (280°C) | 21.37 ± 0.70 | 17.98 ± 1.13 | 17.28 ± 3.42 | 9.68 ± 3.57 | 21.66 ± 2.045 | 19.50 ± 0.85 | 20.74 ± 2.34 | 19.76 ± 2.57 |
| OC3 (480°C) | 26.36 ± 5.88 | 24.57 ± 6.14 | 28.99 ± 14.35 | 18.47 ± 5.013 | 25.30 ± 7.61 | 24.97 ± 0.95 | 20.38 ± 0.63 | 21.97 ± 1.65 |
| OC4 (580°C) | 7.70 ± 1.79 | 6.51 ± 1.99 | 8.0014 ± 4.44 | 8.56 ± 2.51 | 7.60 ± 4.045 | 7.76 ± 1.017 | 4.29 ± 0.0044 | 5.34 ± 2.10 |
| Pyrolized Carbon (OP) | 7.40 ± 1.69 | 10.66 ± 4.45 | 7.35 ± 2.14 | 6.68 ± 3.39 | 7.61 ± 1.80 | 10.45 ± 1.14 | 8.81 ± 0.79 | 10.73 ± 0.53 |
| Organic Carbon (OC)[g] | 77.41 ± 6.13 | 70.047 ± 8.98 | 70.91 ± 20.30 | 47.16 ± 11.23 | 71.71 ± 9.40 | 70.16 ± 5.033 | 67.37 ± 4.48 | 67.88 ± 5.22 |
| EC1 (580°C) | 6.050 ± 1.50 | 9.94 ± 2.92 | 5.24 ± 1.038 | 7.11 ± 3.90 | 7.61 ± 2.43 | 9.58 ± 1.36 | 6.44 ± 0.099 | 8.98 ± 1.36 |
| EC2 (740°C) | 3.43 ± 3.013 | 2.93 ± 2.14 | 5.70 ± 1.85 | 1.63 ± 1.99 | 3.51 ± 2.51 | 2.94 ± 2.34 | 4.057 ± 0.60 | 3.28 ± 0.88 |
| EC3 (840°C) | 0.00 ± 0.00020 | 0.00 ± 0.00021 | 0.00 ± 0.00029 | 0.00 ± 0.00022 | 0.00 ± 0.00014 | 0.00 ± 0.00015 | 0.00 ± 0.00011 | 0.00 ± 0.00010 |
| Elemental Carbon (EC)[g] | 2.082 ± 1.079 | 2.21 ± 0.99 | 3.59 ± 0.75 | 2.047 ± 2.51 | 3.51 ± 1.72 | 2.076 ± 0.16 | 1.69 ± 0.29 | 1.53 ± 0.057 |
| Total Carbon (TC) | 79.49 ± 7.072 | 72.26 ± 8.88 | 74.50 ± 21.052 | 49.20 ± 13.74 | 75.23 ± 11.12 | 72.24 ± 4.88 | 69.06 ± 4.77 | 69.41 ± 5.16 |
| Water-Soluble OC (WSOC) | 29.32 ± 9.03 | 28.35 ± 3.81 | 31.58 ± 11.22 | 25.77 ± 4.05 | 19.53 ± 4.67 | 22.71 ± 4.43 | 16.33 ± 1.17 | 23.15 ± 1.45 |
| Formic acid (CH$_2$O$_2$) | 0.093 ± 0.029 | 0.21 ± 0.049 | 0.069 ± 0.018 | 0.25 ± 0.11 | 0.11 ± 0.097 | 0.20 ± 0.13 | 0.022 ± 0.0044 | 0.15 ± 0.0065 |
| Acetic acid (C$_2$H$_4$O$_2$) | 0.38 ± 0.15 | 0.64 ± 0.17 | 0.45 ± 0.24 | 0.34 ± 0.26 | 0.19 ± 0.15 | 0.047 ± 0.011 | 0.056 ± 0.010 | 0.26 ± 0.024 |
| Oxalic acid (C$_2$H$_2$O$_4$) | 0.039 ± 0.028 | 0.86 ± 0.16 | 0.043 ± 0.061 | 3.26 ± 0.52 | 0.050 ± 0.070 | 0.58 ± 0.26 | 0.00 ± 0.02 | 1.12 ± 0.19 |
| Propionic acid (C$_3$H$_5$O$_2$) | 0.0072 ± 0.010 | 0.024 ± 0.034 | 0.00 ± 0.00020 | 0.034 ± 0.048 | 0.00 ± 0.000099 | 0.00 ± 0.00010 | 0.00 ± 0.000077 | 0.00 ± 0.000071 |
| Levoglucosan (C$_6$H$_{10}$O$_5$) | 17.87 ± 8.03 | 16.99 ± 3.32 | 9.78 ± 1.15 | 4.87 ± 2.89 | 3.15 ± 0.0092 | 2.78 ± 0.041 | 3.12 ± 0.24 | 1.49 ± 0.50 |
| Mannosan (C$_6$H$_{10}$O$_5$) | 3.46 ± 1.25 | 3.53 ± 1.26 | 2.73 ± 0.40 | 0.95 ± 0.34 | 0.00 ± 0.00022 | 0.00 ± 0.00023 | 0.00 ± 0.00017 | 0.00 ± 0.00016 |
| Galactose/Maltitol (C$_6$H$_{12}$O$_6$/C$_{12}$H$_{24}$O$_{11}$) | 0.00 ± 0.00015 | 0.00 ± 0.00016 | 0.00 ± 0.00022 | 0.00 ± 0.00017 | 0.00 ± 0.00011 | 0.00 ± 0.00012 | 0.00 ± 0.00087 | 0.00 ± 0.000079 |
| Glycerol (C$_3$H$_8$O$_3$) | 0.23 ± 0.33 | 0.20 ± 0.28 | 0.98 ± 1.39 | 0.12 ± 0.17 | 0.00 ± 0.0000050 | 0.00 ± 0.0000021 | 0.00 ± 0.0000015 | 0.00 ± 0.0000014 |
| Mannitol (C$_6$H$_{14}$O$_6$) | 0.00 ± 0.000055 | 0.10 ± 0.15 | 0.00 ± 0.000080 | 0.00 ± 0.000061 | 0.00 ± 0.000039 | 0.00 ± 0.000042 | 0.00 ± 0.000056 | 0.00 ± 0.000028 |
| Aluminum (Al) | 0.026 ± 0.24 | 0.063 ± 0.28 | 0.029 ± 0.13 | 0.0098 ± 0.0046 | 0.026 ± 0.059 | 0.069 ± 0.97 | 0.12 ± 1.34 | 0.080 ± 0.61 |
| Silicon (Si) | 0.0077 ± 0.12 | 0.0069 ± 0.098 | 0.0012 ± 0.017 | 0.63 ± 0.00060 | 0.00 ± 0.00030 | 0.021 ± 0.22 | 0.00 ± 0.0021 | 0.021 ± 0.067 |
| Phosphorous (P) | 0.00 ± 0.000084 | 0.00 ± 0.00011 | 0.00 ± 0.00012 | 0.00 ± 0.00011 | 0.00 ± 0.000060 | 0.00 ± 0.000064 | 0.00 ± 0.000048 | 0.00 ± 0.000044 |

Table 1 (cont'd)

| | Average ± Standard Deviation of Percent PM$_{2.5}$ Mass | | | | | | | |
| --- | --- | --- | --- | --- | --- | --- | --- | --- |
| | Temperate | | | | Subtropical | | | |
| | Northern Alaska, USA | | | | Putnam County Lakebed, Florida (FL1) | | | |
| Aging Time | 2 days | | 7 days | | 2 (25%) days | | 7 (25%) days | |
| | Fresh 2 | Aged 2 | Fresh 7 | Aged 7[b] | Fresh 2 | Aged 2 | Fresh 7 | Aged 7 |
| Peat IDs in the average[c] | PEAT013, PEAT014, PEAT019 | | PEAT020, PEAT022 | | PEAT008, PEAT009 | | PEAT005, PEAT006 | |
| Sulfur (S) | 0.031 ± 0.054 | 0.062 ± 0.087 | 0.0099 ± 0.014 | 0.34 ± 0.00013 | 0.19 ± 0.056 | 0.37 ± 0.24 | 0.17 ± 0.037 | 0.74 ± 0.047 |
| Chlorine (Cl) | 0.12 ± 0.068 | 0.087 ± 0.030 | 0.14 ± 0.049 | 0.019 ± 0.000040 | 0.12 ± 0.0064 | 0.067 ± 0.024 | 0.14 ± 0.022 | 0.056 ± 0.00047 |
| Potassium (K) | 0.046 ± 0.016 | 0.16 ± 0.15 | 0.052 ± 0.046 | 0.47 ± 0.00022 | 0.0092 ± 0.012 | 0.057 ± 0.035 | 0.0046 ± 0.00044 | 0.12 ± 0.10 |
| Calcium (Ca) | 0.032 ± 0.032 | 0.032 ± 0.045 | 0.035 ± 0.049 | 0.00 ± 0.00057 | 0.0040 ± 0.0056 | 0.00 ± 0.00034 | 0.00 ± 0.00025 | 0.00 ± 0.00023 |
| Scandium (Sc) | 0.00 ± 0.0020 | 0.00 ± 0.0025 | 0.00 ± 0.0029 | 0.00 ± 0.0026 | 0.00 ± 0.0014 | 0.00 ± 0.0015 | 0.022 ± 0.031 | 0.00 ± 0.0010 |
| Titanium (Ti) | 0.00 ± 0.000071 | 0.00 ± 0.000091 | 0.0055 ± 0.0078 | 0.051 ± 0.000093 | 0.0036 ± 0.0050 | 0.00 ± 0.000054 | 0.0086 ± 0.012 | 0.00 ± 0.000037 |
| Vanadium (V) | 0.00 ± 0.000013 | 0.00 ± 0.000017 | 0.00 ± 0.000019 | 0.00 ± 0.000017 | 0.00 ± 0.000094 | 0.00 ± 0.000010 | 0.00 ± 0.0000075 | 0.00 ± 0.0000069 |
| Chromium (Cr) | 0.00051 ± 0.00089 | 0.00028 ± 0.00040 | 0.00 ± 0.000065 | 0.00 ± 0.000057 | 0.00 ± 0.000032 | 0.00 ± 0.000034 | 0.00034 ± 0.00048 | 0.00 ± 0.000023 |
| Manganese (Mn) | 0.0015 ± 0.0014 | 0.00069 ± 0.00098 | 0.0016 ± 0.0023 | 0.0011 ± 0.00020 | 0.0013 ± 0.0012 | 0.00033 ± 0.00047 | 0.00057 ± 0.00080 | 0.0016 ± 0.0018 |
| Iron (Fe) | 0.036 ± 0.014 | 0.10 ± 0.095 | 0.049 ± 0.048 | 0.029 ± 0.00035 | 0.00 ± 0.00019 | 0.047 ± 0.040 | 0.024 ± 0.012 | 0.065 ± 0.0091 |
| Cobalt (Co) | 0.00 ± 0.0000088 | 0.00 ± 0.000011 | 0.00 ± 0.000013 | 0.00013 ± 0.000011 | 0.00 ± 0.0000063 | 0.00021 ± 0.00030 | 0.00020 ± 0.00028 | 0.00 ± 0.0000046 |
| Nickel (Ni) | 0.00028 ± 0.00049 | 0.00 ± 0.000028 | 0.00075 ± 0.0011 | 0.00 ± 0.000028 | 0.00045 ± 0.00064 | 0.00 ± 0.000017 | 0.00069 ± 0.00097 | 0.00043 ± 0.00026 |
| Copper (Cu) | 0.028 ± 0.047 | 0.027 ± 0.034 | 0.0098 ± 0.0028 | 0.15 ± 0.00018 | 0.00 ± 0.000098 | 0.0035 ± 0.0049 | 0.0019 ± 0.0000053 | 0.069 ± 0.090 |
| Zinc (Zn) | 0.026 ± 0.036 | 0.027 ± 0.031 | 0.0026 ± 0.0020 | 0.011 ± 0.000097 | 0.0013 ± 0.0015 | 0.0023 ± 0.0032 | 0.00041 ± 0.000028 | 0.0046 ± 0.00037 |
| Arsenic (As) | 0.0006 ± 0.00078 | 0.00 ± 0.000045 | 0.00 ± 0.000052 | 0.00067 ± 0.000045 | 0.00 ± 0.000025 | 0.00 ± 0.000027 | 0.000062 ± 0.000087 | 0.00034 ± 0.00048 |
| Selenium (Se) | 0.00016 ± 0.00028 | 0.0064 ± 0.0017 | 0.0022 ± 0.0032 | 0.00 ± 0.000080 | 0.0017 ± 0.00092 | 0.00 ± 0.000047 | 0.00034 ± 0.00048 | 0.0034 ± 0.0017 |
| Bromine (Br) | 0.0017 ± 0.0018 | 0.0031 ± 0.0044 | 0.0079 ± 0.00064 | 0.0020 ± 0.000023 | 0.020 ± 0.00098 | 0.0077 ± 0.010 | 0.024 ± 0.0043 | 0.019 ± 0.0012 |
| Rubidium (Rb) | 0.00 ± 0.000022 | 0.0035 ± 0.0048 | 0.0057 ± 0.0059 | 0.0026 ± 0.000028 | 0.00011 ± 0.00016 | 0.00095 ± 0.0013 | 0.00 ± 0.000013 | 0.00066 ± 0.00047 |
| Strontium (Sr) | 0.0017 ± 0.00036 | 0.0076 ± 0.0084 | 0.0068 ± 0.0014 | 0.0028 ± 0.000028 | 0.0023 ± 0.00057 | 0.0038 ± 0.0013 | 0.0018 ± 0.00075 | 0.0046 ± 0.0025 |
| Yttrium (Y) | 0.0013 ± 0.0014 | 0.0037 ± 0.0013 | 0.0057 ± 0.0041 | 0.0054 ± 0.000028 | 0.0014 ± 0.00029 | 0.0012 ± 0.0018 | 0.00085 ± 0.000067 | 0.0022 ± 0.0032 |
| Zirconium (Zr) | 0.0027 ± 0.0028 | 0.0047 ± 0.0014 | 0.0025 ± 0.0027 | 0.011 ± 0.00011 | 0.0016 ± 0.0023 | 0.0003 ± 0.00089 | 0.00074 ± 0.0010 | 0.0013 ± 0.00079 |
| Niobium (Nb) | 0.00 ± 0.000040 | 0.00092 ± 0.00090 | 0.00027 ± 0.00039 | 0.00 ± 0.000051 | 0.0016 ± 0.0023 | 0.00082 ± 0.0012 | 0.00042 ± 0.00060 | 0.00 ± 0.000021 |
| Molybdenum (Mo) | 0.0012 ± 0.0019 | 0.0044 ± 0.0062 | 0.0020 ± 0.00084 | 0.00 ± 0.00011 | 0.00 ± 0.000060 | 0.00063 ± 0.00089 | 0.0025 ± 0.00092 | 0.00 ± 0.000044 |
| Silver (Ag) | 0.00 ± 0.00011 | 0.00 ± 0.00014 | 0.00 ± 0.00016 | 0.00 ± 0.00014 | 0.0010 ± 0.0014 | 0.00 ± 0.000081 | 0.00 ± 0.000060 | 0.00 ± 0.000055 |
| Cadmium (Cd) | 0.00 ± 0.00015 | 0.00 ± 0.00019 | 0.00 ± 0.00022 | 0.00 ± 0.00019 | 0.0034 ± 0.0049 | 0.00 ± 0.00011 | 0.0029 ± 0.00093 | 0.0020 ± 0.0029 |
| Indium (In) | 0.00082 ± 0.0013 | 0.0011 ± 0.0016 | 0.00069 ± 0.00097 | 0.00 ± 0.00013 | 0.00068 ± 0.00096 | 0.0025 ± 0.0036 | 0.0021 ± 0.0030 | 0.0018 ± 0.0026 |
| Tin (Sn) | 0.0045 ± 0.0078 | 0.014 ± 0.020 | 0.0067 ± 0.0025 | 0.00 ± 0.00024 | 0.0037 ± 0.00047 | 0.0034 ± 0.0048 | 0.0028 ± 0.0025 | 0.0074 ± 0.00049 |
| Antimony (Sb) | 0.0065 ± 0.011 | 0.015 ± 0.021 | 0.00 ± 0.00041 | 0.00 ± 0.00036 | 0.00 ± 0.00020 | 0.0072 ± 0.010 | 0.0020 ± 0.0029 | 0.00 ± 0.00015 |
| Cesium (Cs) | 0.0097 ± 0.0095 | 0.022 ± 0.031 | 0.010 ± 0.014 | 0.058 ± 0.0010 | 0.00 ± 0.00056 | 0.00 ± 0.00060 | 0.00 ± 0.00044 | 0.00 ± 0.00041 |
| Barium (Ba) | 0.00 ± 0.00059 | 0.00 ± 0.00077 | 0.00 ± 0.00086 | 0.00 ± 0.00089 | 0.00 ± 0.00042 | 0.00 ± 0.00046 | 0.00 ± 0.00034 | 0.00 ± 0.00031 |
| Lanthanum (La) | 0.015 ± 0.026 | 0.065 ± 0.025 | 0.055 ± 0.0026 | 0.00 ± 0.0015 | 0.042 ± 0.044 | 0.0053 ± 0.0075 | 0.019 ± 0.028 | 0.036 ± 0.021 |
| Wolfram (W) | 0.0034 ± 0.0059 | 0.0082 ± 0.0061 | 0.00 ± 0.00033 | 0.00 ± 0.00029 | 0.0037 ± 0.0018 | 0.0034 ± 0.0049 | 0.0019 ± 0.0028 | 0.00 ± 0.00012 |
| Gold (Au) | 0.00 ± 0.000066 | 0.0032 ± 0.0045 | 0.00 ± 0.000098 | 0.00 ± 0.000085 | 0.00062 ± 0.00088 | 0.00 ± 0.000051 | 0.00022 ± 0.00031 | 0.0012 ± 0.0017 |
| Mercury (Hg) | 0.00034 ± 0.00059 | 0.0014 ± 0.0020 | 0.00 ± 0.000052 | 0.00 ± 0.000045 | 0.00020 ± 0.00028 | 0.0014 ± 0.0020 | 0.00 ± 0.000020 | 0.00024 ± 0.00033 |
| Lead (Pb) | 0.00 ± 0.000066 | 0.0010 ± 0.0015 | 0.00 ± 0.000098 | 0.0036 ± 0.000085 | 0.0015 ± 0.0021 | 0.0014 ± 0.000962 | 0.00076 ± 0.0011 | 0.0012 ± 0.0017 |
| Uranium (U) | 0.0050 ± 0.0044 | 0.0028 ± 0.0027 | 0.0011 ± 0.0015 | 0.0035 ± 0.00015 | 0.0034 ± 0.0044 | 0.00 ± 0.000092 | 0.0026 ± 0.0037 | 0.00 ± 0.000062 |

Table 1 (cont'd)

| | Average ± Standard Deviation of Percent PM$_{2.5}$ Mass | | | | | | | |
|---|---|---|---|---|---|---|---|---|
| | Subtropical | | | | Tropical | | | |
| | Everglades National Park, Florida (FL2) | | | | Borneo, Malaysia | | | |
| Aging Time | 2 days | | 7 days | | 2 days | | 7 days | |
| | Fresh 2 | Aged 2 | Fresh 7 | Aged 7 | Fresh 2 | Aged 2 | Fresh 7 | Aged 7 |
| Peat IDs in the average[c] | PEAT010, PEAT011, PEAT012, PEAT015 | | PEAT016, PEAT017, PEAT018 | | PEAT036, PEAT038 | | PEAT039, PEAT041 | |
| Nitric Acid (HNO$_3$) | 0.38 ± 0.13 | 0.47 ± 0.37 | 0.28 ± 0.042 | 0.25 ± 0.13 | 0.20 ± 0.0080 | 0.26 ± 0.040 | 0.23 ± 0.18 | 0.17 ± 0.026 |
| Ammonia (NH$_3$) | 51.12 ± 27.44 | 14.37 ± 5.54 | 63.89 ± 25.088 | 4.79 ± 0.60 | 20.34 ± 0.0030 | 9.67 ± 2.25 | 25.50 ± 1.98 | 4.88 ± 1.76 |
| Water-Soluble Sodium (Na$^+$) | 0.047 ± 0.018 | 0.056 ± 0.016 | 0.030 ± 0.017 | 0.022 ± 0.0063 | 0.017 ± 0.0090 | 0.033 ± 0.023 | 0.018 ± 0.011 | 0.032 ± 0.017 |
| Water-Soluble Potassium (K$^+$) | 1.11 ± 2.15 | na[d] | 0.025 ± 0.017 | na[d] | 0.031 ± 0.028 | na[d] | 0.048 ± 0.035 | na[d] |
| Chloride (Cl$^-$) | 0.26 ± 0.072 | 0.21 ± 0.12 | 0.22 ± 0.018 | 0.086 ± 0.024 | 0.11 ± 0.024 | 0.10 ± 0.026 | 0.16 ± 0.073 | 0.10 ± 0.00025 |
| Nitrite (NO$_2^-$) | 0.058 ± 0.098 | 0.0020 ± 0.0031 | 0.00085 ± 0.0015 | 0.0023 ± 0.00072 | 0.00 ± 0.00025 | 0.00098 ± 0.0014 | 0.00 ± 0.00030 | 0.015 ± 0.019 |
| Nitrate (NO$_3^-$) | 0.27 ± 0.26 | 2.64 ± 0.76 | 0.14 ± 0.097 | 7.76 ± 1.029 | 0.087 ± 0.046 | 0.91 ± 0.22 | 0.13 ± 0.12 | 4.69 ± 1.34 |
| Sulfate (SO$_4^=$) | 1.40 ± 1.89 | 1.33 ± 0.69 | 0.34 ± 0.022 | 1.99 ± 0.28 | 0.17 ± 0.024 | 0.56 ± 0.18 | 0.13 ± 0.062 | 1.96 ± 0.071 |
| Ammonium (NH$_4^+$) | 0.0013 ± 0.0015 | 0.37 ± 0.60 | 0.0036 ± 0.00092 | 4.55 ± 0.57 | 0.0017 ± 0.0011 | 0.83 ± 0.086 | 0.0027 ± 0.00048 | 4.74 ± 0.77 |
| OC1 (140°C) | 11.40 ± 1.25 | 7.017 ± 3.95 | 18.049 ± 2.22 | 4.012 ± 0.89 | 16.033 ± 2.088 | 6.37 ± 3.36 | 15.20 ± 1.21 | 5.83 ± 3.45 |
| OC2 (280°C) | 23.86 ± 6.033 | 16.25 ± 3.60 | 24.53 ± 3.41 | 12.12 ± 0.86 | 22.44 ± 1.91 | 18.78 ± 4.51 | 23.41 ± 0.25 | 12.14 ± 2.71 |
| OC3 (480°C) | 23.70 ± 7.73 | 21.13 ± 3.73 | 23.33 ± 2.32 | 17.83 ± 3.95 | 25.52 ± 2.55 | 28.64 ± 4.52 | 26.24 ± 1.16 | 20.82 ± 3.30 |
| OC4 (580°C) | 9.010 ± 3.51 | 8.53 ± 2.94 | 6.15 ± 0.95 | 5.65 ± 1.23 | 4.37 ± 0.18 | 8.32 ± 1.099 | 5.56 ± 1.40 | 5.59 ± 0.82 |
| Pyrolized Carbon (OP) | 10.73 ± 2.31 | 9.89 ± 3.86 | 13.036 ± 1.020 | 12.30 ± 1.22 | 10.74 ± 0.66 | 12.56 ± 4.73 | 10.35 ± 0.11 | 13.15 ± 2.69 |
| Organic Carbon (OC)[g] | 78.69 ± 18.69 | 62.82 ± 14.029 | 85.086 ± 5.65 | 51.90 ± 3.86 | 79.10 ± 3.21 | 74.66 ± 18.22 | 80.76 ± 0.99 | 57.53 ± 11.32 |
| EC1 (580°C) | 8.59 ± 4.065 | 8.56 ± 2.77 | 7.53 ± 1.22 | 11.035 ± 1.98 | 6.43 ± 0.48 | 8.57 ± 3.59 | 6.85 ± 0.21 | 9.13 ± 0.94 |
| EC2 (740°C) | 6.54 ± 2.76 | 3.42 ± 3.41 | 7.59 ± 1.66 | 3.35 ± 2.14 | 5.12 ± 0.25 | 6.18 ± 1.64 | 5.14 ± 0.16 | 4.69 ± 0.81 |
| EC3 (840°C) | 0.00 ± 0.00029 | 0.00 ± 0.00026 | 0.00 ± 0.00027 | 0.00 ± 0.00016 | 0.00 ± 0.00017 | 0.00 ± 0.00020 | 0.00 ± 0.00020 | 0.00 ± 0.00018 |
| Elemental Carbon (EC)[g] | 4.40 ± 1.51 | 2.084 ± 0.52 | 2.084 ± 1.81 | 2.092 ± 1.11 | 0.82 ± 0.074 | 2.19 ± 0.50 | 1.63 ± 0.16 | 0.67 ± 0.94 |
| Total Carbon (TC) | 83.090 ± 19.45 | 64.90 ± 14.48 | 87.17 ± 7.38 | 54.00 ± 4.57 | 79.92 ± 3.29 | 76.86 ± 18.72 | 82.39 ± 1.14 | 58.20 ± 10.38 |
| Water-Soluble OC (WSOC) | 31.71 ± 8.36 | 28.89 ± 4.08 | 34.33 ± 4.82 | 23.28 ± 2.80 | 14.62 ± 0.92 | 22.88 ± 2.33 | 17.15 ± 2.80 | 22.90 ± 0.76 |
| Formic acid (CH$_2$O$_2$) | 0.14 ± 0.17 | 0.30 ± 0.052 | 0.054 ± 0.020 | 0.42 ± 0.23 | 0.10 ± 0.014 | 0.26 ± 0.049 | 0.13 ± 0.019 | 0.42 ± 0.10 |
| Acetic acid (C$_2$H$_4$O$_2$) | 0.33 ± 0.25 | 0.38 ± 0.063 | 0.22 ± 0.12 | 0.35 ± 0.13 | 0.29 ± 0.0081 | 0.59 ± 0.24 | 0.58 ± 0.075 | 0.56 ± 0.018 |
| Oxalic acid (C$_2$H$_2$O$_4$) | 0.11 ± 0.058 | 0.94 ± 0.22 | 0.082 ± 0.029 | 3.14 ± 0.56 | 0.26 ± 0.12 | 1.14 ± 0.21 | 0.43 ± 0.22 | 3.36 ± 0.28 |
| Propionic acid (C$_3$H$_5$O$_2$) | 0.0064 ± 0.013 | 0.00 ± 0.00018 | 0.018 ± 0.031 | 0.012 ± 0.020 | 0.045 ± 0.019 | 0.0095 ± 0.013 | 0.012 ± 0.017 | 0.066 ± 0.094 |
| Levoglucosan (C$_6$H$_{10}$O$_5$) | 1.08 ± 1.34 | 0.86 ± 1.073 | 2.22 ± 0.66 | 0.62 ± 0.81 | 2.52 ± 0.016 | 2.28 ± 0.99 | 4.38 ± 0.50 | 2.53 ± 0.19 |
| Mannosan (C$_6$H$_{10}$O$_5$) | 0.00 ± 0.00045 | 0.00 ± 0.00039 | 0.056 ± 0.097 | 0.24 ± 0.42 | 0.00 ± 0.00027 | 0.00 ± 0.00030 | 0.19 ± 0.26 | 0.082 ± 0.12 |
| Galactose/Maltitol (C$_6$H$_{12}$O$_6$/C$_{12}$H$_{24}$O$_{11}$) | 0.00 ± 0.00023 | 0.00 ± 0.00020 | 0.00 ± 0.00021 | 0.00 ± 0.00012 | 0.00 ± 0.00014 | 0.13 ± 0.18 | 0.00 ± 0.00017 | 0.00 ± 0.00014 |
| Glycerol (C$_3$H$_8$O$_3$) | 0.00 ± 0.0000041 | 0.00 ± 0.0000036 | 0.00 ± 0.0000038 | 0.00 ± 0.0000022 | 0.00 ± 0.0000025 | 0.00 ± 0.0000028 | 0.00 ± 0.0000030 | 0.00 ± 0.0000024 |
| Mannitol (C$_6$H$_{14}$O$_6$) | 0.00 ± 0.000083 | 0.00 ± 0.000072 | 0.00 ± 0.000075 | 0.00 ± 0.000043 | 0.011 ± 0.016 | 0.00 ± 0.000055 | 0.00 ± 0.000060 | 0.00 ± 0.000049 |
| Aluminum (Al) | 0.043 ± 0.86 | 0.070 ± 1.20 | 0.00024 ± 0.0041 | 0.00 ± 0.026[c] | 0.033 ± 0.47 | 0.085 ± 0.030 | 0.045 ± 0.64 | 0.15 ± 0.030 |
| Silicon (Si) | 0.027 ± 0.52 | 0.26 ± 3.92 | 0.00 ± 0.00059 | 0.46 ± 0.31 | 0.012 ± 0.17 | 0.082 ± 0.0036 | 0.00 ± 0.00043 | 0.69 ± 0.0043 |
| Phosphorous (P) | 0.00 ± 0.00013 | 0.00 ± 0.00011 | 0.00 ± 0.00012 | 0.00 ± 0.000061 | 0.00 ± 0.000072 | 0.00 ± 0.000071 | 0.00 ± 0.000086 | 0.00 ± 0.000071 |


Table 1 (cont'd)

| | Average ± Standard Deviation of Percent PM$_{2.5}$ Mass | | | | | | | |
| --- | --- | --- | --- | --- | --- | --- | --- | --- |
| | Subtropical | | | | Tropical | | | |
| | Everglades National Park, Florida (FL2) | | | | Borneo, Malaysia | | | |
| Aging Time | 2 days | | 7 days | | 2 days | | 7 days | |
| | Fresh 2 | Aged 2 | Fresh 7 | Aged 7 | Fresh 2 | Aged 2 | Fresh 7 | Aged 7 |
| Peat IDs in the average[c] | PEAT010, PEAT011, PEAT012, PEAT015 | | PEAT016, PEAT017, PEAT018 | | PEAT036, PEAT038 | | PEAT039, PEAT041 | |
| Sulfur (S) | 0.39 ± 0.23 | 0.59 ± 0.27 | 0.42 ± 0.066 | 1.12 ± 0.094 | 0.11 ± 0.12 | 0.39 ± 0.00013 | 0.029 ± 0.0022 | 0.83 ± 0.00026 |
| Chlorine (Cl) | 0.21 ± 0.088 | 0.065 ± 0.029 | 0.24 ± 0.024 | 0.038 ± 0.011 | 0.074 ± 0.0012 | 0.067 ± 0.000035 | 0.085 ± 0.0038 | 0.047 ± 0.000030 |
| Potassium (K) | 0.034 ± 0.015 | 0.51 ± 0.37 | 0.018 ± 0.014 | 0.22 ± 0.052 | 0.051 ± 0.049 | 0.084 ± 0.00010 | 0.028 ± 0.017 | 0.017 ± 0.00010 |
| Calcium (Ca) | 0.00 ± 0.00067 | 0.0081 ± 0.016 | 0.00 ± 0.00061 | 0.010 ± 0.014 | 0.0058 ± 0.0082 | 0.00 ± 0.00037 | 0.00 ± 0.00046 | 0.023 ± 0.00038 |
| Scandium (Sc) | 0.00 ± 0.0030 | 0.00 ± 0.0026 | 0.00 ± 0.0027 | 0.00 ± 0.0014 | 0.00 ± 0.0017 | 0.00 ± 0.0017 | 0.00 ± 0.0020 | 0.00 ± 0.0017 |
| Titanium (Ti) | 0.0061 ± 0.0079 | 0.017 ± 0.035 | 0.00 ± 0.000098 | 0.00 ± 0.000051 | 0.0073 ± 0.010 | 0.00 ± 0.000059 | 0.0066 ± 0.0094 | 0.00 ± 0.000059 |
| Vanadium (V) | 0.0010 ± 0.0020 | 0.00 ± 0.000017 | 0.00 ± 0.000018 | 0.0065 ± 0.0092 | 0.00 ± 0.000011 | 0.00 ± 0.000011 | 0.00 ± 0.000014 | 0.00 ± 0.000011 |
| Chromium (Cr) | 0.00 ± 0.000066 | 0.00056 ± 0.0011 | 0.00 ± 0.000061 | 0.00016 ± 0.00023 | 0.00 ± 0.000038 | 0.00 ± 0.000037 | 0.0026 ± 0.0037 | 0.00 ± 0.000037 |
| Manganese (Mn) | 0.0032 ± 0.0064 | 0.0051 ± 0.0050 | 0.0017 ± 0.0015 | 0.0034 ± 0.0043 | 0.0055 ± 0.0026 | 0.0075 ± 0.00013 | 0.0088 ± 0.00010 | 0.0046 ± 0.00013 |
| Iron (Fe) | 0.023 ± 0.021 | 0.065 ± 0.034 | 0.020 ± 0.016 | 0.091 ± 0.096 | 0.074 ± 0.0078 | 0.074 ± 0.00023 | 0.045 ± 0.020 | 0.043 ± 0.00023 |
| Cobalt (Co) | 0.000055 ± 0.00011 | 0.000045 ± 0.000090 | 0.00024 ± 0.00041 | 0.00 ± 0.0000064 | 0.00 ± 0.0000075 | 0.00061 ± 0.0000074 | 0.00 ± 0.0000090 | 0.000087 ± 0.0000074 |
| Nickel (Ni) | 0.00026 ± 0.00042 | 0.00 ± 0.000029 | 0.00 ± 0.000031 | 0.00038 ± 0.00054 | 0.00064 ± 0.00091 | 0.00 ± 0.000019 | 0.0034 ± 0.0014 | 0.00 ± 0.000019 |
| Copper (Cu) | 0.010 ± 0.0080 | 0.21 ± 0.23 | 0.0033 ± 0.0036 | 0.021 ± 0.0024 | 0.0054 ± 0.0042 | 0.0075 ± 0.00012 | 0.0091 ± 0.0013 | 0.0017 ± 0.00012 |
| Zinc (Zn) | 0.0039 ± 0.0011 | 0.0091 ± 0.0039 | 0.0021 ± 0.0019 | 0.023 ± 0.027 | 0.0043 ± 0.0037 | 0.00 ± 0.000063 | 0.0034 ± 0.0018 | 0.00 ± 0.000063 |
| Arsenic (As) | 0.00059 ± 0.00069 | 0.0013 ± 0.0020 | 0.00 ± 0.000049 | 0.00 ± 0.000025 | 0.00 ± 0.000030 | 0.00 ± 0.000030 | 0.00 ± 0.000036 | 0.0028 ± 0.000030 |
| Selenium (Se) | 0.0011 ± 0.0014 | 0.0023 ± 0.0018 | 0.0037 ± 0.0025 | 0.00016 ± 0.00023 | 0.0019 ± 0.0010 | 0.00 ± 0.000052 | 0.00086 ± 0.0012 | 0.00 ± 0.000052 |
| Bromine (Br) | 0.030 ± 0.015 | 0.0090 ± 0.0049 | 0.022 ± 0.0072 | 0.0088 ± 0.0036 | 0.011 ± 0.0015 | 0.012 ± 0.000015 | 0.012 ± 0.0026 | 0.0044 ± 0.000015 |
| Rubidium (Rb) | 0.00038 ± 0.00077 | 0.0015 ± 0.0014 | 0.0015 ± 0.0026 | 0.00 ± 0.000016 | 0.00039 ± 0.00056 | 0.00035 ± 0.000019 | 0.00 ± 0.000023 | 0.0017 ± 0.000019 |
| Strontium (Sr) | 0.0051 ± 0.0012 | 0.0044 ± 0.0023 | 0.0055 ± 0.0063 | 0.0033 ± 0.0022 | 0.0028 ± 0.00026 | 0.0021 ± 0.000019 | 0.0070 ± 0.00099 | 0.0029 ± 0.000019 |
| Yttrium (Y) | 0.0043 ± 0.0051 | 0.0021 ± 0.0034 | 0.0014 ± 0.00060 | 0.00 ± 0.0000016 | 0.0018 ± 0.0023 | 0.0032 ± 0.000019 | 0.0018 ± 0.0016 | 0.0027 ± 0.000019 |
| Zirconium (Zr) | 0.0041 ± 0.0038 | 0.0049 ± 0.0066 | 0.0040 ± 0.0069 | 0.0051 ± 0.0039 | 0.0048 ± 0.0038 | 0.0016 ± 0.000071 | 0.00052 ± 0.00074 | 0.00 ± 0.000071 |
| Niobium (Nb) | 0.0016 ± 0.0022 | 0.00080 ± 0.0013 | 0.0019 ± 0.0026 | 0.00 ± 0.000029 | 0.00095 ± 0.0014 | 0.00 ± 0.000034 | 0.0021 ± 0.0030 | 0.00026 ± 0.000034 |
| Molybdenum (Mo) | 0.0022 ± 0.0021 | 0.0013 ± 0.0017 | 0.0012 ± 0.0022 | 0.00081 ± 0.0011 | 0.00071 ± 0.0010 | 0.00 ± 0.000071 | 0.0044 ± 0.0018 | 0.0032 ± 0.000071 |
| Silver (Ag) | 0.0014 ± 0.0029 | 0.00 ± 0.00014 | 0.00 ± 0.00015 | 0.00 ± 0.000076 | 0.0025 ± 0.0035 | 0.00 ± 0.000089 | 0.0026 ± 0.0037 | 0.00 ± 0.000089 |
| Cadmium (Cd) | 0.00 ± 0.00022 | 0.00 ± 0.00019 | 0.0075 ± 0.013 | 0.0095 ± 0.0060 | 0.00044 ± 0.00063 | 0.00 ± 0.00012 | 0.00 ± 0.00015 | 0.00 ± 0.00012 |
| Indium (In) | 0.0069 ± 0.0049 | 0.0023 ± 0.0046 | 0.0054 ± 0.0093 | 0.0012 ± 0.0017 | 0.0048 ± 0.0067 | 0.0013 ± 0.000085 | 0.00087 ± 0.0012 | 0.00 ± 0.000085 |
| Tin (Sn) | 0.0061 ± 0.0072 | 0.0058 ± 0.012 | 0.0061 ± 0.0058 | 0.0068 ± 0.0096 | 0.0022 ± 0.0031 | 0.013 ± 0.000016 | 0.0038 ± 0.0054 | 0.012 ± 0.00016 |
| Antimony (Sb) | 0.00028 ± 0.00056 | 0.00040 ± 0.00052 | 0.00033 ± 0.00057 | 0.00050 ± 0.00071 | 0.00 ± 0.00024 | 0.0039 ± 0.00023 | 0.011 ± 0.0097 | 0.00 ± 0.00023 |
| Cesium (Cs) | 0.000088 ± 0.00018 | 0.028 ± 0.037 | 0.037 ± 0.064 | 0.00 ± 0.00057 | 0.028 ± 0.031 | 0.020 ± 0.00066 | 0.0077 ± 0.011 | 0.00 ± 0.00066 |
| Barium (Ba) | 0.00 ± 0.00088 | 0.00 ± 0.00085 | 0.00 ± 0.00081 | 0.00 ± 0.00044 | 0.00 ± 0.00050 | 0.00 ± 0.00050 | 0.00 ± 0.00060 | 0.00 ± 0.00050 |
| Lanthanum (La) | 0.054 ± 0.039 | 0.033 ± 0.039 | 0.036 ± 0.039 | 0.0049 ± 0.0070 | 0.041 ± 0.058 | 0.00 ± 0.00097 | 0.018 ± 0.025 | 0.080 ± 0.00097 |
| Wolfram (W) | 0.010 ± 0.012 | 0.0030 ± 0.0051 | 0.0080 ± 0.014 | 0.00 ± 0.00016 | 0.00 ± 0.00019 | 0.0058 ± 0.00019 | 0.00 ± 0.00023 | 0.00 ± 0.00019 |
| Gold (Au) | 0.0012 ± 0.0013 | 0.00082 ± 0.0016 | 0.0046 ± 0.0045 | 0.00033 ± 0.00047 | 0.00051 ± 0.00072 | 0.00 ± 0.000056 | 0.00041 ± 0.00058 | 0.00 ± 0.000056 |
| Mercury (Hg) | 0.00035 ± 0.00070 | 0.00091 ± 0.0015 | 0.00 ± 0.000049 | 0.00 ± 0.000025 | 0.00 ± 0.000030 | 0.00 ± 0.000030 | 0.00041 ± 0.00058 | 0.000087 ± 0.000030 |
| Lead (Pb) | 0.0017 ± 0.0035 | 0.0012 ± 0.0024 | 0.0018 ± 0.0031 | 0.0028 ± 0.0026 | 0.0031 ± 0.0044 | 0.00052 ± 0.000056 | 0.0016 ± 0.0022 | 0.00 ± 0.000056 |
| Uranium (U) | 0.0027 ± 0.0031 | 0.0023 ± 0.0026 | 0.0044 ± 0.0077 | 0.0017 ± 0.0023 | 0.00 ± 0.00010 | 0.0033 ± 0.00010 | 0.0057 ± 0.00076 | 0.0062 ± 0.00010 |

[a]Analytical uncertainties are used for species below the minimum detection limit, mostly for carbohydrate species and elements with an average concentration of 0.00
[b]Only one sample was analyzed for elements by x-ray fluorescence with abundance and measurement uncertainty
[c]Peat ID code, detailed operation parameters are reported in Watson et al. (2019)
[d]Data not available; water-soluble $K^+$ data were contaminated for aged samples due to the use of potassium iodide denuder downstream of the oxidation flow reactor
[e]WSOC measures from Peat sample ID PEAT028 was invalidated due to a crack in the test tube. Therefore, only two measurements are used to calculate the average and standard deviation.
[f]Data not available due to the invalidated citric acid impregnated filter sample
[g]The carbon analysis follows the IMPROVE_A thermal/optical reflectance protocol (Chow et al., 2007) that is applied in long-term U.S. non-urban IMPROVE and urban Chemical Speciation Network.
Organic carbon (OC) is the sum of OC1+OC2+OC3+OC4 plus pyrolized carbon (OP). Elemental carbon (EC) is the sum of EC1+EC2+EC3 minus OP. Total carbon is the sum of OC and EC. Since a
large fraction of OP (7–13 %) are found in smoldering peat combustion emissions--indicative of higher molecular-weight compounds that are likely to char, the resulting EC are lower than the individual
EC fraction after OP correction.
Table 2. Equivalence measures[a] for comparison of $PM_{2.5}$ peat source profiles.

**A | All Fresh (Profile #1) vs. All Aged (Profile #2) by Biome (group comparison of fresh and aged samples)**

| Peat region[b] | Peats Included | n1[c] | n2[c] | < 1 σ | 1 - 2 σ | 2 - 3 σ | > 3 σ | Correlation Coefficient | P-value[d] |
|---|---|---|---|---|---|---|---|---|---|
| Boreal | Russia + Siberia | 12 | 12 | 93.60% | 5.60% | 0.80% | 0.00% | 0.995 | 0.00012 |
| Boreal + Temperate | Russia + Siberia + Alaska | 17 | 17 | 95.20% | 4.80% | 0.00% | 0.00% | 0.996 | 0.00010 |
| Temperate | Alaska | 5 | 5 | 96.00% | 4.00% | 0.00% | 0.00% | 0.997 | 0.00008 |
| Subtropical 1 | Florida-1 (FL1) | 4 | 4 | 77.60% | 14.40% | 5.60% | 2.40% | 0.993 | 0.94570 |
| Subtropical 2 | Florida-2 (FL2) | 7 | 7 | 77.78% | 21.43% | 0.79% | 0.00% | 0.986 | 0.00001 |
| Subtropical 1 + Temperate | Florida-1 + Alaska | 9 | 9 | 96.83% | 3.17% | 0.00% | 0.00% | 0.996 | 0.00073 |
| Subtropical 2 + Temperate | Florida-2 + Alaska | 12 | 12 | 81.75% | 18.25% | 0.00% | 0.00% | 0.992 | 0.00001 |
| Tropical | Malaysia | 4 | 4 | 78.57% | 18.25% | 1.59% | 1.59% | 0.994 | 0.00195 |
| Subtropical 1 + Tropical | Florida-1 + Malaysia | 8 | 8 | 83.33% | 15.87% | 0.00% | 0.79% | 0.995 | 0.01686 |
| Subtropical 2 + Tropical | Florida-2 + Malaysia | 11 | 11 | 80.16% | 19.05% | 0.79% | 0.00% | 0.991 | 0.00003 |

**B | Fresh 2 vs. Aged 2 by Biome (paired comparison for 2-day aging)**

| Peat region | Peats Included | n1 | n2 | < 1 σ | 1 - 2 σ | 2 - 3 σ | > 3 σ | Correlation Coefficient | P-value |
|---|---|---|---|---|---|---|---|---|---|
| Boreal | Russia + Siberia | 6 | 6 | 94.40% | 3.20% | 2.40% | 0.00% | 0.997 | 0.00088 |
| Boreal + Temperate | Russia + Siberia + Alaska | 9 | 9 | 95.20% | 4.00% | 0.80% | 0.00% | 0.997 | 0.00237 |
| Temperate | Alaska | 3 | 3 | 86.40% | 11.20% | 0.80% | 1.60% | 0.997 | 0.02474 |
| Subtropical 1 | Florida-1 | 2 | 2 | 78.86% | 13.82% | 3.25% | 4.07% | 0.994 | 0.30785 |
| Subtropical 2 | Florida-2 | 4 | 4 | 86.51% | 11.90% | 0.79% | 0.79% | 0.992 | 0.00000 |
| Subtropical 1 + Temperate | Florida-1 + Alaska | 5 | 5 | 92.00% | 7.20% | 0.80% | 0.00% | 0.997 | 0.04329 |
| Subtropical 2 + Temperate | Florida-2 + Alaska | 7 | 7 | 95.24% | 3.97% | 0.00% | 0.79% | 0.996 | 0.00002 |
| Tropical | Malaysia | 2 | 2 | 80.00% | 5.33% | 5.33% | 9.33% | 0.996 | 0.95960 |
| Subtropical 1 + Tropical | Florida-1 + Malaysia | 4 | 4 | 88.89% | 8.73% | 1.59% | 0.79% | 0.996 | 0.62905 |
| Subtropical 2 + Tropical | Florida-2 + Malaysia | 6 | 6 | 93.65% | 5.56% | 0.00% | 0.79% | 0.995 | 0.00002 |

**C | Fresh 7 vs. Aged 7 by Biome (paired comparison for 7-day aging)**

| Peat region | Peats Included | n1 | n2 | < 1 σ | 1 - 2 σ | 2 - 3 σ | > 3 σ | Correlation Coefficient | P-value |
|---|---|---|---|---|---|---|---|---|---|
| Boreal | Russia + Siberia | 6 | 6 | 76.00% | 20.80% | 1.60% | 1.60% | 0.992 | 0.00007 |
| Boreal + Temperate | Russia + Siberia + Alaska | 8 | 8 | 76.80% | 20.00% | 0.80% | 2.40% | 0.993 | 0.00003 |
| Temperate | Alaska | 2 | 2 | 64.86% | 25.68% | 2.70% | 6.76% | 0.993 | 0.00000 |
| Subtropical 1 | Florida-1 | 2 | 2 | 63.20% | 13.60% | 7.20% | 16.00% | 0.998 | 0.00027 |
| Subtropical 2 | Florida-2 | 3 | 3 | 66.67% | 9.52% | 3.17% | 20.63% | 0.975 | 0.00003 |
| Subtropical 1 + Temperate | Florida-1 + Alaska | 4 | 4 | 88.10% | 7.94% | 3.97% | 0.00% | 0.994 | 0.00004 |
| Subtropical 2 + Temperate | Florida-2 + Alaska | 5 | 5 | 73.02% | 19.84% | 3.97% | 3.17% | 0.984 | 0.00001 |
| Tropical | Malaysia | 2 | 2 | 41.33% | 21.33% | 24.00% | 13.33% | 0.989 | 0.00017 |
| Subtropical 1 + Tropical | Florida-1 + Malaysia | 4 | 4 | 72.22% | 23.81% | 0.79% | 3.17% | 0.993 | 0.00156 |
| Subtropical 2 + Tropical | Florida-2 + Malaysia | 5 | 5 | 73.02% | 8.73% | 1.59% | 16.67% | 0.983 | 0.00004 |

**D | Fresh 2 vs. Fresh 7 by Biome (comparison between different experiments for unaged fresh profiles)**

| Peat region | Peats Included | n1 | n2 | < 1 σ | 1 - 2 σ | 2 - 3 σ | > 3 σ | Correlation Coefficient | P-value |
|---|---|---|---|---|---|---|---|---|---|
| Boreal | Russia + Siberia | 6 | 6 | 97.62% | 2.38% | 0.00% | 0.00% | 0.999 | 0.00004 |
| Boreal + Temperate | Russia + Siberia + Alaska | 9 | 8 | 100.00% | 0.00% | 0.00% | 0.00% | 0.999 | 0.00148 |
| Temperate | Alaska | 3 | 2 | 91.27% | 6.35% | 0.79% | 1.59% | 0.996 | 0.12876 |
| Subtropical 1 | Florida-1 | 2 | 2 | 90.32% | 6.45% | 1.61% | 1.61% | 0.999 | 0.00001 |
| Subtropical 2 | Florida-2 | 4 | 3 | 97.62% | 1.59% | 0.79% | 0.00% | 0.999 | 0.00032 |
| Subtropical 1 + Temperate | Florida-1 + Alaska | 5 | 4 | 99.21% | 0.79% | 0.00% | 0.00% | 0.998 | 0.00308 |
| Subtropical 2 + Temperate | Florida-2 + Alaska | 7 | 5 | 100.00% | 0.00% | 0.00% | 0.00% | 0.998 | 0.02743 |
| Tropical | Malaysia | 2 | 2 | 81.10% | 10.24% | 3.15% | 5.51% | 0.999 | 0.00006 |
| Subtropical 1 + Tropical | Florida-1 + Malaysia | 4 | 4 | 94.49% | 4.72% | 0.79% | 0.00% | 1.000 | 0.03537 |
| Subtropical 2 + Tropical | Florida-2 + Malaysia | 6 | 5 | 98.43% | 1.57% | 0.00% | 0.00% | 0.999 | 0.00013 |

**E | Aged 2 vs. Aged 7 by Biome (comparison between different experiments for the 2- and 7-day aging times)**

| Peat region | Peats Included | n1 | n2 | < 1 σ | 1 - 2 σ | 2 - 3 σ | > 3 σ | Correlation Coefficient | P-value |
|---|---|---|---|---|---|---|---|---|---|
| Boreal | Russia + Siberia | 6 | 6 | 95.20% | 3.20% | 1.60% | 0.00% | 0.997 | 0.00018 |
| Boreal + Temperate | Russia + Siberia + Alaska | 9 | 8 | 94.40% | 3.20% | 1.60% | 0.80% | 0.998 | 0.00002 |
| Temperate | Alaska | 3 | 2 | 66.22% | 27.03% | 5.41% | 1.35% | 0.996 | 0.00000 |
| Subtropical 1 | Florida-1 | 2 | 2 | 83.20% | 9.60% | 1.60% | 5.60% | 1.000 | 0.00017 |
| Subtropical 2 | Florida-2 | 4 | 3 | 88.89% | 8.73% | 0.00% | 2.38% | 0.994 | 0.00298 |
| Subtropical 1 + Temperate | Florida-1 + Alaska | 5 | 4 | 94.44% | 5.56% | 0.00% | 0.00% | 0.999 | 0.00000 |
| Subtropical 2 + Temperate | Florida-2 + Alaska | 7 | 5 | 81.75% | 16.67% | 0.00% | 1.59% | 0.997 | 0.00003 |
| Tropical | Malaysia | 2 | 2 | 81.33% | 13.33% | 1.33% | 4.00% | 0.997 | 0.00002 |
| Subtropical 1 + Tropical | Florida-1 + Malaysia | 4 | 4 | 92.06% | 7.14% | 0.79% | 0.00% | 0.999 | 0.00002 |
| Subtropical 2 + Tropical | Florida-2 + Malaysia | 6 | 5 | 93.65% | 3.97% | 0.79% | 1.59% | 0.996 | 0.00035 |

[a]For the $t$-test, a cutoff probability level of 5% is selected; if $P < 0.05$, there is a 95% probability that the two profiles are different. For correlations, $r > 0.8$ suggests similar profiles, $0.5 < r < 0.8$ indicates a moderate similarity, and $r < 0.5$ denotes little or no similarity. The $R/U$ ratio indicates the percentage of the >93 reported chemical abundances differ by more than an expected number of uncertainty intervals. The normal probability density function of 68%, 95.5%, and 99.7% for $\pm 1s$, $\pm 2s$, and $\pm 3s$, respectively, is used to evaluate the $R/U$ ratios. The two profiles are considered to be similar, within the uncertainties of the chemical abundances when 80% of the $R/U$ ratios are within $\pm 3s$, with $r > 0.8$ and $P > 0.05$. Species with $R/U$ ratios $> 3s$ are further examined as these may be markers that further allow source contributions to be distinguishes by receptor measurements. They may also reflect the sampling and analysis artifacts that are not representative of the larger population of source profiles.

[b]Unless otherwise noted, Boreal represents Russia and Siberia regions, Temperate represents northern Alaska region, Subtropical is divided into Subtropical 1 for Putnam (FL1) and Subtropical 2 for Everglades (FL2) peats, and Tropical represents Island of Borneo, Malaysia region.

[c]n1 and n2 denote number of samples in comparison

[d]Student $t$-test $P$-values


Table 3. Organic carbon diagnostic ratios for different peat samples.

| Peat Type | Atmospheric Aging time | OC/TC ± σ[a] | OM[b]/OC ± σ[a] | WSOC[c]/OC ± σ[a] | (Levoglucosan/2.25)[d]/OC ± σ[a] | (Oxalic acid/3.75)[e]/OC ± σ[a] | (Levoglucosan/2.25)[d]/WSOC ± σ[a] | (Oxalic acid/3.75)[e]/WSOC ± σ[a] |
|---|---|---|---|---|---|---|---|---|
| Odintsovo, Russia | Fresh 2 | 0.97 ± 0.11 | 1.7 ± 0.15 | 0.64 ± 0.075 | 0.27 ± 0.066 | 0.00047 ± 0.00029 | 0.42 ± 0.10 | 0.00073 ± 0.00045 |
| | Aged 2 | 0.97 ± 0.30 | 2.1 ± 0.46 | 0.70 ± 0.17 | 0.24 ± 0.10 | 0.0057 ± 0.0017 | 0.35 ± 0.13 | 0.0082 ± 0.0019 |
| | Fresh 7 | 0.97 ± 0.12 | 1.6 ± 0.14 | 0.59 ± 0.065 | 0.28 ± 0.030 | 0.0012 ± 0.001 | 0.48 ± 0.040 | 0.0021 ± 0.0017 |
| | Aged 7 | 0.95 ± 0.16 | 2.2 ± 0.26 | 0.71 ± 0.18 | 0.21 ± 0.051 | 0.019 ± 0.0055 | 0.30 ± 0.089 | 0.026 ± 0.0090 |
| Pskov, Siberia | Fresh 2 | 0.96 ± 0.12 | 1.3 ± 0.12 | 0.32 ± 0.039 | 0.04 ± 0.016 | 0.00023 ± 0.000050 | 0.12 ± 0.049 | 0.00069 ± 0.00015 |
| | Aged 2 | 0.96 ± 0.26 | 1.4 ± 0.27 | 0.44 ± 0.13 | 0.027 ± 0.0066 | 0.0051 ± 0.0021 | 0.063 ± 0.017 | 0.012 ± 0.0050 |
| | Fresh 7 | 0.99 ± 0.17 | 1.2 ± 0.14 | 0.40 ± 0.046 | 0.051 ± 0.013 | 0.00025 ± 0.000067 | 0.13 ± 0.055 | 0.00063 ± 0.00015 |
| | Aged 7 | 0.96 ± 0.14 | 1.5 ± 0.18 | 0.56 ± 0.17 | 0.031 ± 0.0043 | 0.019 ± 0.0073 | 0.057 ± 0.018 | 0.035 ± 0.016 |
| Northern Alaska, USA | Fresh 2 | 0.97 ± 0.12 | 1.3 ± 0.10 | 0.38 ± 0.12 | 0.10 ± 0.047 | 0.00013 ± 0.00010 | 0.27 ± 0.15 | 0.00035 ± 0.00028 |
| | Aged 2 | 0.97 ± 0.17 | 1.4 ± 0.18 | 0.40 ± 0.075 | 0.11 ± 0.025 | 0.0033 ± 0.00073 | 0.27 ± 0.063 | 0.0080 ± 0.0018 |
| | Fresh 7 | 0.95 ± 0.38 | 1.4 ± 0.39 | 0.45 ± 0.20 | 0.061 ± 0.019 | 0.00016 ± 0.00023 | 0.14 ± 0.052 | 0.00037 ± 0.00053 |
| | Aged 7 | 0.96 ± 0.35 | 1.8 ± 0.44 | 0.55 ± 0.16 | 0.046 ± 0.029 | 0.018 ± 0.0053 | 0.084 ± 0.052 | 0.034 ± 0.0076 |
| Putnam County Lakebed, Florida, USA (FL1) | Fresh 2 | 0.95 ± 0.19 | 1.3 ± 0.18 | 0.27 ± 0.074 | 0.02 ± 0.0026 | 0.00019 ± 0.00026 | 0.072 ± 0.017 | 0.00068 ± 0.0010 |
| | Aged 2 | 0.97 ± 0.10 | 1.4 ± 0.10 | 0.32 ± 0.067 | 0.018 ± 0.0013 | 0.0022 ± 0.0010 | 0.054 ± 0.011 | 0.0068 ± 0.0033 |
| | Fresh 7 | 0.98 ± 0.094 | 1.5 ± 0.10 | 0.24 ± 0.024 | 0.021 ± 0.0021 | na | 0.085 ± 0.009 | na |
| | Aged 7 | 0.98 ± 0.10 | 1.4 ± 0.11 | 0.34 ± 0.034 | 0.010 ± 0.0034 | 0.0044 ± 0.00082 | 0.029 ± 0.010 | 0.013 ± 0.0023 |
| Everglades, Florida, USA (FL2) | Fresh 2 | 0.95 ± 0.32 | 1.2 ± 0.28 | 0.40 ± 0.14 | 0.0061 ± 0.0077 | 0.00036 ± 0.00021 | 0.015 ± 0.019 | 0.00089 ± 0.00054 |
| | Aged 2 | 0.97 ± 0.31 | 1.5 ± 0.33 | 0.46 ± 0.12 | 0.0061 ± 0.0077 | 0.0044 ± 0.00082 | 0.013 ± 0.017 | 0.0086 ± 0.0024 |
| | Fresh 7 | 0.98 ± 0.11 | 1.1 ± 0.079 | 0.40 ± 0.063 | 0.012 ± 0.0035 | 0.00026 ± 0.000092 | 0.029 ± 0.009 | 0.00064 ± 0.00024 |
| | Aged 7 | 0.96 ± 0.11 | 1.6 ± 0.12 | 0.45 ± 0.063 | 0.0053 ± 0.007 | 0.016 ± 0.0031 | 0.012 ± 0.016 | 0.036 ± 0.0078 |
| Borneo, Malaysia | Fresh 2 | 0.99 ± 0.057 | 1.2 ± 0.051 | 0.18 ± 0.014 | 0.014 ± 0.00058 | 0.00087 ± 0.00042 | 0.077 ± 0.005 | 0.0047 ± 0.0023 |
| | Aged 2 | 0.97 ± 0.33 | 1.3 ± 0.31 | 0.31 ± 0.081 | 0.014 ± 0.0067 | 0.0041 ± 0.0012 | 0.044 ± 0.020 | 0.013 ± 0.0028 |
| | Fresh 7 | 0.98 ± 0.018 | 1.2 ± 0.015 | 0.21 ± 0.035 | 0.024 ± 0.0027 | 0.0014 ± 0.00072 | 0.11 ± 0.023 | 0.0067 ± 0.0036 |
| | Aged 7 | 0.99 ± 0.26 | 1.5 ± 0.29 | 0.40 ± 0.079 | 0.02 ± 0.0041 | 0.016 ± 0.0033 | 0.049 ± 0.0040 | 0.039 ± 0.0035 |

[a]Uncertainty associated with each ratio is calculated based on the square root of the individual uncertainties multiplied by the ratio (Bevington, 1969).

[b]OM (organic mass) is calculated by subtracting major ions (i.e., sum of $NH_4^+$, $NO_3^-$, and $SO_4^{2-}$), crustal components (2.2Al + 2.49 Si + 1.63 Ca + 1.94 Ti + 2.42 Fe) and elemental carbon from $PM_{2.5}$ mass

[c]WSOC: water-soluble organic carbon

[d]Levoglucosan/2.25 represents carbon content in levoglucosan, based on the chemical composition $C_6H_{10}O_5$.

[e]Oxalic acid/3.75 represents carbon content in oxalic acid based on the chemical composition $C_2H_2O_4$.

(a)   Upstream Filter Packs[a,b]

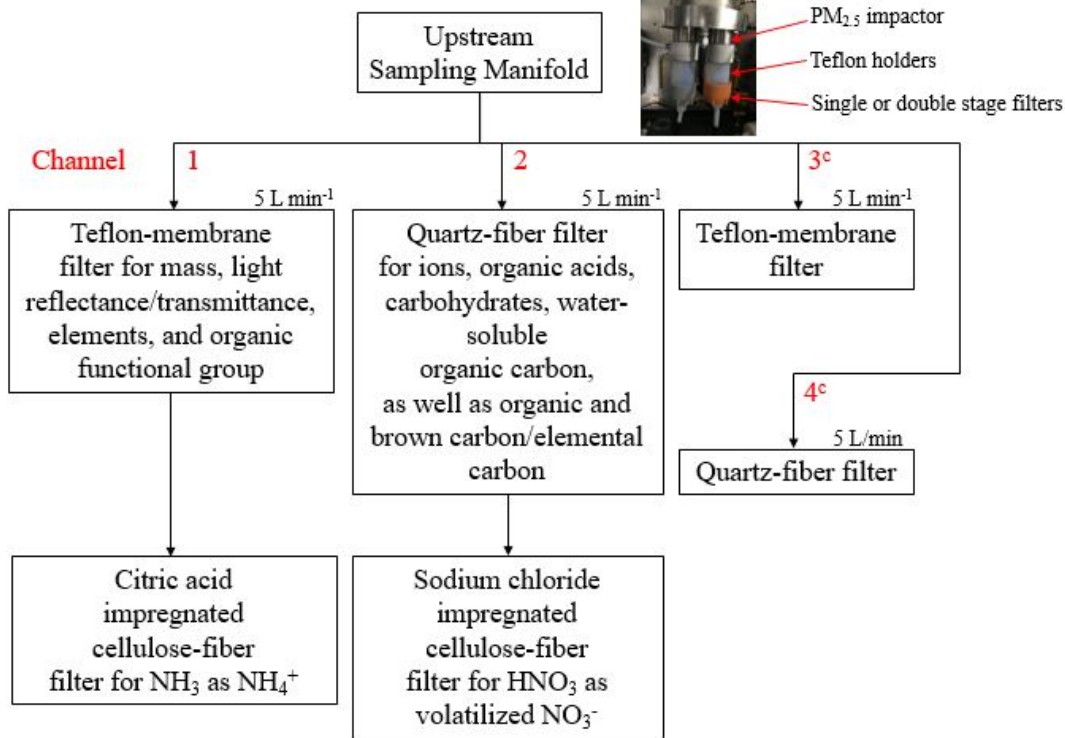


(b)   Downstream Filter Packs[a,b]

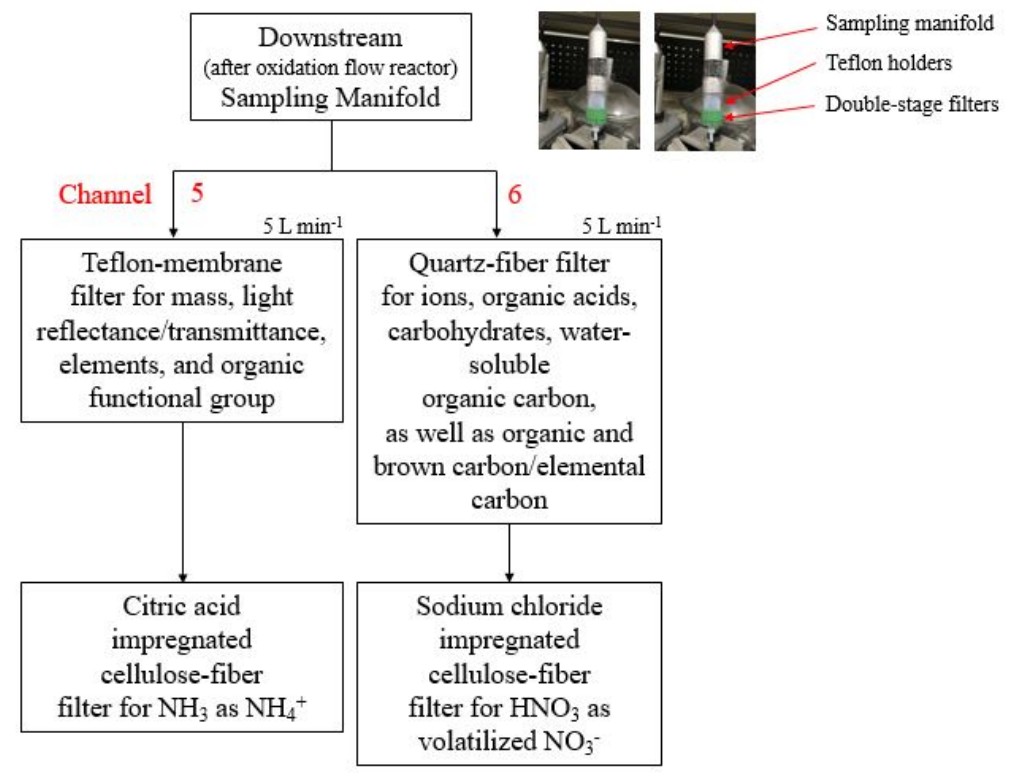



 [a]The filter types are: 1) Teflon-membrane filter (Teflo©, 2 μm pore size, R2PJ047, Pall Life Sciences, Port Washington, NY,
USA); 2) quartz-fiber filters (Tissuquartz, 2500 QAT-UP, Pall Life Sciences); and 3) citric acid and sodium chloride impregnated
cellulose-fiber filters (31ET, Whatman Labware Products, St. Louis, MO, USA).
[b]Analyses include: 1) mass by gravimetry (Model XP6 microbalance, Mettler-Toledo, Columbus, OH, USA); 2) light
reflectance/transmittance by UV/Vis spectrometry (Lambda35, Perkin Elmer, Waltham, MA, USA); 3) multiple elements by
energy-dispersive x-ray fluorescence (XRF) (Epsilon 5 PANalytical, Westborough, MA, USA); 4) four anions (chloride [$Cl^-$],
nitrite [$NO_2^-$], nitrate [$NO_3^-$], and sulfate [$SO_4^=$]); three cations (water-soluble sodium [$Na^+$], potassium [$K^+$], and ammonium
[$NH_4^+$]); and ten organic acids (i.e., formic acid, acetic acid, lactic acid, methanesulfonic acid, oxalic acid, propionic acid,
succinic acid, maleic acid, malonic acid, and glutaric acid) by ion chromatography (IC) with conductivity detector (Dionex
Model ICS-5000+, Thermo Scientific, Waltham, MA, USA); 5) 17 carbohydrates (i.e., levoglucosan, mannosan, galactosan,
glycerol, 2-methylerythritol, arabitol, mannitol, xylitol, erythritol, adonitol, inositol, glucose, galactose, arabinose, fructose,
sucrose, and trehalose) by IC with pulsed amperometric detector (Dionex Model ICS3000, Thermo Scientific, Waltham, MA,
USA); 6) water-soluble organic carbon (WSOC) by total organic carbon analyzer with non-dispersive infrared (NDIR) detector
(Shimadzu Corporation, Kyoto, Japan); 7) organic functional groups by Fourier-Transform Infrared (FTIR) spectroscopy
(VERTEX 70, Bruker, Billerica, MA, USA); and 8) organic, elemental, and brown carbon (OC, EC, and BrC) by
multiwavelength thermal/optical carbon analyzer (DRI Model 2015, Magee Scientific, Berkeley, CA, USA).
[c]Teflon-membrane filter samples from Channel 3 are to be analyzed for additional organic nitrogen speciation using Fourier
transform-ion cyclotron resonance mass spectrometry (FT-ICR-MS) at the Michigan Technological University. Quartz-fiber filter
samples from Channel 4 are to be analyzed for polar and non-polar organics at the Hong Kong Premium Services and Research
Laboratory.
Figure 1. Filter pack sampling configurations for upstream and downstream channels of the
oxidation flow reactor.

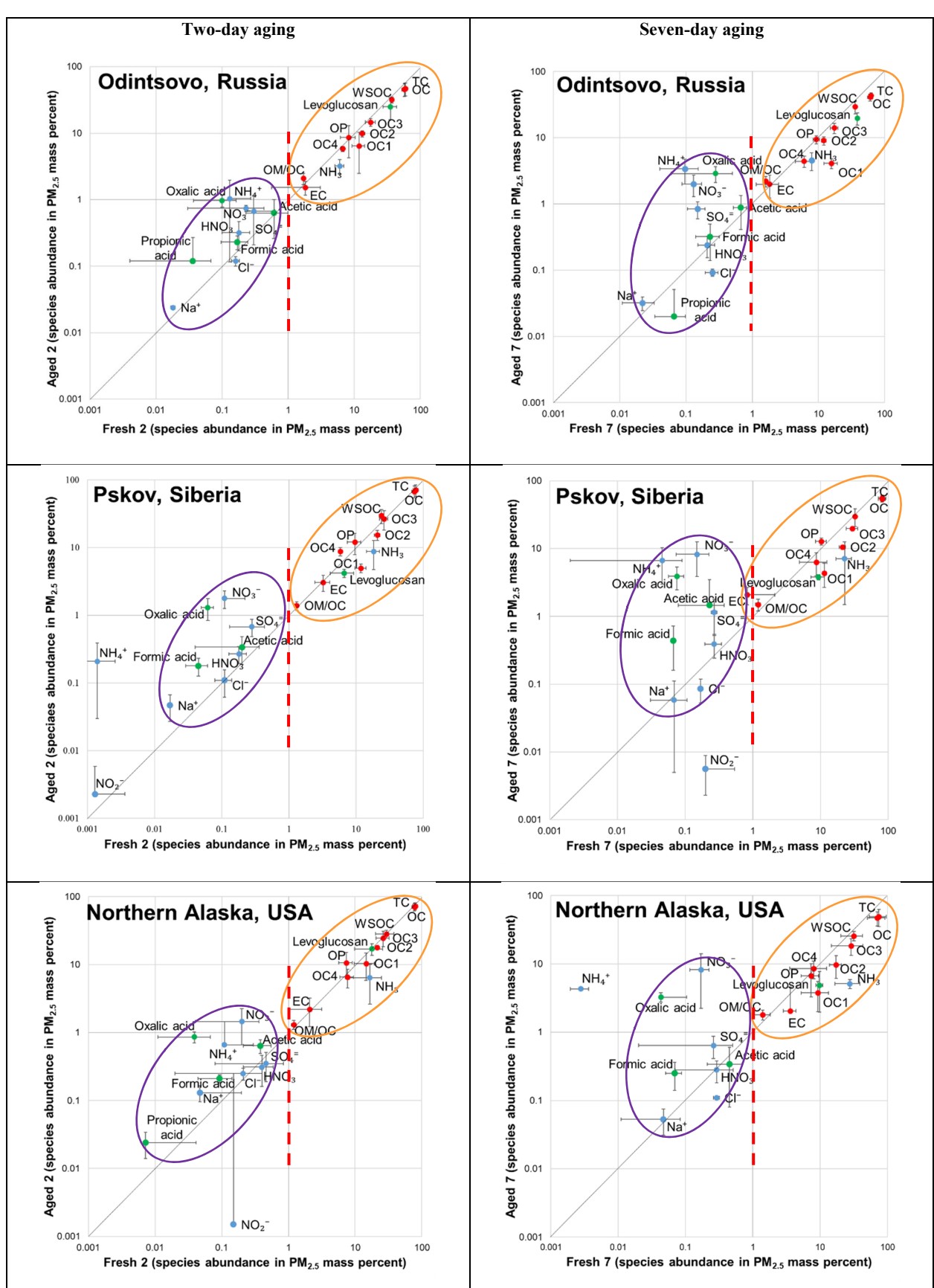

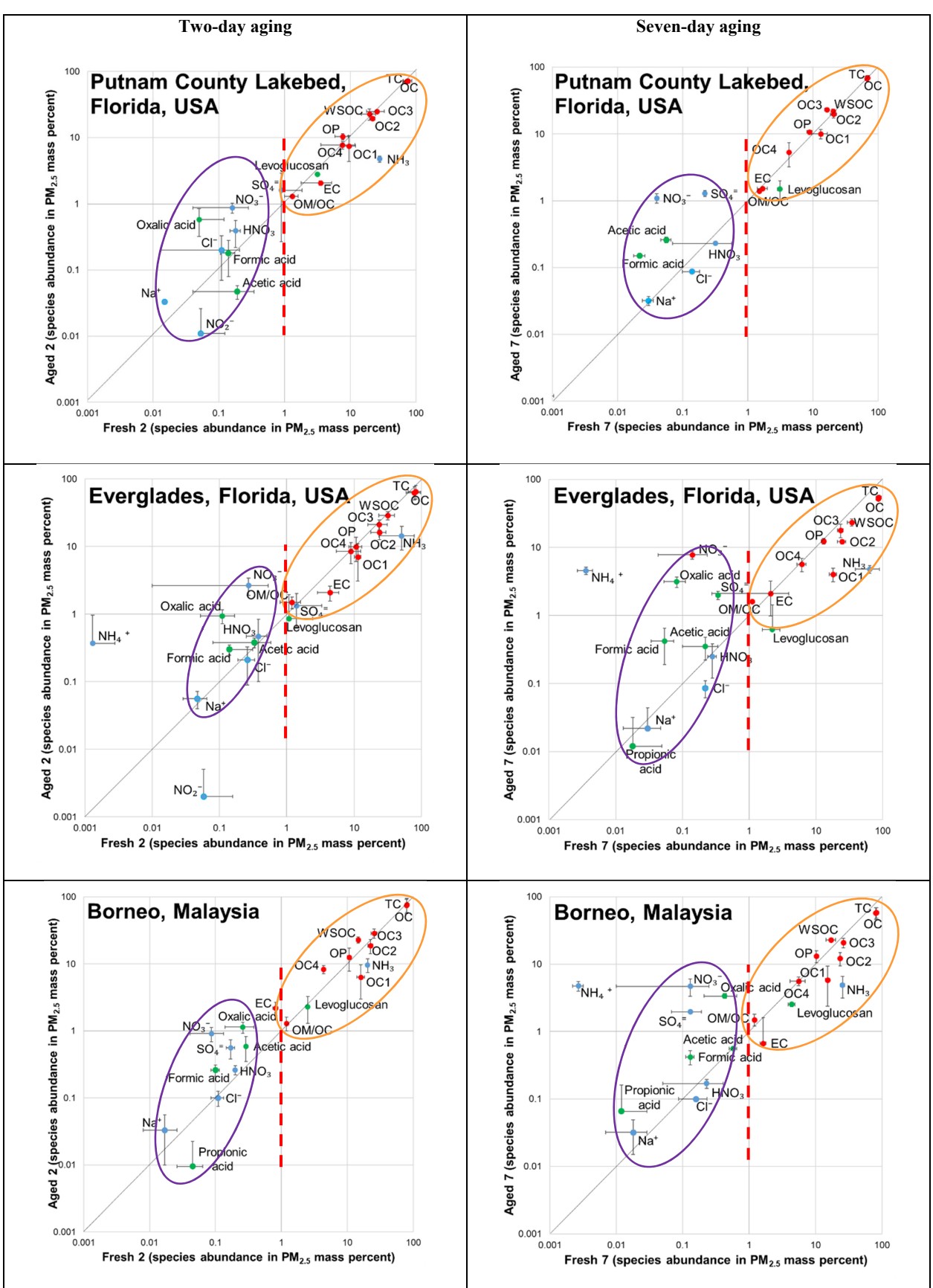

Figure 2.  Comparison between fresh and aged profile chemical abundances for each of the six
types of peat with 2- and 7-day aging times. Standard deviations associated with averages in x and
y axes are also shown. Vertical dashlines (red) on 1 % in x-axis intended to delineate the two
distinguished clusters: centered around 0.1 % for reactive/ionic species and centered around 10 %
for carbon compounds.

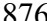

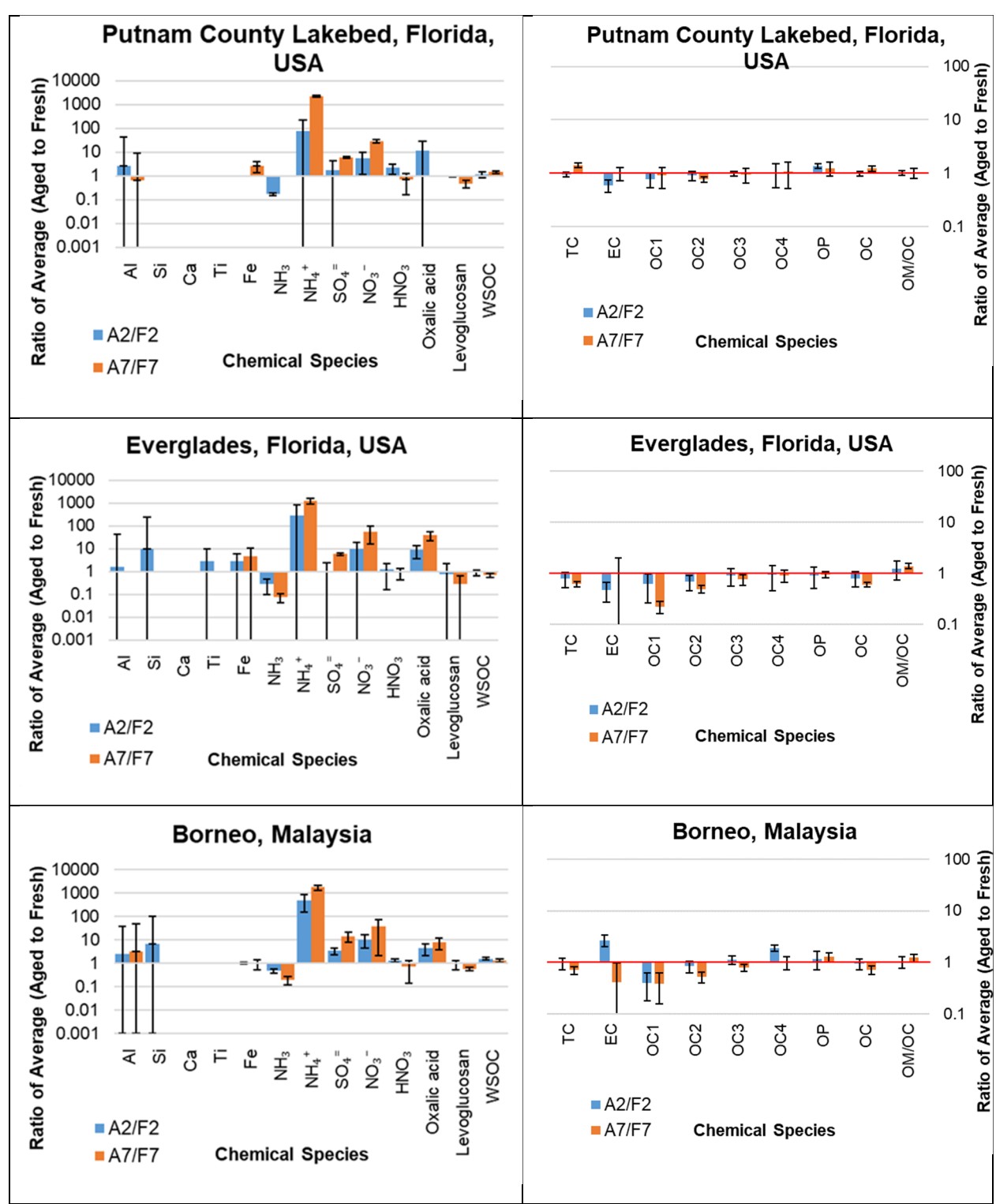

Figure 3. Ratios of average Aged (A) to Fresh (F) chemical species for 2-days (A2/F2) and 7-days (A7/F7) of atmospheric aging of six types of peats. Vertical bars represent the standard deviations associated with each ratio. Note that different scales were used in the two Y axes, with 0.001 to 10,000 on the left axis and 0.1 to 100 on the right axis (species abbreviations are shown in Fig. 1; OM is organic mass).

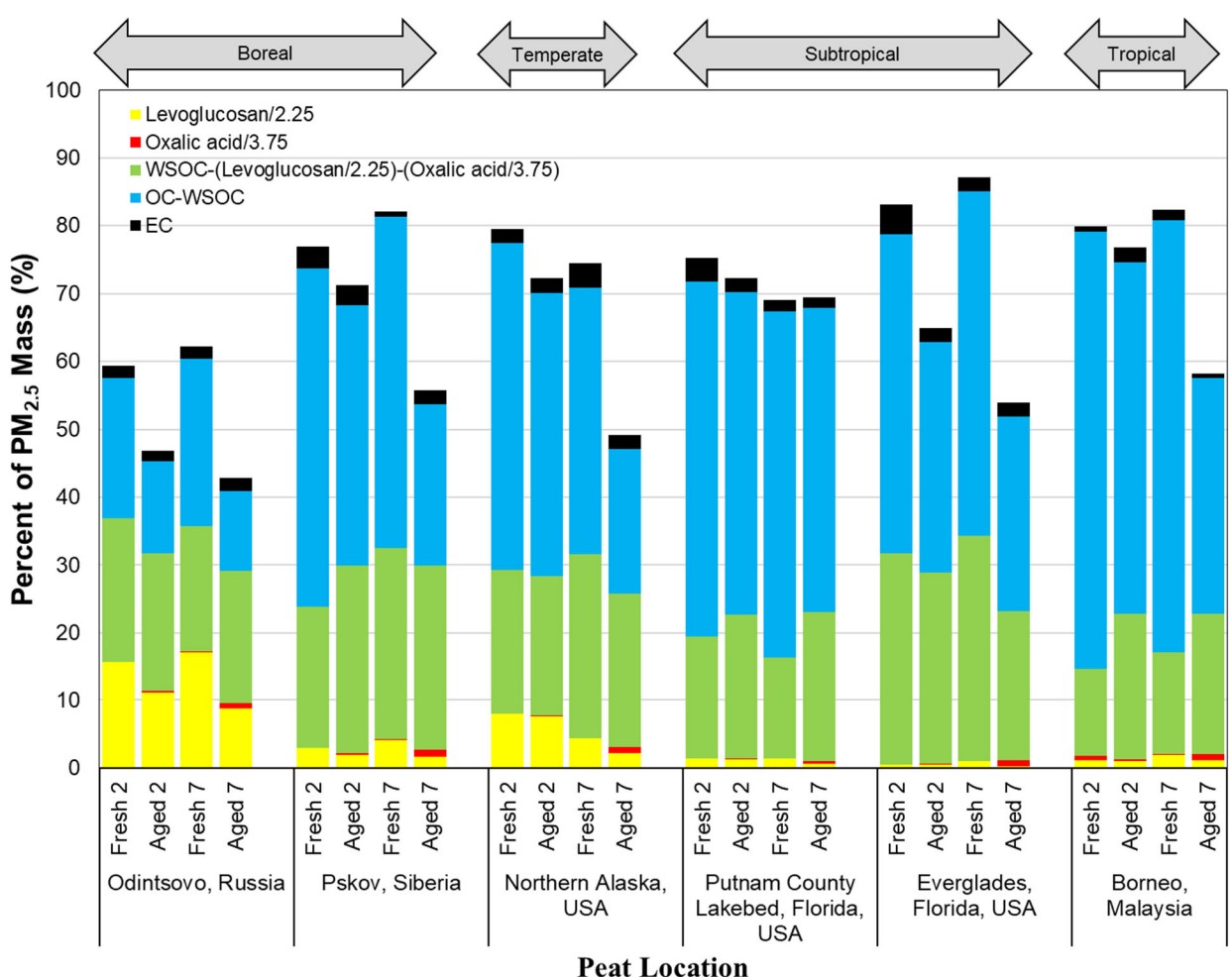

Figure 4. Abundances of fresh and aged carbon-containing components in PM2.5 (levoglucosan [C6H10O5] is divided by 2.25 and oxalic acid [C2H2O4] is divided by 3.75 to obtain the carbon content. These levels are subtracted from the water-soluble organic carbon [WSOC] to obtain the remainder, and WSOC is subtracted from organic carbon [OC] to obtain non-soluble carbon. Elemental carbon [EC] is unaltered).

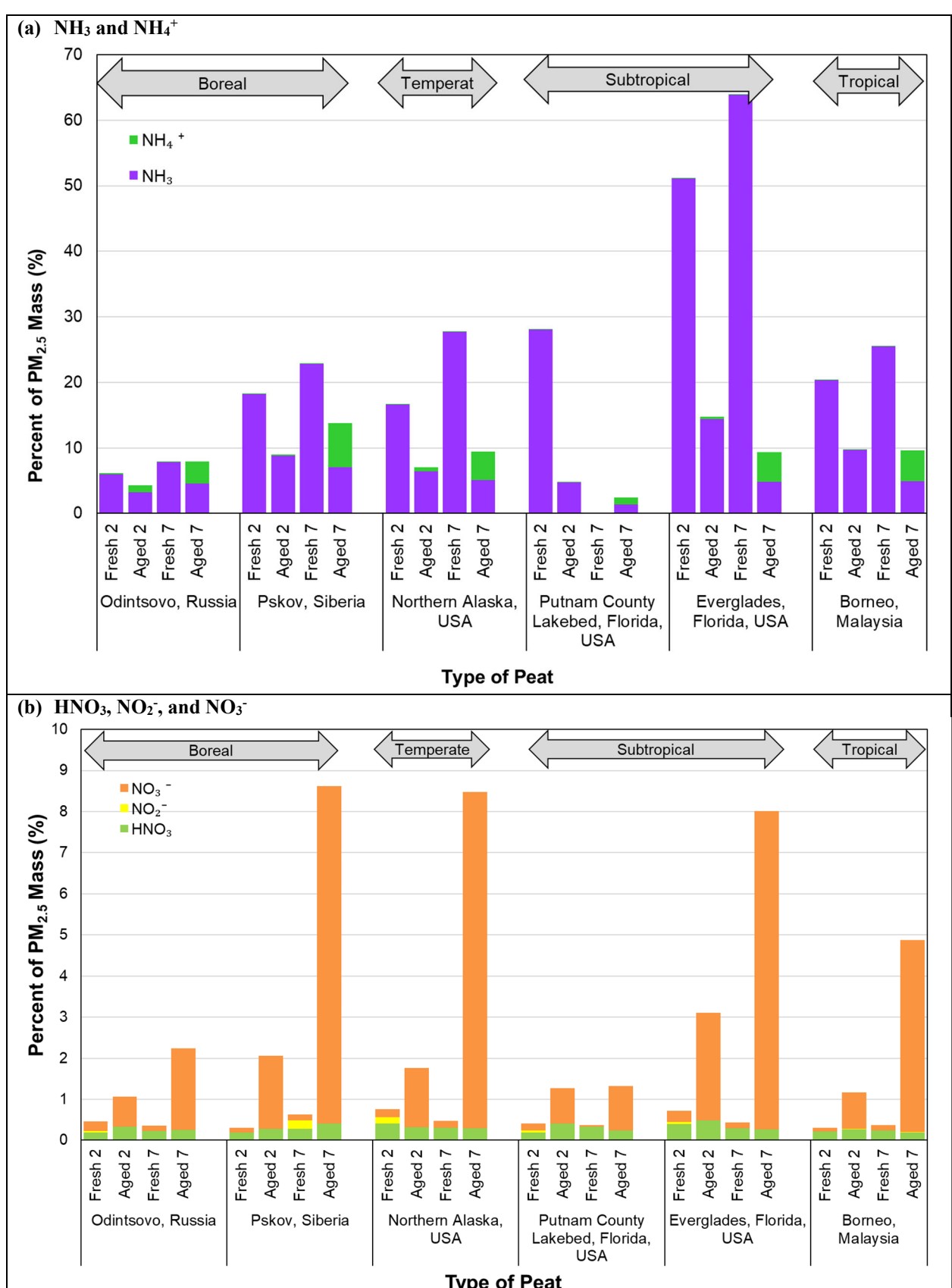

Figure 5. Comparison of nitrogen species for: a) $NH_3$ and $NH_4^+$; and b) $HNO_3$, $NO_2^-$, and $NO_3^-$
between fresh and aged profiles for six types of peats.

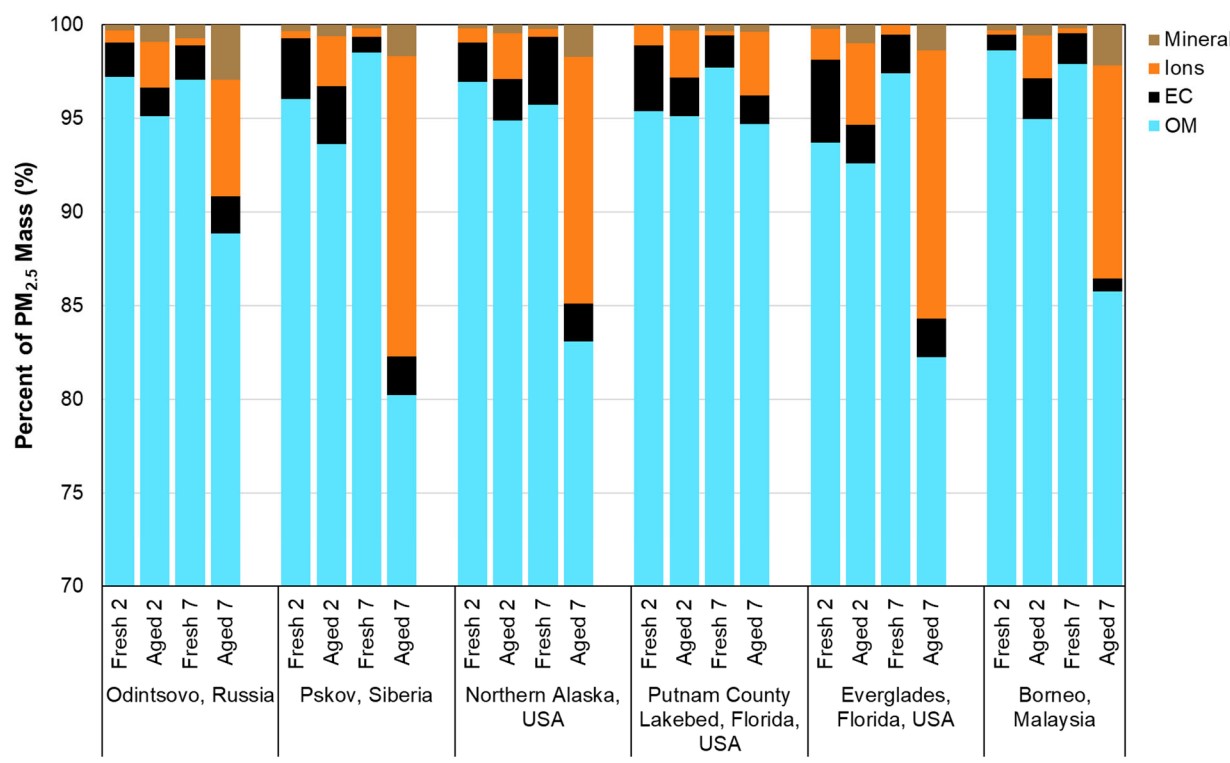

Figure 6.  Reconstruction of PM$_{2.5}$ mass with organic mass (OM, see Table 3 for OM/OC ratios),
elemental carbon (EC), major ions (i.e., sum of NH$_4^+$, NO$_3^-$, and SO$_4^=$), and mineral component
(=2.2 Al + 2.49 Si + 1.63 Ca + 1.94 Ti + 2.42 Fe) for six types of peat between fresh and aged
profiles.