# Peer review of "Changes in PM$_{2.5}$ Peat Combustion Source Profiles with Atmospheric Aging in an Oxidation Flow Reactor"

_Atmospheric Measurement Techniques, 2019_

## Referee Comment (RC1) · Anonymous Referee #1 · 15 Jul 2019

**General comments:**

This manuscript presented an extensive experimental data set for both fresh and aged PM$_{2.5}$ source profiles of smoldering-dominated combustions of peat collected from six geographically different areas, representing four main climate regions. The reported results could provide a good reference for the emission factors especially of organic and elemental carbon species before and after atmospheric aging processes, facilitating better constrained modelling studies based on receptor-oriented source apportionment analysis. However, the reasons for the similarities and differences in the corresponding source profiles of the six types of peat seem to be not well explained; and discrepancies between 2-day and 7-day aging which relate to the influence of photochemical aging on the evolution of chemical characteristics of biomass burning particles are lack of further interpretation. Details about the oxidation experiments using the PAM-OFR are insufficiently provided, although which might have been described elsewhere. The OH exposure or the photochemical age is definitely important, while other parameters such as the initial concentration of gaseous precursors, humidity, and seed particles are also key to the heterogeneous oxidation processes. Differences in the final products and their yields are therefore dependent on the abovementioned influence factors and some wall-loss effects during different chamber experiments. In this sense, how can the authors conclude that the volatilization of SVOCs during longer aging processes would serve as the main cause for the reduction of OM abundance in PM$_{2.5}$? More convincing explanation and corresponding evidence/data are necessary.

In general, this work contains a large number of chemical data characterizing the emission factors of laboratory-generated peat smoke particles, which could be useful for air quality modeling and further application in biomass-burning-aerosol-related research fields; yet some details and further interpretations need to be clarified and improved accordingly. I would suggest a major revision on the current version before consideration for the final publication in AMT. In addition to the above concerns, some specific comments are detailed as below.

**Major comments:**

1.  As stated in the general comments, one of the major concerns is the determinant reason behind the reduction of OM abundance in $PM_{2.5}$ after an even longer photochemical aging. Further discussion would be required for the identification of the crucial influence from volatilization of SVOCs.

2.  Table 1 summarizes the average $PM_{2.5}$ source profiles for both fresh and aged peat samples of six different origins. The detailed chemical information could be quite useful. While my major concern is that why the mass fraction of WSOC in $PM_{2.5}$ decreased after experiencing 2-day or 7-day aging for most of the peat samples? Besides, the WSOC fraction of Aged 7 was almost lower than that of Aged 2; what is the reason? This seems to be opposite to the general results concluded in previous studies which suggest that the oxygenated organic fraction tends to increase with atmospheric aging processes, contributing to a higher water-soluble organic fraction as the oxygenated organics are normally more polar/water-soluble that fresh biomass burning organic compounds. More in-depth discussion is necessary.

3.  Page 9, Line 213-214: The analysis for $PM_{2.5}$ Florida peat source profiles suggested that the two subtropical profiles should not be combined with other biomes. In this sense, how should the readers understand the equivalence measures for combined 'Subtropical + Temperate', or rather 'Florida + Alaska' in Table 2, where a high P-value was also reported? Consequently, how about the related experimental data for these two types of peat source profiles? Related clarification is needed.

**Specific comments:**

1.  **Abstract**: The expression of '5 orders of magnitude' sounds confusing. Is it supposed to be the discrepancy between reactive/ionic species and the carbon

content, within ~3 orders of magnitude? Following this, the authors mentioned about 'the two distinguishable clusters' in Sect. 3.3 of Line 244, Page 10. I would agree that species abundance in $PM_{2.5}$ mass percent > 1% or 10% are distinct. However, it's not clear to me why the results around 0.1% were regarded as one distinguishable cluster, as quite few data were actually covered within this range as displayed in Figure 2. Please clarify this point accordingly.

2. **Abstract**: It's a bit strange to say 'low temperature OC'; are you trying to mean 'highly volatile OC'?

3. What is the relationship between 'Elemental Carbon (EC)' and 'EC1, EC2, EC3' in this study? I assumed the EC here was the sum of EC1+2+3, similar to that of OC; however, the mass fraction of EC is much lower than that of EC1 or EC2, as summarized in Table 1. Please provide the corresponding discussion.

4. **Figure** 3: Why does the ratio of EC of Borneo, Malaysia increase for the A2/F2 but decrease for the A7/F7 scenario, which is different from all the other types of peat? Additionally, do you have any idea on the increase of EC ratio for the A7/F7 case of Pskov, Siberia?

5. In Sect. 3.4.2, the authors used the IMPROVE soil formula by Malm et al. (1994) to calculate the mass of mineral components. How did you think of the uncertainty in such an estimation, considering that large variabilities in the corresponding mineral species even exist for the six different types of peat? Further, is it appropriate to apply an empirical equation for the US country into the conditions for different origins representing various climate regions (i.e., boreal, temperate, subtropical, and tropical)? Corresponding details are preferred.

**Technical corrections:**

1. **Abstract**, line 37: *"…the reduction of OM abundances in $PM_{2.5}$ by 3–18 % after 7

*days aging". A similar issue exist in some other sentences (e.g., Lines 238, 279, 287, 478, 502, etc.), since the 7-day here is just an equivalent duration for laboratory oxidation but not a real time period. Please check through the manuscript.*

2. Page 5, line 95: *"...and elsewhere  it is transported over long distances".*

3. Page 7, line 161: *"...A portion (0.5 cm$^2$) of the other half quartz-fiber filter ..."*

4. Page 17, line 477: *"... the majority of the TC is  OC..."*

---

## Referee Comment (RC2) · Anonymous Referee #2 · 18 Jul 2019

**Summary:**

This work describes laboratory studies to comprehensively characterize gases and particles in fresh and aged peat biomass burning organic aerosol (BBOA). A Potential Aerosol Mass Oxidation Flow Reactor (PAM-OFR) was used to oxidize peat emissions. Filter-based measurements provided PM2.5 mass concentrations, elemental concentrations, eight different thermally-resolved carbon fractions (OC, EC, pyrolyzed carbon), organic acids, water soluble organic carbon, carbohydrate concentrations, NH4, and HNO3 concentrations. Mass reconstruction and moisture content analyses are also provided.

This manuscript addresses a lack of peat BBOA related source profiles, providing a wealth of information on gas- and particle-phase peat BBOA chemical composition with and without atmospheric aging. The intercomparison between peat samples from six locations to represent different biomes is particularly novel. Although this work has the potential to be highly useful for future source apportionment studies, I do not recommend publication unless major revisions are made. In particular, there is very little information provided on PAM-OFR operation characteristics, which makes it extremely difficult to assess whether the reactor was used properly to mimic atmospherically relevant conditions. In reading this paper, it seems as though there have been two additional manuscripts submitted using this data set and/or these techniques (Watson et al., 2019, and Cao et al., 2019), and although they are repeatedly cited, they have not yet been peer reviewed/published (per the citations), so I was unable to verify if the necessary information has been provided in these works. This significantly weakens the impact of this work, since the techniques are neither verifiable nor repeatable.

Specific suggestions for improvement are provided in the following general and technical/minor comments.

**General Comments:**

1. BBOA oxidation is incredibly challenging to characterize using a PAM-OFR due to chemical and physical heterogeneity and rapid/complex kinetics. More attention should therefore be given to contextualizing the results presented here in light of PAM-OFR challenges. The PAM wiki is a useful site that provides recommendations for reactor operation (https://sites.google.com/site/pamwiki/hardware/tutorial_and_recs).
   a) In general, there is a lack of information provided regarding PAM-OFR operating conditions. What were the flow rates (and by extension residence times) through the PAM-OFR? What were dilution ratios? Were dilution ratios kept constant for samples collected before and after the PAM-OFR? Was the reactor allowed to reach steady state prior to sample collection? What were typical photon fluxes measured at each oxidation condition? Without this information, the results are entirely without context and essentially meaningless.
   b) How was the OFR calibrated for these studies (e.g., with SO2? CO? With or without BBOA?)? It seems that this is not the only manuscript to come out of this data set – is the PAM-OFR calibration procedure discussed in related articles? However, it would be good to provide even a basic description of calibration details here, perhaps in the supplement.
   c) Was external OH reactivity (OHRext ~ $\Sigma k_i c_i$, where $k_i$ is the OH reaction rate constant for species i and $c_i$ is the concentration of reacting species i) characterized in this or other studies? Peng et al. (2015, 2016) and Li et al. (2015) describe suppression of OH by interfering VOC species. The OHRext should be characterized/estimated for your system, particularly because many different VOCs generated from biomass burning can react externally with OH. It should be explicitly stated whether or not parallel gas-phase measurements (e.g., from a PTR-MS) were conducted. If so, the authors should provide some analysis and discussion about how the measured VOCs potentially interfered with their OHRext. If not, hopefully the authors attempted to remove VOCs (e.g., with VOC denuders), or, failing to at least do that, provide some discussion about the *potential* for

interference. Without any attention to this caveat of OFR experiments, the results are questionable.

d) With OFR-185, photolysis at both 254 nm and 185 nm may occur, particularly at high light intensities. Peng et al. (2016) provides a detailed examination of exposure ratios (photon flux/OHexp) that have improved understanding of the potential for photolysis for different species. I recommend examining this manuscript (particularly figures 1 and 2) and discussing the potential for photolysis under your experimental conditions. The calculation for percent interference by photolysis is straightforward and should be performed for any OFR study.

e) With OFR-185, HOx recycling can impact OH formation (Peng et al., 2015, Palm et al., 2016). As with OHRext and photolysis, the impact of HOx recycling (the removal of OH through $H_2O + h\upsilon$ (185nm) $\rightarrow$ H + OH, then $H + O_2 \rightarrow HO_2$) under the experimental conditions needs to be addressed.

f) In lines 238-240, differences in the sum of species at different levels of aging are attributed to semivolatile organic compound (SVOC) losses. Did you perform "dark" experiments (i.e., collect particles and gases through the PAM-OFR without the lights on) at any point? Particles and gases collected through the PAM might be subject to different losses compared to those collected before the PAM (Palm et al., 2016). Since you are comparing fresh and aged profiles, which were collected before and after the PAM, respectively, the potential for wall losses needs to be addressed.

g) Several estimation equations have been developed to better characterize the PAM-OFR under different operating conditions. The OFR exposures estimator (available for download at https://sites.google.com/site/pamwiki/hardware/estimation-equations) is immensely helpful for understanding how different species are expected to interfere with desired OFR chemistry. Estimation equations for LVOC condensational losses for the PAM-OFR are also available on the PAM wiki. I would suggest using these tools to better characterize PAM-OFR operating conditions and citing the sources provided therein.

2. In many places, more discussion of previous work is needed.

a) In paragraph 2 of the introduction (lines 71-81), chemical profile measurements are discussed in the context of different fresh source contributions, yet the only citation provided is Chow et al. (2002). Please provide similar citations for each of these source contributions.

b) As stated above, using a PAM-OFR to study BBOA is particularly challenging. There have been several studies that have improved the community's understanding of PAM-OFR BBOA oxidation. This manuscript would greatly benefit from further discussion of previous BBOA PAM-OFR experiments to provide further context for results. A few that come to mind include Cubison et al. (2011) and Ortega et al. (2013). Furthermore, to my knowledge, Sumlin et al. (2017) were the first to use an Aerodyne PAM-OFR to characterize both chemical and optical properties for aged and fresh peat BBOA. Given the similarity in fuel type, oxidation method, and scope of measurements, this study would provide useful context for your results in this and future publications (particularly the publication wherein UV/Vis and FTIR measurements will be discussed).

c) In line 121, it is more appropriate to cite the first description of the PAM (Kang et al., 2007) and at least the Aerodyne PAM documentation (reference 2 in this manuscript, lines 524-525) rather than your own co-authored publications, unless the PAM-OFR was modified for this study in ways described in Cao et al. (2019). I was able to verify that Watson et al. (2019; published as a discussion paper in ACPD) does not describe any PAM-OFR modifications at this point, and therefore the citations are incomplete. If Cao et al. (2019) describes modifications to the PAM-OFR, this needs to be explicitly stated.

**Technical/Minor Comments:**

1. Line 38: Either change "reconfirms" to "confirming," or change "reconfirms" to "confirms" and remove the preceding comma.
2. Lines 38-41: the use of "intermediate profile" in this sentence is confusing. Consider rewording this sentence for clarity.
3. Line 86: Consider using "improved" rather than "perfected," as there are still many remaining challenges associated with using the PAM-OFR.
4. Lines 113-116: Please revise this text to make the statement a complete sentence.
5. Line 289: Change the double-dash to a comma.
6. Table 1: Since this table is so long, I would suggest carrying the table column labels across to each page to improve table readability.
7. Figure 6: I would suggest changing the y-axis range to ~70-100 so differences in less-abundant species at the top of the bars are easier to distinguish.
8. Figure S1: The high-oxidation condition is given in the caption as 6.79 rather than 7 (as it is discussed in the manuscript) and should be changed.

**References:**

Peng, Z., Day, D. A., Stark, H., Li, R., Lee-Taylor, J., Palm, B. B., Brune, W. H. and Jimenez, J. L.: HOx radical chemistry in oxidation flow reactors with low-pressure mercury lamps systematically examined by modeling, Atmos Meas Tech, 8(11), 4863–4890, doi:10.5194/amt-8-4863-2015, 2015.

Peng, Z., Day, D. A., Ortega, A. M., Palm, B. B., Hu, W., Stark, H., Li, R., Tsigaridis, K., Brune, W. H. and Jimenez, J. L.: Non-OH chemistry in oxidation flow reactors for the study of atmospheric chemistry systematically examined by modeling, Atmos Chem Phys, 16(7), 4283–4305, doi:10.5194/acp-16-4283-2016, 2016

Li, R., Palm, B. B., Ortega, A. M., Hlywiak, J., Hu, W., Peng, Z., Day, D. A., Knote, C., Brune, W. H., de Gouw, J. A. and Jimenez, J. L.: Modeling the Radical Chemistry in an Oxidation Flow Reactor: Radical Formation and Recycling, Sensitivities, and the OH Exposure Estimation Equation, J. Phys. Chem. A, 119(19), 4418–4432, doi:10.1021/jp509534k, 2015.

Brett B. Palm, Pedro Campuzano-Jost, Amber M. Ortega, Douglas A. Day, Lisa Kaser, Werner Jud,Thomas Karl, Armin Hansel, James F. Hunter, Eben S. Cross, Jesse H. Kroll, Zhe Peng, William H. Brune,and Jose L. Jimenez, Atmos. Chem. Phys., 16, 2943-2970, 2016.

Cubison, M. J., Ortega, A. M., Hayes, P. L., Farmer, D. K., Day, D., Lechner, M. J., Brune, W. H., Apel, E., Diskin, G. S., Fisher, J. A., Fuelberg, H. E., Hecobian, A., Knapp, D. J., Mikoviny, T., Riemer, D., Sachse, G. W., Sessions, W., Weber, R. J., Weinheimer, A. J., Wisthaler, A. and Jimenez, J. L.: Effects of aging on organic aerosol from open biomass burning smoke in aircraft and laboratory studies, Atmos Chem Phys, 11(23), 12049–12064, doi:10.5194/acp-11-12049-2011, 2011.

Ortega, A. M., Day, D. A., Cubison, M. J., Brune, W. H., Bon, D., de Gouw, J. A. and Jimenez, J. L.: Secondary organic aerosol formation and primary organic aerosol oxidation from biomass-burning smoke in a flow reactor during FLAME-3, Atmos Chem Phys, 13(22), 11551–11571, doi:10.5194/acp-13-11551-2013, 2013.

Sumlin, B. J., Pandey, A., Walker, M. J., Pattison, R. S., Williams, B. J. and Chakrabarty, R. K.: Atmospheric Photooxidation Diminishes Light Absorption by Primary Brown Carbon Aerosol from Biomass Burning, Environ. Sci. Technol. Lett., doi:10.1021/acs.estlett.7b00393, 2017.

---

## Author Comment (AC1) · 20 Aug 2019

The attached zip file contains specific responses to Reviewer 1 comments as well as the modifications to the manuscript and supplement.

Please also note the supplement to this comment:
https://www.atmos-meas-tech-discuss.net/amt-2019-198/amt-2019-198-AC1-supplement.zip
* * *

---

## Author Comment (AC2) · 20 Aug 2019

The attached zip files contains detailed responses to Reviewer 2 comments along with the revised manuscript and supplement.

Please also note the supplement to this comment: https://www.atmos-meas-tech-discuss.net/amt-2019-198/amt-2019-198-AC2-supplement.zip

---

## Author Response (AR1)

**Response to Reviewer #1 Comments**

**General and Major Comments and Responses**

*General Comment 1 and Major Comment 3*

This manuscript presented an extensive experimental data set for both fresh and aged PM$_{2.5}$ source profiles of smoldering-dominated combustions of peat collected from six geographically different areas, representing four main climate regions. The reported results could provide a good reference for the emission factors especially of organic and elemental carbon species before and after atmospheric aging processes, facilitating better constrained modelling studies based on receptor-oriented source apportionment analysis. However, the reasons for the similarities and differences in the corresponding source profiles of the six types of peat seem to be not well explained. The analysis for PM$_{2.5}$ Florida peat source profiles suggested that the two subtropical profiles should not be combined with other biomes. In this sense, how should the readers understand the equivalence measures for combined 'Subtropical + Temperate', or rather 'Florida + Alaska' in Table 2, where a high P-value was also reported? Consequently, how about the related experimental data for these two types of peat source profiles? Related clarification is needed.

*Response 1*

Three performance measures (i.e., correlation coefficient [r]; percent distributions of weighted difference -- residual [R]/uncertainty [U], the R/U ratios; and Student *t*-test) are used to provide guidance in grouping or compositing the 40 sets of fresh vs aged source profiles for further comparison. These measures are useful for qualitative data interpretations. The first comparison was made between the two Florida (subtropical) profiles to examine within region variations (Table S4). This is followed by the comparisons among the four biomes (i.e., boreal, temperate, subtropical, and tropical) in Table 2 which yielded statistical differences on paired comparisons except when combining the two fresh Florida profiles.

As pointed out by the Reviewer, the two Florida profiles should not be combined with the other biomes. The revised Table 2 shows the comparisons that separate the "Subtropical" into Subtropical 1 and 2 regions to represent peats from Putnam County Lakebed (Florida-1 [FL1]) and Everglades National Park (Florida-2 [FL2]) regions. The equivalence measures show similar results with or without separating the two Florida peats, except for the abnormalities found in Putnam (FL1) peats.

Among the six tested peats, the Putnam (FL1) peat fuel with the highest carbon content (56.6 ± 0.37%) and lowest oxygen content (31.4 ± 0.36%) (see Table 1 of Watson et al. (2019)) exhibited species abundance different from the other peats. As noted in the revised text, the "sum of species" to PM$_{2.5}$ ratios decreased by 6–11% after atmospheric aging except for Putnam (FL1) peat, which shows similar mass fractions between the fresh and aged profiles. This is attributed to the lack of variations in organic carbon (OC), the largest PM$_{2.5}$ component. After atmospheric aging, the OC abundance in PM$_{2.5}$ for Putnam (FL1) peat only changes by ~0.5–1.5 %, much lower than the ~12–33% decreases for other peats; these are explained in different sections of the revised text.

Additional text revisions are added (Lines 233-265) to clarify the comparisons for "Section 3.1 Similarities and differences among peat profiles":

*The equivalence measures are used to provide guidance in compositing and comparing the 40 sets of fresh vs. aged profiles. The first comparison is made between two Florida samples from locations separated by ~485 km (i.e., Putnam County Lakebed [FL1] and Everglades National*

*Park [FL2]), representing different geological areas and land uses. Panel A of Table S4 shows that the two profiles yield high correlations (r >0.994), but are statistically different (P <0.002); with over 93 % of the chemical abundance differences within ±3σ. However, when combining both fresh Florida profiles (i.e., all Fresh 2 vs. all Fresh 7 in Panel B), statistical differences are not found, with over 98 % of abundance differences within ± 1σ and P >0.5. Notice that statistical differences are found between the two fresh Florida profiles (i.e., FL1 Fresh 2 vs. FL2 Fresh 2 and FL1 Fresh 7 vs. FL2 Fresh 7 in Panel A) with few (< 0.81 % and 5.6 %) R/U ratios exceeding 3σ; combining the two Florida profiles may cancel out some of the differences. However, paired comparisons of other combined profiles show statistical differences with low P-values (P <0.002). To further demonstrate the differences, these two Florida profiles are classified as Subtropical 1 and Subtropical 2 to compare with other biomes.*

*Similarities and differences in peat profiles by biome are summarized in Table 2. Comparisons are made for: 1) paired fresh vs. aged profiles (i.e., All Fresh vs. All Aged; Fresh 2 vs. Aged 2; and Fresh 7 vs. Aged 7); 2) different experimental tests (i.e., Fresh 2 vs. Fresh 7); and 3) two aging times (i.e., Aged 2 vs. Aged 7). Equivalence measures show that most of these profiles are highly correlated (r >0.97, mostly >0.99) but statistically different (P <0.05), with a few exceptions.*

*Group comparisons between fresh and aged samples (Panel A of Table 2) show statistical differences for all but Putnam (FL1) peat (P >0.94). This is consistent with Watson et al (2019) where atmospheric aging (7 days) reduced organic carbon EFs (i.e., $EF_{OC}$) by ~20 – 33 % for all but Putnam (FL1) peats ($EF_{OC}$ remained within ±0.5 %). As OC is a major component of $PM_{2.5}$, no apparent changes in OC and carbon fractions abundances may dictate the lack of statistical differences between the fresh and aged profiles.*

*Paired comparisons for 2-day aging (Panel B of Table 2) show no statistical differences between the Fresh 2 vs. Aged 2 Putnam (FL1) and Malaysian profiles (P >0.30 and 0.95), which may be due to the low number of samples (n=2) in the comparison; this results in no statistical differences for combined Putnam (FL1) and Malaysian peat comparison (P >0.62). Similar to the findings of combining both fresh Florida profiles (i.e., all Fresh2 vs. all Fresh 7 in Table S4), the two fresh Alaskan profiles (Fresh 2 vs. Fresh 7 in Panel D of Table 2) do not show statistical differences (P >0.12).*

**General Comment 2**

Discrepancies between 2-day and 7-day aging which relate to the influence of photochemical aging on the evolution of chemical characteristics of biomass burning particles are lack of further interpretation. Details about the oxidation experiments using the PAM-OFR are insufficiently provided, although which might have been described elsewhere. The OH exposure or the photochemical age is definitely important, while other parameters such as the initial concentration of gaseous precursors, humidity, and seed particles are also key to the heterogeneous oxidation processes.

**Response 2**

Details on oxidation experiments using the PAM-OFR are addresses in Cao et al. (2019) and have been summarized in the revised supplemental material (Section S.1). Refer to Reviewer #2 comments and responses that address this issue. Reviewer #1 is correct in that initial gaseous precursor concentrations, humidity, and seed particles are key to the heterogeneous oxidation process. However, this manuscript emphasizes the variations between fresh and aged profiles after the oxidation process, not the fundamental chemical mechanisms that control the oxidative aging.

The PAM-OFR is used to provide sufficient oxidation to enhance gas-to-particle conversion over a short time period.

Although there are many source profiles available for fresh source contributions to PM$_{2.5}$ concentrations, there is a dearth of regional source profiles to estimate pollution impacts from regional-scale sources. As smoldering peat fires produce long-lasting smoke that extend from urban- (~100 km) to regional- (~1,000 km) scales, the potential environmental impacts need to be investigated, especially in southeast Asia. As no information on PM$_{2.5}$ speciated source profiles for peat combustion is available, this manuscript pioneers the use of PAM-OFR to simulate profile aging and illustrates the changes between fresh vs. aged source profiles. As noted by the Reviewer, this work contains a large amount of chemical data characterizing the emissions of laboratory-generated peat smoke particles, which could be useful for air quality modeling and further application on biomass-burning-aerosol-related research fields.

The selected aging times are limited to the maximum flow rate through the OFR (~10 L min$^{-1}$); relatively consistent dilution ratios (~3 to 5); and short sampling duration (~50–70 minutes) to achieve optimal particle loadings (~500 µg/filter) for subsequent chemical analyses (see Table S1 for operation condition). The manuscript intends to contrast the species abundances among fresh (diluted and unaged), intermediate-aged (~2 days), and well-aged (~7 days) source profiles that mimic source profile changes during atmospheric transport between source and receptor. The actual impact on source contribution estimates using fresh vs. aged profiles in chemical mass balance (CMB) or positive matrix factorization (PMF) (e.g., Watson et al., 2016) receptor models can be calculated based on the sensitivity tests.

Differences between 2- and 7-day aging times varied by peat types. These are discussed in Sections 3.1 to 3.6 for PM$_{2.5}$ mass; sum of species to PM$_{2.5}$ mass ratios; carbon abundances (i.e., OC and thermally evolved carbon fractions), organic mass [OM]/OC ratios, water-soluble organic carbon [WSOC], carbohydrates, and organic acids); nitrogen species, sulfate, and chloride abundances; and mass reconstruction. The "ratio of average" comparison in Figure 3 depicted that longer aging time (from 2- to 7-days) resulted in additional increases in ionic species (e.g., ammonium, sulfate, and nitrate) and organic acids, but decreases in low-temperature carbon fractions (e.g., OC1 and OC2 thermally evolved at 140 and 280 °C). Since species abundances are much lower for ionic species (~0.1% of PM$_{2.5}$ mass) than those of carbon abundance (~1–10 %, see Figure 2), most of the data analyses are focused on carbon.

As much of the decreases (7–22 %) in OC abundance is attributed to changes in low temperature OC1 and OC2, new Figures S2 and S3 are added to highlight the additional degradation from 2- to 7-days of atmospheric aging. The following text are revised (Lines 328–342):

*High temperature OC3 and OC4 contain more polar and/or high molecular-weight organic components (Chen et al., 2007) that are less likely to photochemically degrade. Large fractions of pyrolized carbon (OP of 7–13 %) are also found, indicative of higher molecular-weight compounds that are likely to char (Chow et al., 2018; Chow et al., 2004; Chow et al., 2001).*

*Reduction in OC abundances after atmospheric aging is attributed mostly to decreases in low temperature OC1 and OC2 abundances in the OFR as shown in the fresh vs. aged ratios of average abundances (Fig. 3). Figure S3a shows reductions in OC1 abundances after 2- and 7-days of atmospheric aging is apparent but at a similar level: ranging from 2–10 % and 3–14 %, respectively. Additional OC1 reductions from 2- to 7-days are most apparent for Russia and*

*Everglades (FL2) peats at the 6–10 % level. Similar reductions are found for OC2 (Fig. S3b): ranging from 3–11 % and 3–12 % after the 2- and 7-days of aging, respectively. Prolonged aging times resulted in additional 4–8 % OC2 reduction for all but Russian and Putnam (FL1) peats. As oxidation of organic compounds with OH radicals is an efficient chemical aging process (Chim et al., 2018), some of the VOCs and SVOCs may have been liberated (Smith et al., 2009).*

**General Comment 3 and Major Comment 1**

How can the authors conclude that the volatilization of SVOCs during longer aging processes would serve as the main cause for the reduction of OM abundance in $PM_{2.5}$? One of the major concerns is the determinant reason behind the reduction of OM abundance in $PM_{2.5}$ after an even longer photochemical aging. Further discussion would be required for the identification of the crucial influence from volatilization of SVOCs.

**Response 3**

Oxidation of organic compounds with gas-phase OH radicals is an efficient chemical aging process (Chim et al., 2018). The losses of low temperature OC1 and OC2 after atmospheric aging suggest volatilization of low-molecular weight and high vapor pressure OC components. These are further evidenced by field and laboratory chamber experiments that showed prominent mass spectrometric wood combustion markers (e.g., fragments of levoglucosan or other anhydrous sugars, pentene, butenal, and furfuryl alcohol) in OC1 and OC2 fractions that are likely degraded during atmospheric aging (Diab et al., 2015; Grabowsky et al., 2011). This is consistent with the flow tube reactor study of squalene by Smith et al. (2009) that particles lose carbon leading to particle volatilization.

However, as profiles age, reduction in "sum of species" and OC abundances can be offset by the formation of oxygenated organics. The increases on OM/OC ratios are further clarified with the addition of new Figure S4.

As OC abundances change by oxidation and varied by peat type, OM in this study represent unmeasured mass in organic compounds. It is determined by subtracting other components (i.e., mineral, ions, and EC) from $PM_{2.5}$ mass. Therefore, the reduction of OM abundance in $PM_{2.5}$ (Figure 6) by 3–18% after 7-days of aging can be attributed to effects of increased oxygenated organics, SVOC volatilization; and an increase in ionic species. The following sentences are revised to clarify:

--Lines 354–359:

*Table 3 shows that OM/OC ratios ranged from 1.1–1.7 and 1.3–2.2 for fresh and aged profiles, respectively. The lower OM/OC ratios in fresh emissions are consistent with those reported for other types of biomass burning (Chen et al., 2007; Reid et al., 2005). Figure S4 shows a general upward trend in OM/OC ratios after atmospheric aging with additional 14–21 % increases from 2- to 7-days for all but Putnam (FL1) peat. The increase in OM/OC ratios with aging are likely due to an increase in oxygenated organics.*

--Lines 483–486:

*Although the 7-day aging time increased the OM/OC ratios (by 12–19 %), the abundances of OM in PM$_{2.5}$ are reduced (3–18 %). This can be attributed to the combined effects of increased oxygenated organics; SVOC volatilization (Smith et al., 2009); and an increase in ionic species as shown in the average aged/fresh ratios in Fig. 3.*

**Major Comment 2**

Why the mass fraction of WSOC in PM$_{2.5}$ decreased after experiencing 2-day or 7-day aging for most of the peat samples?... the WSOC fraction Aged 7 was almost lower than that of Aged 2; what is the reason?...in previous studies which suggest that the oxygenated organic fraction tends to increase with atmospheric aging processes, contributing to a higher water-soluble organic fraction as the oxygenated organics are normally more polar/water-soluble than fresh biomass burning organic compounds.

**Response 4**

The WSOC/PM$_{2.5}$ ratio is not a good indicator to understand the changes in WSOC abundances during atmospheric aging as PM$_{2.5}$ also contains non-water-soluble and non-carbonaceous aerosol. The WSOC/PM$_{2.5}$ ratios of Malaysian peat are used in the text to compare with past studies, not for paired comparison between fresh and aged profiles. To further explore changes of WSOC during aging, a new Table (now Table S6) is added. The large variabilities associated with the differences in WSOC abundance (i.e., aged minus fresh) suggest that no differences exist within ±3 standard deviations, with the exceptions of the 7-day Putnam (FL) and 2-day Malaysian peats.

As WSOC is part of the OC, the WSOC/OC ratio is a better indicator to illustrate the effect of atmospheric aging. Irrespective of decreases in levoglucosan carbon/WSOC ratios and increased oxalic acid carbon/WSOC ratios after atmospheric aging (see Figure 4), the new Figure S5 shows apparent increases in WSOC/OC ratios with higher ratios after 7-day aging for all but the two Florida peats where similar WSOC/OC ratios were found between 2- and 7-days aging. This is consistent with the analogy pointed out by the Reviewer that "… atmospheric aging results in higher fractions of WSOC". The following text is revised to clarify this (Lines 376–388).

*However, the WSOC/PM$_{2.5}$ ratio is not a good indicator of changes in WSOC abundances during atmospheric aging as PM$_{2.5}$ also contains non-water-soluble and non-carbonaceous aerosol. Table S7 shows large variabilities associated with the differences (i.e., aged minus fresh), suggesting that no differences exist within ±3 standard deviations. The only exceptions are for the 7-day Putnam (FL1) peat and 2-day Malaysian peat, where aging resulted in 7–8 % increases of WSOC abundances in PM$_{2.5}$.*

*As WSOC is part of the OC, the WSOC/OC ratio is a better indicator of atmospheric aging. WSOC/OC ratios (Table 3) vary between fresh (0.18–0.64) and aged (0.31–0.71) profiles. Figure S5 shows a general increase of WSOC/OC ratios from fresh to aged profiles. Longer aging time from 2- to 7-days results in 5–10 % higher WSOC/OC ratios for all but the two Florida peats. OC water-solubility also varies by peat type. Russian peat OC emissions are largely water-soluble, whereas Malaysian peat emissions are mostly water-insoluble, with WSOC/OC ratios of 0.59– 0.71 and 0.18–0.40, respectively.*

**Specific Comments**

**Specific Comment 1**

Abstract: The expression of '5 orders of magnitude' sounds confusing. Is it supposed to be the discrepancy between reactive/ionic species and the carbon content, within ~3 orders of

magnitude? Following this, the authors mentioned about 'the two distinguishable clusters' in Sect. 3.3 of Line 244, Page 10. I would agree that species abundance in $PM_{2.5}$ mass percent > 1% or 10% are distinct. However, it's not clear to me why the results around 0.1% were regarded as one distinguishable cluster, as quite few data were actually covered within this range as displayed in Figure 2. Please clarify this point accordingly.

***Response 5***

Table 1 shows large variations of species abundance in $PM_{2.5}$ from $10^{-5}$ to $10^1$. However, the Reviewer is right that most species varied within ~three orders of magnitude. This is clarified in the revised text.

Because of the low abundances in reactive/ionic species, only a few species were included in original Figure 2. To demonstrate the two distinguished clusters, the revised Figure 2 included additional three ions (i.e., $Na^+$, $Cl^-$, and $NO_2^-$) and three organic acids (i.e., formic acid, acetic acid, and propionic acid) that are below 1 % abundances. A 1:1 line and two circles are added to each graph in Figure 2 to delineate the two clusters. This is explained in the revised Figure 2 caption as well as in Abstract and text.

***Specific Comment 2***

Abstract: It's a bit strange to say 'low temperature OC'; are you trying to mean 'highly volatile OC'?

***Response 6***

The low temperature OC1 and OC2 are referred to thermally-evolved carbon at 140 and 280 °C following the IMPROVE_A carbon analysis protocol (Chow et al., 2007) that is applied in long-term U.S. $PM_{2.5}$ networks. This low temperature carbon is likely considered highly volatile OC. The sentence is revised (Lines 32–35) as:

*Organic carbon (OC) accounted for 58–85 % of $PM_{2.5}$ mass in fresh profiles with low EC abundances (0.67–4.4 %). OC abundances decreased by 20–33 % for well-aged profiles, with reductions of 3–14 % for the volatile OC fractions (e.g., OC1 and OC2, thermally evolved at 140 and 280 °C).*

***Specific Comment #3***

What is the relationship between "Elemental Carbon (EC) and "EC1, EC2, EC3" in this study? I assumed the EC here was the sum of EC1+2+3, similar to that of OC; however, the mass fraction of EC is much lower than that of EC1 or EC2, as summarized in Table 1. Please provide the corresponding discussion.

***Response 7***

Elemental carbon (EC) is the sum of EC1+EC2+EC3 minus pyrolyzed carbon (i.e., OP) whereas organic carbon (OC) is the sum of OC1+OC2+OC3+OC4 minus OP. Since a large fraction of OP (7–13%) are found--indicative of higher molecular-weight compounds that are likely to char, the resulting EC may be lower than those of EC fractions after OP correction. This explanation is added to the footnote of Table 1.

***Specific Comment 4***

Figure 3: Why does the ratio of EC of Borneo, Malaysia increase for the A2/F2 but decrease for the A7/F7 scenario, which is different from all the other types of peat? Additionally, do you have any idea on the increase of EC ratio for the A7/F7 case of Pskov, Siberia?

*Response 8*

For smoldering dominant peat emissions, the abundance of EC in $PM_{2.5}$ are low in the range of 0.82 to 4.4 % with no apparent changes between fresh and aged profiles. Figure 3 shows large uncertainties are associated with the A2/F2 and A7/F7 ratios. The decrease in A7/F7 ratio for Malaysian peats is mainly due to the low and variable EC abundances ($0.67 \pm 0.94$) in aged profiles. Similarly, the increase in A7/F7 ratio for Siberian peat is also due to the low and variable EC abundance ($0.83 \pm 1.30$) in aged profiles.

*Specific Comment 5*

In Sect. 3.4.2, the authors used the IMPROVE soil formula by Malm et al. (1994) to calculate the mass of mineral components. How do you think of the uncertainty in such an estimation, considering that large variabilities in the corresponding mineral species even exist for the six different types of peat? Further, is it appropriate to apply an empirical equation for the US country into the conditions for different origins representing various climate regions (i.e., boreal, temperate, subtropical, and tropical)? Corresponding details are preferred.

*Response 9*

As shown in Figure 6, mineral components only account for a small fraction (0.07–2.9 %) of $PM_{2.5}$. These variations may be due to the extent of the degraded peats (Miettinen et al., 2017) used in the experiments. The IMPROVE soil formula from Malm et al. (1994) is selected as it has been applied in many other studies (e.g., Chan et al., 1997; Pant et al., 2015; Rogula-Kozlowska et al., 2012) which provides an adequate estimate of geological mineral in reconstructed mass.

Since geological minerals are not a major component of $PM_{2.5}$, variations in the assumption regarding metal oxides or multipliers do not contribute to large variations in reconstructed mass (Chow et al., 2015). The following revisions are made to clarify this (Lines 494–498):

*The IMPROVE soil formula has been applied in many other studies (e.g., Chan et al., 1997; Pant et al., 2015; Rogula-Kozlowska et al., 2012) which provides an adequate estimate of geological mineral in reconstructed mass. Since geological minerals are not a major component of $PM_{2.5}$, variations in the assumption regarding metal oxides or multipliers do not contribute to large variations in reconstructed mass (Chow et al., 2015).*

*Technical Corrections*

1.  Abstract, line 37: "…the reduction of OM abundances in $PM_{2.5}$ by 3–18 % after 7 days aging". A similar issue exists in some other sentences (e.g., Lines 238, 279, 287, 478, 502, etc.), since the 7-day here is just an equivalent duration for laboratory oxidation but not a real time period. Please check through the manuscript.

*Response 10:* The Reviewer is correct that 7-days is an equivalent duration of laboratory oxidation, not a time period. This is clarified in the "Abstract" (Lines 25–28) and in the "Introduction" (Lines 116-118):

Lines 25-28:

*Smoke from laboratory chamber burning of peat fuels from Russia, Siberia, U.S.A. (Alaska and Florida), and Malaysia representing boreal, temperate, subtropical, and tropical regions was sampled before and after passing through a potential aerosol mass-oxidation flow reactor (PAM-OFR) to simulate intermediate-aged (~2 days) and well-aged (~7 days) source profiles.*

Lines 116-118:

*Comparisons between fresh (diluted and unaged) and aged (represent intermediate-aged [~2 days] and well-aged [~7 days] laboratory simulated oxidation with an OFR) PM$_{2.5}$ speciated profiles are made to highlight chemical abundance changes with photochemical aging.*

2. Page 5, line 95: "…and elsewhere  it is transported over long distances".

***Response 11:*** Corrected

3. Page 7, line 161: "…A portion (0.5 cm2 ) of the other half quartz-fiber filter  …"

***Response 12:*** Corrected

4. Page 17, line 477: "… the majority of the TC is  OC…"

***Response 13:*** Corrected

Table S6

Differences of WSOC abundances[a] in PM$_{2.5}$ between the aged and fresh profiles

| Peat Location | Differences and associated uncertainties between aged and fresh WSOC abundances in PM$_{2.5}$ | |
| --- | --- | --- |
| | 2-day aging | 7-day aging |
| Odintsovo, Russia | -5.17 ± 4.16[b] | -6.56 ± 6.72 |
| Pskov, Siberia | 6.04 ± 7.34 | -2.62 ±8.91 |
| Northern Alaska, USA | -0.97 ± 9.80 | -5.81 ± 11.93 |
| Putnam County Lakebed, (FL1), USA | 3.18 ± 6.44 | 6.82 ± 1.86 |
| Everglades National Park, (FL2), USA | -2.82 ± 9.30 | -11.05 ± 5.57 |
| Borneo, Malaysia | 8.26 ± 2.51 | 5.75 ± 2.90 |

[a]See Table 1 for WSOC abundances in PM$_{2.5}$.

[b]Difference in WSOC abundance= Aged minus Fresh. Plus or minus signs indicate the increase and decrease, respectively in WSOC/PM2.5 ratios after atmospheric aging; the uncertainty of the difference is based on square root of the sum of the squared uncertainties associated with each averaged profile.

[Figure]

Figure S2. Further reduction of OC abundances in PM$_{2.5}$ (~7–22%) from 2- to 7-days of aging are found for all but Putnam (FL1) peat profiles (Fresh 2 vs. Aged 2 and Fresh 7 vs. Aged 7 represent the comparison of 2- and 7-days of atmospheric aging, respectively).

[Figure]

**Two-day aging**              **Seven-day aging**

[Figure]

[Figure]

Figure S3. Reduction of low temperature OC1 (a) and OC2 (b) after 2- and 7-days of atmospheric aging. The OC1 and OC2 are carbon fractions thermally evolved at 140 and 280 °C in a helium atmosphere following IMPROVE_A thermal/optical reflectance protocol (Chow et al, 2007) that are applied in U.S. long term IMPROVE network and Chemical Speciation Network (CSN). (Fresh 2 vs. Aged 2 and Fresh 7 vs. Aged 7 represent the comparison of 2- and 7-days of atmospheric aging, respectively).

[Figure]

Figure S4. The OM/OC ratios between fresh and aged aerosol (Fresh 2 vs. Aged 2 and Fresh 7 vs. Aged 7 represent the comparison of 2- and 7-days of atmospheric aging, respectively).

[Figure]

Figure S5. Ratios of water-soluble organic carbon (WSOC) OC between fresh and aged peat profiles (Fresh 2 vs. Aged 2 and Fresh 7 vs. Aged 7 represent the comparison of 2- and 7-days of atmospheric aging, respectively)

Response to Reviewer #2 Comments

**General Comments**

***General Comment 1***

  This work describes laboratory studies to comprehensively characterize gases and particles in fresh and aged peat biomass burning organic aerosol (BBOA). A Potential Aerosol Mass Oxidation Flow Reactor (PAM-OFR) was used to oxidize peat emissions. Filter-based measurements provided PM$_{2.5}$ mass concentrations, elemental concentrations, eight different thermally-resolved carbon fractions (OC, EC, pyrolyzed carbon), organic acids, water soluble organic carbon, carbohydrate concentrations, NH$_4$, and HNO$_3$ concentrations. Mass reconstruction and moisture content analyses are also provided. This manuscript addresses a lack of peat BBOA related source profiles, providing a wealth of information on gas- and particle-phase peat BBOA chemical composition with and without atmospheric aging. The intercomparison between peat samples from six locations to represent different biomes is particularly novel.

  Although this work has the potential to be highly useful for future source apportionment studies, I do not recommend publication unless major revisions are made. In particular, there is very little information provided on PAM-OFR operation characteristics, which makes it extremely difficult to assess whether the reactor was used properly to mimic atmospherically relevant conditions. In reading this paper, it seems as though there have been two additional manuscripts submitted using this data set and/or these techniques (Watson et al., 2019, and Cao et al., 2019), and although they are repeatedly cited, they have not yet been peer reviewed/published (per the citations), so I was unable to verify if the necessary information has been provided in these works. This significantly weakens the impact of this work, since the techniques are neither verifiable nor repeatable. Specific suggestions for improvement are provided in the following general and technical/minor comments. BBOA oxidation is incredibly challenging to characterize using a PAM-OFR due to chemical and physical heterogeneity and rapid/complex kinetics. More attention should therefore be given to contextualizing the results presented here in light of PAM-OFR challenges. The PAM wiki is a useful site that provides recommendations for reactor operation (https://sites.google.com/site/pamwiki/hardware/tutorial_and_recs).

***Response 1 (Including Parts A, B, and C)***

- **Part A:** The following has been added to the Section S.1 (Experimental Details and Oxidation Flow Reactor Operation [pages S-2 to S4]) supplemental material to document the OFR approach. Excerpts are taken from this to address the subsequent comments.

  *Oxidation Flow Reactors (OFRs) intend to simulate photochemical changes in gas and particle mixtures as they age during atmospheric transport. This is accomplished by directing fresh emissions through a chamber that is illuminated with ultraviolent (UV) light to simulate the Sun's illumination of the mixture. OFRs differ from smog chambers in that the UV radiation is more intense and there is a continuous flow through the system, rather than the stagnant mixture that is examined in the smog chamber at UV levels closer to ambient levels (Hidy, 2019; Lee et al., 2009). Various OFR systems have been developed and applied (Aerodyne, 2019b; Bin Babar et al., 2017; Cazorla and Brune, 2010; Ezell et al., 2010; Huang et al., 2017; Karjalainen et al., 2016; Lambe et al., 2011; Mitroo et al., 2018; Pourkhesalian et al., 2015; Reece et al., 2017; Smith et al., 2009) since the original Teflon bag of Kang et al. (2007) that was externally illuminated with mercury vapor lamps. These units range in volume from 0.15 L (Keller and Burtscher, 2012) to 1200 L (Ezell et al., 2010) and are made from fluorinated ethylene propylene (FEP) Teflon films, stainless steel, quartz or Iridite/Anodine coated aluminum with the intent to*

1

minimize reactions with the chamber walls. Although many published articles reference the characterization and operational details of Kang et al. (2007), it is evident that there have been many changes since their initial development.

Important OFR design parameters are (Huang et al., 2017): 1) gas introduction mixing prior to and within the OFR chamber; 2) chamber volume and range of flow rates that determine residence time within the chamber; 3) reaction chamber materials that minimize artifacts (e.g., reactant adsorption and outgassing); and 4) sensors applied to detect the types of reactants and end-products. General findings are: 1) larger diameters and shorter residence times minimize gas and particle losses to chamber surfaces; 2) rapid mixing of pollutants provides more accurate reaction rate measurements; and 3) passivated conductive surfaces minimize electrostatic effects on particles. Although the Caltech Photooxidation Flow Tube reactor (Huang et al., 2017) appears to be the best characterized via modeling and experiment, the Aerodyne (2019b) potential aerosol mass (PAM)-OFR is in more widespread use owing to its compactness, reliability, expanding user-base (PAMWiki, 2019), and commercial availability. The Aerodyne OFR was used for the experiments reported here.

Figure S1 illustrates the configuration for these experiments. Two tubular low-pressure mercury (Hg) lamps in the OFR with Teflon sleeves provided UV light at 185 and 254 nm wavelengths (BHK, 2019) and two lamps with doped quartz sleeves provided illumination at 254 nm. Lamps were cooled by a continuous flow of relatively inert, nitrogen ($N_2$) gas. The main reactions for this OFR185 mode that create $O_3$, OH (hydroxyl radical), and $HO_2$ (hydroperoxyl radical) oxidants are:

$$H_2O + hv \ (185 \ nm) \ \rightarrow OH + H \tag{1}$$
$$O_2 + hv \ (185 \ nm) \ \rightarrow \ 2O(^3P) \tag{2}$$
$$O_2 + O(^3P) \rightarrow \ O_3 \tag{3}$$
$$O_3 + hv \ (254 \ nm) \rightarrow \ O_2 + O(^1D) \tag{4}$$
$$O(^1D) + H_2O \rightarrow 2OH \tag{5}$$
$$H + O_2 \rightarrow HO_2 \tag{6}$$

The OH is most influential in photochemical aging, and OH production within the OFR is related to the Hg lamp intensity, which in turn is related to the voltages applied to the lamps. Bhattarai et al. (2018) demonstrate that UV fluxes are almost linearly associated with lamp voltage from 2 to 7 V, and similar linear results were found for the profile aging tests reported here (Cao et al., 2019). OH production is related to lamp intensity by inference from first order reactions of OH with $SO_2$ which has a well-characterized rate constant ($k_{SO2,OH}= 9.49x10^{-13} \ cm^3$ molecule$^{-1}$ sec$^{-1}$ at 1 atm and 298 °K) (Davis et al., 1979; Sander et al., 2006) by the relationship:

$$OH = -1/k_{SO2,OH} \ ln \ (C_{SO2,out}/C_{SO2,in}) \tag{7}$$

where

$k_{SO2,OH}=$ reaction rate of $SO_2$ with OH (cm$^3$ molecule$^{-1}$ sec$^{-1}$)
$C_{SO2,in}=SO_2$ concentration injected into the OFR (ppb)
$C_{SO2,out}=SO_2$ concentration at the OFR outlet (ppb)

UV lamps were operated at 2 and 3.5 volts with a flow rate of 10 L min$^{-1}$ and a plug-flow residence time of ~80 s in the 13.3 L anodine-coated reactor, which translates to OH exposures ($OH_{exp}$) of ~2.6 x $10^{11}$ and 8.8 x $10^{11}$ molecules-sec cm$^{-3}$ at 2 volts and 3.5 volts, respectively. These values for $OH_{exp}$ are within the range of $1x10^{10}$ to ~$2x10^{12}$ molecules-sec cm$^{-3}$ reported in

*other OFR experiments.  The lamps were powered and brought into steady state operations before drawing the sample stream through the OFR*

*The Aerodyne OFR surface-to-volume ratio is 0.24 $cm^{-1}$, which is larger than many of the other OFR types and is intended to minimize particle and gas losses with lamp off.  $SO_2$ concentrations measured ranging from 100 to 800 ppb at the OFR inlet showed less than 1% changes when measured at the OFR outlet.  Similar results were found for carbon monoxide (CO) and ozone ($O_3$), indicating minimal losses to the reactor surfaces.  This is in contrast to the Teflon bag of Kang et al. (2007) that experienced $SO_2$ losses as high as 20%.  Lambe et al.(2011) found transmission efficiencies of 0.91±0.09 for $CO_2$ and 1.2±0.4 for $SO_2$ with a later quartz glass OFR design.*

*Lambe et al. (2011) found particle transmission efficiencies exceeding 80% for mobility diameters >150 nm, but as low as 40% for 50 nm particles with a quartz OFR.  Karjalainen et al. (2016) measured 60% particle losses for ~20 nm particles, ~25% particle losses for 50 nm particles, and <10% losses for particle sizes >100 nm with a stainless steel OFR. Palm et al. (2016) compared mass concentrations in ambient air within a forest with the same air drawn through an Aerodyne OFR and transfer lines, finding only a 4% particle loss.  Bhattarai et al. (2018) found similar results for ammonium sulfate (($NH_4$)$_2SO_4$) particles, with 50% transmission for 20 nm particles and >90% transmission for particles >100 nm.*

*For this study, UV lamp stability and linearity was determined by moving a TOCON_C6 photodiode (Sglux GmbH, Germany) detector along the central axis of the OFR and recording its readings as function of the voltage supplied to the lamps, verifying that the UV flux was linearly associated with lamp voltage from 2 to 7 V, but it was undetectable for UV <1.5V and leveled off at ~350 $\mu W$ $cm^{-2}$ in the range of 7-10 V.  Experiments were limited to 2 and 3.5 V which is well-within the linear range.  Irradiation fluxes were 2.5 x $10^{13}$ photons $cm^{-2}$ $s^{-1}$at 2 V and 12.5 x $10^{13}$ photons $cm^{-2}$ $s^{-1}$ at 3.5 V.  Fluxes were constant both in time and along the OFR axis, indicating that consistent oxidant amounts can be produced for a given voltage within the linear range, similar to the findings of Bhattarai et al. (2018).  Periodic performance tests of light intensity should be made over time as there may be some deterioration of lamp performance with use, and the measurements need to be repeated when lamps are replaced.*

*Since high $O_3$ concentrations were generated when UV lamps were on, a potassium iodide (KI) denuder (1/3 KI with 2/3 silica) was installed at the outlet of the reactor to remove over 99.99% of the $O_3$ and maintain a stable baseline of < 20 ppb.  This possibly compromised some of the potassium ion measurement in the aged profiles.*

*As discussed in the main text, the biggest uncertainty is not the estimation of oxidant exposure in the OFR, but the conversion of this exposure to atmospheric aging times.  Changes in the atmospheric multipollutant environment as emissions from several sources mix in the atmosphere are not represented within the OFR.  Added to this are the unknown effects of the high oxidant exposures within the OFR relative to atmospheric exposures and the wide variability of atmospheric OH from the assumed 1.5x$10^6$ molecules $cm^{-3}$ which is commonly, but not universally, used to translate OH exposure to atmospheric aging.*

- **Part B:** While the Reviewer approaches this from the perspective of an OFR expert, our results are more directed toward the receptor-oriented source apportionment community. The OFR portion of the experiment is not the controlling uncertainty in terms of source profiles.  We do not maintain that our results or our approach are the only ways to account

for profile aging, but they do fill a needed knowledge gap. To this end we add the following context (Lines 90-110) in the "Introduction" section:

[revised manuscript text omitted]

**General Comment 2**

In general, there is a lack of information provided regarding PAM-OFR operating conditions. What were the flow rates (and by extension residence times) through the PAM-OFR? What were dilution ratios? Were dilution ratios kept constant for samples collected before and after the PAM-OFR? Was the reactor allowed to reach steady state prior to sample collection? What were typical photon fluxes measured at each oxidation condition? Without this information, the results are entirely without context and essentially meaningless.

**Response 2**

As noted in the above revisions, the air stream extracted from the burn chamber was diluted with clean air by factors of 3 to 5, flow rate through the OFR was 10 L $min^{-1}$, which corresponded to an 80 s plug flow aging time, and the UV lamps were warmed up to steady state prior to each burn. Photon fluxes are also specified in the revisions.

**General Comment 3**

How was the OFR calibrated for these studies (e.g., with SO2? CO? With or without BBOA?)? It seems that this is not the only manuscript to come out of this data set – is the PAM-OFR calibration procedure discussed in related articles? However, it would be good to provide even a basic description of calibration details here, perhaps in the supplement.

**Response 3**

As noted in the above revisions to the supplemental material, OH concentrations were estimated from $SO_2$ decay following the procedures recommended by past studies and using known reaction constants..

**General Comment 4**

Was external OH reactivity (OHRext ~ $\Sigma k_i c_i$, where $k_i$ is the OH reaction rate constant for species i and $c_i$ is the concentration of reacting species i) characterized in this or other studies? Peng et al. (2015, 2016) and Li et al. (2015) describe suppression of OH by interfering VOC species. The OHRext should be characterized/estimated for your system, particularly because many different VOCs generated from biomass burning can react externally with OH. It should be explicitly stated whether or not parallel gas-phase measurements (e.g., from a PTR-MS) were conducted. If so, the authors should provide some analysis and discussion about how the measured

VOCs potentially interfered with their OHRext. If not, hopefully the authors attempted to remove VOCs (e.g., with VOC denuders), or, failing to at least do that, provide some discussion about the potential for interference. Without any attention to this caveat of OFR experiments, the results are questionable.

***Response 4***

Potential $OH_{exp}$ is recognized in revisions to the experimental section of the main text, and it is noted that the detailed VOC data needed for this was not available for these experiments. A case is made that this is not the controlling uncertainty for converting OH exposure to aging times, as the $1.5x10^6$ molecules $cm^{-3}$ atmospheric concentration is unjustifiably assumed by many articles that translate $OH_{exp}$ to days of aging. Whether the aging is 1 to 3 days for nearby pollution sources or 5 to 8 days for distant regional sources doesn't matter for source apportionment purposes at our current understanding of profile aging. The large differences between fresh, intermediate-aged, and well-aged profiles is readily apparent from the comparisons.

***General Comment 5***

With OFR-185, photolysis at both 254 nm and 185 nm may occur, particularly at high light intensities. Peng et al. (2016) provides a detailed examination of exposure ratios (photon flux/OHexp) that have improved understanding of the potential for photolysis for different species. I recommend examining this manuscript (particularly figures 1 and 2) and discussing the potential for photolysis under your experimental conditions. The calculation for percent interference by photolysis is straightforward and should be performed for any OFR study.

***Response 5***

With due respect for the Reviewer's OFR expertise, we refer to Response 4. This is not a study of the OFR, but of potential source profile changes.

***General Comment 6***

With OFR-185, HOx recycling can impact OH formation (Peng et al., 2015, Palm et al., 2016). As with OHRext and photolysis, the impact of HOx recycling (the removal of OH through $H2O + h\upsilon$ (185nm) $\rightarrow$ H + OH, then $H + O2 \rightarrow HO2$) under the experimental conditions needs to be addressed.

***Response 6***

With due respect for the Reviewer's OFR expertise, we refer to Response 4. This is not a study of the OFR, but of potential source profile changes. This would only affect the assumed aging times, for which we have demonstrated that the controlling uncertainty derives from the large variability in ambient OH exposures.

***General Comment 7***

In lines 238-240, differences in the sum of species at different levels of aging are attributed to semivolatile organic compound (SVOC) losses. Did you perform "dark" experiments (i.e., collect particles and gases through the PAM-OFR without the lights on) at any point? Particles and gases collected through the PAM might be subject to different losses compared to those collected before the PAM (Palm et al., 2016). Since you are comparing fresh and aged profiles, which were collected before and after the PAM, respectively, the potential for wall losses needs to be addressed.

*Response 7*

The above revisions describe experiments with CO, SO$_2$, and O$_3$ transmission through the reactor without UV radiation indicating negligible losses for these gases compared with earlier tests for Teflon and quartz surfaces. Apparently the passivated coating is effective. Tests by others are cited that show minimal particle losses.

*General Comment 8*

Several estimation equations have been developed to better characterize the PAM-OFR under different operating conditions. The OFR exposures estimator (available for download at https://sites.google.com/site/pamwiki/hardware/estimation-equations) is immensely helpful for understanding how different species are expected to interfere with desired OFR chemistry. Estimation equations for LVOC condensational losses for the PAM-OFR are also available on the PAM wiki. I would suggest using these tools to better characterize PAM-OFR operating conditions and citing the sources provided therein.

*Response 8*

With due respect for the Reviewer's OFR expertise, we refer to Response 4. This is not a study of the OFR, but of potential source profile changes. This would only affect the assumed aging times, for which we have demonstrated that the controlling uncertainty derives from the large variability in ambient OH exposures.

*General Comment 9*

In many places, more discussion of previous work is needed. In paragraph 2 of the introduction (lines 71-81), chemical profile measurements are discussed in the context of different fresh source contributions, yet the only citation provided is Chow et al. (2002). Please provide similar citations for each of these source contributions.

*Response 9*

The paragraph is revised as follows (Lines 63–77):

*Many of these source profiles are compiled in country-specific source profile data bases (Cao, 2018; CARB, 2019; Liu et al., 2017b; Mo et al., 2016; Pernigotti et al., 2016; U.S.EPA, 2019) and have been widely used for source apportionment and speciated emission inventories.*

*Chemical profiles measured at the source have been sufficient to identify and quantify nearby, and reasonably fresh, source contributions. These source types include gasoline- and diesel-engine exhaust, biomass burning, cooking, industrial processes, and fugitive dust. Ambient VOC and PM concentrations have been reduced as a result of control measures applied to these sources, and additional reductions have been implemented for toxic materials such as lead, nickel, vanadium, arsenic, diesel particulate matter, and several organic compounds. As these fresh emission contributions in neighborhood- and urban-scale environments (Chow et al., 2002) decrease, regional-scale contributions that may have aged for intermediate (~2 days) or long (~7 days) periods prior to arrival at a receptor gain in importance. These profiles experience augmentation and depletion of chemical abundances owing to photochemical reactions among their gases and particles, as well as interactions upon mixing with other source emissions.*

*General Comment 10*

As stated above, using a PAM-OFR to study BBOA is particularly challenging. There have been several studies that have improved the community's understanding of PAMOFR BBOA oxidation. This manuscript would greatly benefit from further discussion of previous BBOA PAM-OFR experiments to provide further context for results. A few that come to mind include Cubison

et al. (2011) and Ortega et al. (2013). Furthermore, to my knowledge, Sumlin et al. (2017) were the first to use an Aerodyne PAM-OFR to characterize both chemical and optical properties for aged and fresh peat BBOA. Given the similarity in fuel type, oxidation method, and scope of measurements, this study would provide useful context for your results in this and future publications (particularly the publication wherein UV/Vis and FTIR measurements will be discussed).

**Response 10**

The focus of this manuscript is on peat burning, not on all biomass burning. The manuscript is not intended to be a review of all work using OFRs, although this would be a useful contribution. The companion manuscript of Cao et al. (2019), which has unfortunately been under review for over three months, elaborated on this. A table (Table A) from that manuscript summarizing OFR uses for emissions testing is attached.

**General Comment 11**

In line 121, it is more appropriate to cite the first description of the PAM (Kang et al., 2007) and at least the Aerodyne PAM documentation (reference 2 in this manuscript, lines 524–525) rather than your own co-authored publications, unless the PAM-OFR was modified for this study in ways described in Cao et al. (2019). I was able to verify that Watson et al. (2019; published as a discussion paper in ACPD) does not describe any PAM-OFR modifications at this point, and therefore the citations are incomplete. If Cao et al. (2019) describes modifications to the PAM-OFR, this needs to be explicitly stated.

**Response 11**

Changes are made in both the Introduction (Lines 101–110) and Experiment (Lines 124-133) sections:

Lines 101-110:

*Changes in source profiles have been demonstrated in large smog chambers (Pratap et al., 2019), wherein gas/particle mixtures are illuminated with ultraviolet (UV) light for several hours and their end products are measured. Such chambers are specially constructed and limited to laboratory testing. A more recent method for simulating such aging is the oxidation flow reactor (OFR), based on the early studies of Kang et al. (2007), revised and improved by several researchers (e.g., Jimenez, 2018; Lambe et al., 2011), and commercially available from Aerodyne (2019a, b). Although the Aerodyne potential aerosol mass (PAM)-OFR has many limitations, as explained in the supplemental material (Section S.1), it is a practical method for understanding how profiles might change with different degrees of atmospheric aging. A growing users group (PAMWiki, 2019) provides increasing knowledge of its characteristics and operations.*

Lines 124-133:

*The supplemental material describes sampling configuration shown in Fig. S1 and OFR operation. Briefly, peat smoke generated in a laboratory combustion chamber (Tian et al., 2015) was diluted with clean air (by factors of three to five) to allow for nucleation and condensation at ambient temperatures (Watson et al., 2012). These diluted emissions were then passed through an unmodified Aerodyne PAM-OFR in the OFR185 mode without ozone ($O_3$) injection. Hydroxyl radical (OH) production as a function of UV lamp voltage was estimated by inference from sulfur dioxide ($SO_2$) decay using well-established rate constants. UV lamps were operated at 2 and 3.5 volts with a flow rate of 10 L min$^{-1}$ and a plug-flow residence time of ~80 s in the*

*13.3 L anodine-coated reactor, which translates to OH exposures (OH$_{exp}$) of ~2.6 x 10$^{11}$ and ~8.8 x 10$^{11}$ molecules-sec cm$^{-3}$ at 2 volts and 3.5 volts, respectively.*

**Technical/Minor Comments:**

1. Line 38: Either change "reconfirms" to "confirming," or change "reconfirms" to "confirms" and remove the preceding comma.

   **Response:** This paragraph in Abstract has been revised as follows (Lines 32-37):

   *Organic carbon (OC) accounted for 58–85 % of PM$_{2.5}$ mass in fresh profiles with low EC abundances (0.67–4.4 %). OC abundances decreased by 20–33 % for well-aged profiles, with reductions of 3–14 % for the volatile OC fractions (e.g., OC1 and OC2, thermally evolved at 140 and 280 °C). Ratios of organic matter (OM) to OC abundances increased by 12–19 % from intermediate- to well-aged smoke. Ammonia (NH$_3$) to PM$_{2.5}$ ratios decreased after intermediate aging.*

2. Lines 38-41: the use of "intermediate profile" in this sentence is confusing. Consider rewording this sentence for clarity.

   **Response:** The aging time is discussed in the first two paragraphs of the Abstract as shown in the following revised sentences (Lines 25-37):

   *Smoke from laboratory chamber burning of peat fuels from Russia, Siberia, U.S.A. (Alaska and Florida), and Malaysia representing boreal, temperate, subtropical, and tropical regions was sampled before and after passing through a potential aerosol mass-oxidation flow reactor (PAM-OFR) to simulate intermediate-aged (~2 days) and well-aged (~7 days) source profiles. Species abundances in PM$_{2.5}$ between aged and fresh profiles varied by several orders of magnitude with two distinguishable clusters, centered around 0.1% for reactive and ionic species and centered around 10 % for carbon.*

   *Organic carbon (OC) accounted for 58–85 % of PM$_{2.5}$ mass in fresh profiles with low EC abundances (0.67–4.4 %). OC abundances decreased by 20–33 % for well-aged profiles, with reductions of 3–14 % for the volatile OC fractions (e.g., OC1 and OC2, thermally evolved at 140 and 280 °C). Ratios of organic matter (OM) to OC abundances increased by 12–19 % from intermediate- to well-aged smoke. Ammonia (NH$_3$) to PM$_{2.5}$ ratios decreased after intermediate aging.*

3. Line 86: Consider using "improved" rather than "perfected," as there are still many remaining challenges associated with using the PAM-OFR.

   **Response:** Corrected

4. Lines 113-116: Please revise this text to make the statement a complete sentence.

   **Response:** The revised sentences are as follows (Lines 118-122).

   *The objectives of this study are to: 1) evaluate similarities and differences among the peat source profiles from four biomes; 2) examine the extent of gas-to-particle oxidation and volatilization between 2- and 7-days of simulated atmospheric aging; and 3) characterize carbon and nitrogen properties in peat combustion emissions.*

5. Line 289: Change the double-dash to a comma.

   **Response:** Corrected (now Lines 330-332):

*Large fractions of pyrolized carbon (OP of 7–13 %) are also found, indicative of higher molecular-weight compounds that are likely to char (Chow et al., 2001; Chow et al., 2004; Chow et al., 2018).*

6. Table 1: Since this table is so long, I would suggest carrying the table column labels across to each page to improve table readability.

   **Response:** Table 1 has been revised to show column labels on each page

7. Figure 6: I would suggest changing the y-axis range to ~70-100 so differences in less-abundant species at the top of the bars are easier to distinguish.

   **Response:** Figure 6 is revised with y-axis of 70–100 % to highlight changes in less abundant species.

8. Figure S1: The high-oxidation condition is given in the caption as 6.79 rather than 7 (as it is discussed in the manuscript) and should be changed.

   **Response:** Corrected

[Figure]

Figure 6. Reconstruction of PM$_{2.5}$ mass with organic matter (OM, see Table 3 for OM/OC ratios), elemental carbon (EC), major ions (i.e., sum of NH$_4^+$, NO$_3^-$, and SO$_4^=$), and mineral component (=2.2 Al + 2.49 Si + 1.63 Ca + 1.94 Ti + 2.42 Fe) for six types of peat between fresh and aged profiles.

[Figure]

Figure S1. Configuration of peat combustion experimental set up. (FTIR: Fourier-transform infrared spectrometer; OFR: oxidation flow reactor; OFR lamps were operated at 2 and 3.5 volts to simulate aging of ~2 and 7 days, respectively) (Watson et al., 2019).

Table S1. Operational parameters for the 40 peat combustion tests

| Peat Type | Peat ID | Voltage[a] (V) | Aging Time (days) | Reactor Relative Humidity (%) | Dilution Ratio | Modified Combustion Efficiency (MCE) | Peat Dry Mass before Burn (g) | Peat Dry Mass after Burn (g) | Sampling Duration (minutes) | Fresh Loading µg per filter | Aged Loading µg per filter | Ratio Aged/Fresh ± Std Dev | Fresh[b] $PM_{2.5}$ Mass µg m$^{-3}$ | Aged[b] $PM_{2.5}$ Mass µg m$^{-3}$ |
|---|---|---|---|---|---|---|---|---|---|---|---|---|---|---|
| Odintsovo, Russia | PEAT030 | 2 | 2 | 35 | 3.13 | 0.76 | 16.0 | 1.0 | 44 | 361.00 | 319.00 | 0.88 ± 0.019 | 1640.91 | 1450.00 |
| | PEAT031 | 2 | 2 | 35 | 3.22 | 0.81 | 15.4 | 1.0 | 40 | 388.00 | 304.00 | 0.78 ± 0.017 | 1940.00 | 1520.00 |
| | PEAT032 | 2 | 2 | 35 | 3.22 | 0.84 | 15.1 | 1.0 | 39 | 415.00 | 444.00 | 1.07 ± 0.018 | 2128.21 | 2276.92 |
| | PEAT033 | 3.5 | 7 | 30 | 3.33 | 0.82 | 15.1 | 0.9 | 45 | 361.00 | 427.00 | 1.18 ± 0.022 | 1604.44 | 1897.78 |
| | PEAT034 | 3.5 | 7 | 26 | 2.94 | 0.79 | 15.7 | 0.7 | 41 | 464.00 | 417.00 | 0.90 ± 0.015 | 2263.41 | 2034.15 |
| | PEAT035 | 3.5 | 7 | 30 | 2.95 | 0.84 | 15.2 | 0.8 | 40 | 319.00 | 286.00 | 0.90 ± 0.022 | 1595.00 | 1430.00 |
| Pskov, Siberia | PEAT023 | 2 | 2 | 20 | 5.03 | 0.84 | 47.1 | 1.9 | 67 | 558.00 | 557.00 | 1.00 ± 0.031 | 1665.67 | 1662.69 |
| | PEAT025 | 2 | 2 | 55 | 4.71 | 0.85 | 25.8 | 1.0 | 70 | NA[d] | 257.00 | NA[d] | NA[d] | 734.29 |
| | PEAT026 | 2 | 2 | 40 | 4.68 | 0.84 | 26.5 | 1.0 | 61 | 302.00 | 187.00 | 0.62 ± 0.0062 | 990.16 | 613.11 |
| | PEAT027 | 3.5 | 7 | 40 | 4.68 | 0.87 | 25.6 | 1.0 | 52 | 206.00 | 142.00 | 0.69 ± 0.031 | 792.31 | 546.15 |
| | PEAT028 | 3.5 | 7 | 50 | 4.72 | 0.83 | 25.7 | 1.1 | 57 | 384.00 | 411.00 | 1.07 ± 0.019 | 1347.37 | 1442.11 |
| | PEAT029 | 3.5 | 7 | 35 | 4.74 | 0.85 | 26.1 | 1.1 | 68 | 256.00 | 304.00 | 1.19 ± 0.032 | 752.94 | 894.12 |
| Northern Alaska, USA | PEAT013 | 2 | 2 | 30 | 4.78 | 0.84 | 58.2 | 13.2 | 95 | 246.00 | NA[d] | NA[d] | 517.89 | NA[d] |
| | PEAT014 | 2 | 2 | 22 | 2.88 | 0.84 | 34.0 | 5.1 | 45 | 476.00 | 429.00 | 0.90 ± 0.014 | 2115.56 | 1906.67 |
| | PEAT019 | 2 | 2 | 30 | 2.70 | 0.82 | 42.2 | 6.8 | 72 | 628.00 | 659.00 | 1.05 ± 0.012 | 1744.44 | 1830.56 |
| | PEAT020 | 3.5 | 7 | 30 | 2.69 | 0.85 | 39.6 | 12.2 | 52 | 437.00 | 410.00 | 0.94 ± 0.016 | 1680.77 | 1576.92 |
| | PEAT021[c] | 3.5 | 7 | 28 | 2.78 | 0.87 | 40.7 | 13.4 | 48 | 366.00 | NA[d] | NA[d] | 1525.00 | NA[d] |
| | PEAT022 | 3.5 | 7 | 22 | 2.77 | 0.87 | 38.1 | 14.4 | 48 | 187.00 | 300.00 | 1.60 ± 0.053 | 779.17 | 1250.00 |
| Putnam County Lakebed, Florida, USA | PEAT007[c] | 2 | 2 | 40 | 5.02 | 0.57 | 41.7 | 2.5 | 84 | NA[d] | NA[c] | NA[d] | NA[d] | NA[d] |
| | PEAT008 | 2 | 2 | 25 | 5.02 | 0.65 | 40.4 | 1.8 | 73 | 706.00 | 668.00 | 0.95 ± 0.010 | 1934.25 | 1830.14 |
| | PEAT009 | 2 | 2 | 27 | 5.27 | 0.68 | 40.3 | 2.9 | 68 | 440.00 | 404.00 | 0.92 ± 0.017 | 1294.12 | 1188.24 |
| | PEAT042[e] | 2 | 2 | 36 | 5.04 | 0.72 | 37.5 | 1.9 | 65 | 382.00 | 357.00 | 0.93 ± 0.019 | 1175.38 | 1098.46 |
| | PEAT043[e] | 2 | 2 | 22 | 5.01 | 0.71 | 37.0 | 1.9 | 68 | 381.00 | 363.00 | 0.95 ± 0.019 | 1120.59 | 1067.65 |
| | PEAT044[e] | 2 | 2 | 22 | 4.98 | 0.73 | 38.3 | 2.0 | 69 | 356.00 | 363.00 | 1.02 ± 0.021 | 1031.88 | 1052.17 |
| | PEAT004[c] | 3.5 | 7 | 40 | 4.89 | 0.63 | 39.6 | 1.9 | 81 | NA[d] | 594.00 | NA[d] | NA[d] | 1466.67 |
| | PEAT005 | 3.5 | 7 | 43 | 4.89 | 0.67 | 37.5 | 2.0 | 88 | 713.00 | 847.00 | 1.19 ± 0.011 | 1620.45 | 1925.00 |
| | PEAT006 | 3.5 | 7 | 44 | 4.90 | 0.58 | 38.3 | 2.5 | 91 | 648.00 | 657.00 | 1.01 ± 0.011 | 1424.18 | 1443.96 |
| Everglades National Park, Florida, USA | PEAT010 | 2 | 2 | 25 | 5.13 | 0.91 | 41.3 | 13.9 | 111 | 182.00 | 340.00 | 1.87 ± 0.062 | 327.93 | 612.61 |
| | PEAT011 | 2 | 2 | 25 | 4.10 | 0.90 | 61.2 | 21.5 | 135 | 545.00 | 487.00 | 0.89 ± 0.012 | 807.41 | 721.48 |
| | PEAT012 | 2 | 2 | 17 | 4.09 | 0.95 | 66.5 | 29.1 | 119 | 262.00 | 247.00 | 0.94 ± 0.027 | 440.34 | 415.13 |
| | PEAT015 | 2 | 2 | 30 | 3.97 | 0.87 | 31.8 | 11.0 | 55 | 227.00 | 223.00 | 0.98 ± 0.032 | 825.45 | 810.91 |
| | PEAT016 | 3.5 | 7 | 33 | 4.21 | 0.90 | 64.7 | 31.1 | 85 | 232.00 | 410.00 | 1.77 ± 0.046 | 545.88 | 964.71 |
| | PEAT017 | 3.5 | 7 | 48 | 4.03 | 0.88 | 64.2 | 16.1 | 113 | 496.00 | 971.00 | 1.96 ± 0.024 | 877.88 | 1718.58 |
| | PEAT018 | 3.5 | 7 | 40 | 4.04 | 0.89 | 61.8 | 35.2 | 57 | 225.00 | 369.00 | 1.64 ± 0.044 | 789.47 | 1294.74 |
| Borneo, Malaysia | PEAT036 | 2 | 2 | 37 | 2.97 | 0.87 | 30.3 | 9.3 | 66 | 406.00 | 322.00 | 0.79 ± 0.017 | 1230.30 | 975.76 |
| | PEAT037[c] | 2 | 2 | 42 | 2.98 | 0.82 | 29.9 | 7.0 | 69 | 368.00 | NA[d] | NA[d] | 1066.67 | NA[d] |
| | PEAT038 | 2 | 2 | 43 | 3.02 | 0.83 | 30.4 | 4.2 | 65 | 508.00 | 459.00 | 0.90 ± 0.014 | 1563.08 | 1412.31 |
| | PEAT039 | 3.5 | 7 | 42 | 3.03 | 0.82 | 29.4 | 7.6 | 61 | 343.00 | 406.00 | 1.18 ± 0.024 | 1124.59 | 1331.15 |
| | PEAT040[c] | 3.5 | 7 | 38 | 3.00 | 0.81 | 31.0 | 4.1 | 66 | 458.00 | NA[c] | NA[d] | 1387.88 | NA[d] |
| | PEAT041 | 3.5 | 7 | 38 | 3.02 | 0.81 | 31.5 | 7.0 | 71 | 419.00 | 459.00 | 1.10 ± 0.019 | 1180.28 | 1292.96 |

[a]Ultraviolet lamp voltages (OFR185 mode) were used to simulate 2- and 7-days of atmospheric aging

[c]These unpaired samples (fresh and aged, n=5) are not included in the averages by peat type

[e]Samples are with 60 % fuel moisture (n=3) and are treated separately from others (25 % fuel moisture)

Table A.  OFR source testing examples and PM enhancement after oxidation (Cao et al., 2019)

| Source/Reference | PM Enhancement[a] | Location and Time | Comments[b] |
|---|---|---|---|
| Multiple vehicle engine exhaust (Tkacik et al., 2014) | • PM increased 5 (0.5 day aging) to 10 (2-day aging) times on average, mostly due to SOA and $NH_3NO_3$.
• $NH_4NO_3$ increased twice as much as SOA. | • Fort Pitt Tunnel, Pittsburgh, PA, May 2013 | • 90-96% light duty gasoline vehicles.
• .03-9.3 days equivalent aging, assuming $3x10^6$ molecules/cm$^3$ average daily OH.
• AACSM measured major PM components |
| Gasoline Direct Injection Engine Exhaust (Karjalainen et al., 2016) | • PM increased by factor of ~22 for cold start, factor of 8 for highway driving, and factor of ~4 for highway cruising, mostly due to SOA. | • Laboratory roller dynamometer with New European Driving Cycle (NEDC) | • Stainless steel 13 L OFR185
• 2011 turbocharged 1.4L turbo-charged engine in passenger car
• 10% ethanol in low sulfur gasoline fuel (<10 ppmwS)
• ~8 days equivalent aging assuming $1.5x10^6$ molecules/cm$^3$ ambient average OH
• AMS measured major PM components |
| Diesel engine exhaust (Jathar et al., 2017) | • SOA was 12 to 25 times POA without after treatments, 80-800 time POA with after treatments. | • Laboratory engine dynamometer. Diesel and biodiesel fuels, with and without after treatment | • Aerodyne OFR185
• 0.4 to 2 days equivalent aging assuming $1.5x10^6$ molecules/cm$^3$ ambient average OH
• 4.5 L Deer4045 Powertech engine with oxidation catalyst and particulate filter
• Output sampled by AMS, SMPS, PAX |
| Heated cooking oil (Liu et al., 2017a) | • Average SOA production of 1.35±0.3 μg/min. | • Laboratory, heated various oils to 240°C | • OFR254-40 irradiation
• ~1.3 days aging assuming $1.5x10^6$ molecules/cm$^3$ ambient average OH
• AMS measured major PM components
• Filtered out primary particles prior to OFR |

| | | for 2 min. | |
|---|---|---|---|
| Wood, peat, shrub, and grass burning (Ortega et al., 2013) | • Organic aerosol mass changed from 0.8 for ponderosa pine to 2.1 times POA for sage | • Laboratory burn chamber | • OFR185 irradiation
• ~0.1 to ~5 days assuming $1.5 \times 10^6$ molecules/cm$^3$ ambient average OH
• Fuels included ponderosa pine, lodgepole pine, peat, Alaskan duff, gallberry, black spruce, pocosin, turkey oak, saw grass, wire grass, ceanothus, manzanita, white spruce, wheat straw, chamise, and sage.
• AMS measured major components. |
| Peat and biomass burning (Bhattarai et al., 2018) | • Particle numbers increased by 2.2 to 28 for the different fuels.
• OC mass decreased by 9% to 13% for peat emissions, 7% for fir/aspen emissions, and 0% for shrub emissions. | • Laboratory combustion chamber (Tian et al., 2015) | • Aerodyne OFR185
• Siberian, Florida, and Malaysian peats. High desert shrubs, Douglas fir, and aspen.
• Output sampled onto Teflon and Teflon-coated glass fiber filters with XAD backup for laboratory analyses. SMPS measured particle number and PAX measured particle absorption.
• 7-day equivalent aging assuming $1.56 \times 10^6$ molecules/cm$^3$ ambient average OH |
| Oak leaf and heartwood burning (Fortenberry et al., 2018) | • Leaf PM organic concentrations changed by 1.6 and 1.06 times after 1-3 and 6-10 days aging.
• Heartwood PM organic concentrations changed by 0.72 and 0.84 times after 1-3 and 6-10 days aging. | • Laboratory combustion chamber | • Aerodyne OFR 185
• 0, 1-3, and 6-10 day equivalent aging assuming $1.5 \times 10^6$ molecules/cm$^3$ ambient average OH
• Output sampled by AMS, SMPS, and TAG for PM characterization |
| Solid fuel cook stoves (Reece et al., 2017) | • TSF organic aerosol increased 2.5 times POA after 4 days and 2 times after 14 days
• RS organic aerosol increased 2 times | • Laboratory combustion chamber. Water boiling | • Custom built 7L OFR (Table 1)
• Three stone fire (TSF), rocket stove (RS), and forced-draft gasifier fan stove (FDGS) were tested with dry red oak wood fuel
• AACSM, PSX, SMPS, and filter samples for PM characterization
• 2 to 14 days aging, assuming $1.5 \times 10^6$ molecules/cm$^3$ |

| | | POA after 3 days and 1.5 times after 12 days. | test and cold start/sim mering cycles | |
| | | • FDGS increased 1.2 times after 3 and 11 days. | | |

[a]POA=Primary organic aerosol, SOA=Secondary Organic Aerosol.

[b]Particle measurement instruments:  AMS=Aerodyne Mass Spectrometer (various types), AACSM=Aerodyne Aerosol Chemical Speciation Monitor, TAG=Thermal desorption Aerosol Gas chromatograph, SMPS=Scanning Mobility Particle Sizer, PAX=Photoacoustic extinctiometer.